# One-Step Residual Shifting Diffusion for Image Super-Resolution via Distillation

Daniil Selikhanovych [* 1]   David Li [* 2]   Aleksei Leonov [* 3 4]   Nikita Gushchin [* 5 6]   Sergei Kushneriuk [4]
Alexander Filippov [4]   Evgeny Burnaev [5 6]   Iaroslav Koshelev [4]   Alexander Korotin [5 6]

## Abstract

Diffusion models for super-resolution (SR) produce high-quality visual results but require expensive computational costs. Despite the development of several methods to accelerate diffusion-based SR models, some (e.g., SinSR) fail to produce realistic perceptual details, while others (e.g., OSEDiff) may hallucinate non-existent structures. To overcome these issues, we present **RSD**, a new distillation method for ResShift. Our method is based on training the student network to produce images such that a new fake ResShift model trained on them will coincide with the teacher model. RSD achieves single-step restoration and outperforms the teacher by a noticeable margin in various perceptual metrics (LPIPS, CLIPIQA, MUSIQ). We show that our distillation method can surpass SinSR, the other distillation-based method for ResShift, making it on par with state-of-the-art diffusion SR distillation methods with limited computational costs in terms of perceptual quality. Compared to SR methods based on pre-trained text-to-image models, RSD produces competitive perceptual quality and requires fewer parameters, GPU memory, and training costs. We provide experimental results on various real-world and synthetic datasets, including RealSR, RealSet65, DRealSR, ImageNet, and DIV2K. We provide the code at https://github.com/Daniil-Selikhanovych/RSD.

---

[*]Equal contribution   [1]Kandinsky Lab, Moscow, Russia [2]Mohamed bin Zayed University of Artificial Intelligence, Abu Dhabi, United Arab Emirates [3]Moscow Independent Research Institute of Artificial Intelligence, Moscow, Russia [4]Luzin Research Center, Moscow, Russia [5]Applied AI Institute, Moscow, Russia [6]AXXX, Moscow, Russia. Correspondence to: Daniil Selikhanovych <selikhanovychdaniil@gmail.com>.

*Proceedings of the $43^{rd}$ International Conference on Machine Learning*, Seoul, South Korea. PMLR 306, 2026. Copyright 2026 by the author(s).

## 1. Introduction

Single image super-resolution (SR) (Irani & Peleg, 1991; Glasner et al., 2009; Dong et al., 2016) is an inverse imaging problem that reconstructs a high-resolution (HR) image from a degraded low-resolution (LR) observation. These degradations are usually *complex and unknown* in real-world scenarios involving digital single-lens reflex cameras (Ignatov et al., 2017; Cai et al., 2019; Wei et al., 2020), referred to as the blind real-world image SR problem (Real-ISR). The SR problem is highly ill-posed, and many methods have been proposed in the literature to address it.

Recently, diffusion models (DMs) have emerged as a strong alternative to GAN-based SR methods (Wang et al., 2021; Zhang et al., 2021) due to their ability to model complex data distributions (Dhariwal & Nichol, 2021). Their competitive perceptual quality for Real-ISR is supported by higher human preference scores compared to GAN-based approaches (Saharia et al., 2023; Wang et al., 2024a).

However, early DMs for SR were computationally expensive, requiring dozens or hundreds for the number of function evaluations (NFE) of the denoiser, which limits their real-time deployment on consumer devices. Subsequent work focused on methods to accelerate these models while maintaining perceptual quality. Among them, ResShift (Yue et al., 2023) achieves perceptually better results compared to state-of-the-art (SOTA) models of other classes, including GANs (Wang et al., 2021; Zhang et al., 2021) and transformers (Liang et al., 2021) while using only 15 NFE.

Unfortunately, ResShift inference remains $10\times$ times slower than GANs, as shown in ResShift (Table 2), which highlights the challenge of accelerating Real-ISR DMs while preserving perceptual quality. SinSR (Wang et al., 2024b) showed that reducing the NFE of ResShift further degrades performance and proposed a 1-step **knowledge distillation** approach based on deterministic sampling, which is inspired by DDIM (Song et al., 2021a). However, SinSR tends to produce blurred results (Figure 3), which was also reported in recent studies (Wu et al., 2024a; Chen et al., 2025; Dong et al., 2025).

Another acceleration strategy is to condition pre-trained *text-to-image* (T2I) models on the LR input using LoRA

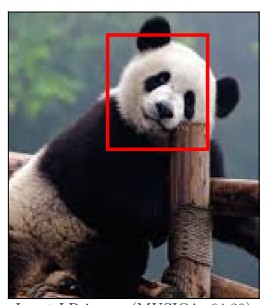
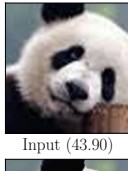
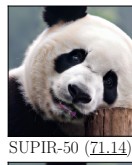
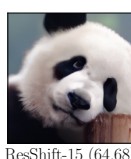
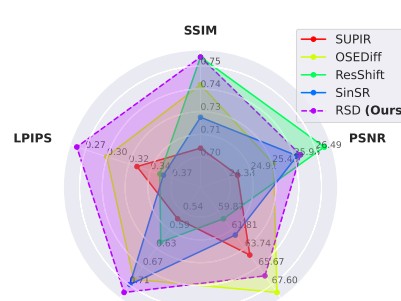

Input (43.90)   SUPIR-50 (71.14)   ResShift-15 (64.68)

Input LR image (MUSIQ↑: 64.39)   OSEDiff-1 (69.29)   SinSR-1 (69.22)   Ours-1 (**73.19**)

*Figure 1.* **Left.** A comparison between the recent diffusion methods for Real-ISR - ResShift, SinSR, OSEDiff, SUPIR - and the proposed RSD method. RSD has the following advantages: **(1)** It achieves superior perceptual quality compared to SinSR; **(2)** It requires less computational resources compared to OSEDiff; see Table 4. ("-N" behind the method name is the NFE, and the value in the bracket is MUSIQ↑ for full images). Please zoom in ×5 times for a better view. **Right.** Comparison among diffusion SR methods on RealSR. RSD (**Ours**) achieves top scores on most metrics while remaining computationally efficient compared to T2I methods - OSEDiff and SUPIR.

(Hu et al., 2022) and distill them with **variational score distillation** (VSD) (Wang et al., 2023b; Yin et al., 2024b), as proposed in OSEDiff (Wu et al., 2024a). Although this approach greatly reduced NFE from tens or even hundreds to one across the class of T2I-based SR models (Wu et al., 2024b; Yu et al., 2024; Lin et al., 2025) and achieved better perceptual results than ResShift and SinSR, T2I-based SR models exhibit notable drawbacks: they incur high training and inference costs due to having ×2.5 − 10 more parameters than SinSR, as we show in Table 4, and achieve lower full-reference fidelity metrics such as PSNR or SSIM (Wang et al., 2004) compared to ResShift and SinSR ( Table 2 and Table 3 ).

These limitations motivate the following two questions.

1. Are knowledge and variational score distillations the best methods for efficient 1-step Real-ISR DMs?

2. Can we unite the best of two worlds for SinSR and OSEDiff to achieve a 1-step diffusion SR model that has *good perceptual quality* comparable to recent T2I-based SR models, like OSEDiff, with *limited training and inference computational costs*, like SinSR, at the same time?

**Contributions**. Our main contributions are the following:

**(I) Theory**. Inspired by SinSR and recent image-to-image DM distillation methods (He et al., 2024; Gushchin et al., 2025), we propose a novel objective for 1-step distillation of diffusion SR models in **discrete time** and derive its tractable version. We show the difference between the proposed objective and the VSD objective. Motivated by ResShift's good perception-distortion trade-off across DMs and its justified diffusion process, we build our method on top of it and name it **RSD**: **R**esidual **S**hifting **D**istillation.

**(II) Practice**. We show that RSD, trained with the proposed objective, outperforms the teacher for the Real-ISR problem in multiple perceptual metrics, including LPIPS (Zhang

et al., 2018a), CLIPIQA (Wang et al., 2023a), and MUSIQ (Ke et al., 2021). We show that our discrete-time RSD objective substantially outperforms the related continuous-time IBMD objective (Gushchin et al., 2025) for the Real-ISR problem in perceptual metrics, as well as in computational training and inference costs (Appendix A.3). Our method improves the trade-off between fidelity, perceptual quality, and computational efficiency for Real-ISR with DMs in several aspects, as shown in Figure 1 and Table 4:

1. **Perceptual quality**. Compared to SinSR, the other 1-step ResShift distillation method, RSD achieves better perceptual results on synthetic and Real-ISR benchmarks.

2. **Performance-efficiency trade-off**. Compared to the T2I-based 1-step diffusion SR model of OSEDiff, our method achieves competitive perceptual quality while requiring substantially lower computational cost and budget, bringing diffusion SR closer to real-time applications.

## 2. Related Work

**GAN-based SR models**. With the rise of GANs (Goodfellow et al., 2014), GAN-based SR methods (Ledig et al., 2017; Wang et al., 2021; Zhang et al., 2021) achieved much better perceptual quality than previous regression-based approaches (Lim et al., 2017; Zhang et al., 2018b). Real-ESRGAN (Wang et al., 2021) and BSRGAN (Zhang et al., 2021) suggested complex degradation pipelines to synthesize LR-HR image pairs to model real-world data. Earlier methods assumed a pre-defined degradation (e.g., bicubic), limiting generalization. The degradations of Real-ESRGAN and BSRGAN improved the results of GAN-based Real-ISR models and have also been widely used by diffusion SR models (Yue et al., 2023; Wu et al., 2024a).

**Diffusion SR models.** SR methods, which are based on DMs (Sohl-Dickstein et al., 2015; Song & Ermon, 2019; Ho et al., 2020), can be categorized by how they utilize

the LR image. The first category uses the LR image as a condition for the denoiser (Choi et al., 2021; Rombach et al., 2022; Saharia et al., 2023; Luo et al., 2023). Another line of work argues that the Gaussian prior requires large NFE and is suboptimal for SR, where the LR image already provides structural information. Following this motivation, the second category starts the denoising in the noised LR image (Yue et al., 2023; Liu et al., 2023b). Its representative method, ResShift (Yue et al., 2023), has two advantages: **(1)** it achieves high perceptual results for blind Real-ISR using only 15 NFE and operates in the latent space of an autoencoder (Esser et al., 2021), while I2SB (Liu et al., 2023b) considers only simple degradations and requires hundreds of NFEs for denoising in the pixel space; **(2)** compared to LDM (Rombach et al., 2022), it has ×2-4 faster inference with 15 NFE only and better perception-distortion trade.

**Acceleration of diffusion SR models**. Despite superior generative performance (Dhariwal & Nichol, 2021), DMs suffer from slow inference. To mitigate this issue, various acceleration techniques have been proposed, with distillation among the most effective. These methods have also been extended to diffusion SR models. SinSR (Wang et al., 2024b) applied knowledge distillation to ResShift, achieving performance comparable to the teacher with only 1 NFE with consistency-preserving supervised loss. In our work, we draw inspiration from distillation methods that involve training an auxiliary "fake" model (Yin et al., 2024a; Huang et al., 2024; Zhou et al., 2024; Gushchin et al., 2025). We give a detailed discussion of the relation of our method to these approaches in Appendix A. CTMSR (You et al., 2025) proposed a 1-step distillation-free method, which is based on recent advances in consistency training (Song et al., 2023; Song & Dhariwal, 2024) and used the ResShift architecture.

**T2I-based SR models**. Recent studies (Wu et al., 2024a;b; Dong et al., 2025) show that ResShift and SinSR underperform in perceptual metrics and may synthesize non-realistic structures compared to SR models, which apply pre-trained T2I models. This limitation can be explained by the restricted generalization due to the lack of large-scale SR training data. In contrast, T2I models were trained on billions of natural image-text pairs and became the natural choice for Real-ISR applications. To adapt T2I models for the SR problem, such methods have two components: **(1)** LR conditioning with controllers such as LoRA layers (OSEDiff (Wu et al., 2024a), PiSA-SR (Sun et al., 2025)), ControlNet (Zhang et al., 2023) (SeeSR (Wu et al., 2024b), DiffBIR (Lin et al., 2025), SUPIR (Yu et al., 2024)) or other modules (StableSR (Wang et al., 2024a), PASD (Yang et al., 2025)); **(2)** prompts for LR images are used as predefined (StableSR, DiffBIR, TSD-SR (Dong et al., 2025)) or extracted with additional models (SeeSR, SUPIR, PASD).

**Acceleration of T2I-based SR models**. The most notable challenge in T2I-based Real-ISR models is the high computational cost, as many of them require tens or hundreds of NFE (StableSR, DiffBIR, PASD, SeeSR, SUPIR). Recent one-step diffusion distillation methods utilize variational score distillation (VSD) (Wang et al., 2023b; Yin et al., 2024b) (OSEDiff (Wu et al., 2024a)), adversarial diffusion distillation (ADD) (Sauer et al., 2025) (AddSR (Xie et al., 2024)), or target score distillation (TSD-SR (Dong et al., 2025)). These methods significantly accelerate the inference of T2I-based SR models but do not solve the problem of large T2I architectures with billion parameters. To decrease the size of T2I-based Real-ISR models, AdcSR (Chen et al., 2025) proposed the knowledge distillation method applied to OSEDiff, which is based on adversarial training of the compressed student network by removing and pruning teacher modules. The second challenge is the prediction instability depending on the noise realization, which can lead to poor fidelity and unfaithful details (Sun et al., 2024). PiSA-SR (Sun et al., 2025) proposed the T2I-based SR model, which can adjust the perception-distortion trade-off (Blau & Michaeli, 2018) during inference without re-training.

## 3. Method

We start by recalling the ResShift formulation in §3.1. Then, we propose our method for distillation of the ResShift teacher in a one-step generator and derive its computationally tractable form in §3.2. We expand the method to the multistep training in §3.3 and add additional supervised losses in §3.4. We then formulate the final objective for our RSD method in §3.5. We discuss the novelty of RSD in relation to existing distillation Real-ISR methods in §3.6.

**Remark.** While we derive our distillation method for ResShift, we note that ResShift is essentially a conditional DDPM (Ho et al., 2020), where the forward process ends in a Gaussian centered at the LR image. RSD can be generalized for any DMs built on DDPM (Appendix A.4).

### 3.1. Background

As part of diffusion models, ResShift can be described by specifying the forward (noising) process, the parameterization of the reverse (denoising) process, and the objective for training the reverse process.

**Forward process**. Consider a pair of $(\text{LR}, \text{HR})$ images $(y_0, x_0) \sim p_{\text{data}}(y_0, x_0)$. For a residual $e_0 = y_0 - x_0$, ResShift uses the forward process with the Gaussian kernel:

$$q(x_t|x_{t-1}, y_0) = \mathcal{N}(x_t|x_{t-1} + \alpha_t\, e_0, \kappa^2 \alpha_t \mathbf{I}), \quad (1)$$

where $\alpha_t = \eta_t - \eta_{t-1}$, $\alpha_1 = \eta_1$, and $\{\eta\}_{t=1}^T$ is a schedule, while $\kappa$ is a hyper-parameter controlling the noise variance.

The corresponding posterior distribution is given as:

$$q(x_{t-1}|x_t,x_0,y_0)=\mathcal{N}\left(x_{t-1}\Big|\frac{\eta_{t-1}}{\eta_t}x_t+\frac{\alpha_t}{\eta_t}x_0, \beta_t\mathbf{I}\right) \quad (2)$$

$$\beta_t \stackrel{\text{def}}{=} \frac{\kappa^2\eta_{t-1}}{\eta_t}\alpha_t \quad (3)$$

The transition distribution (1) leads to an analytically tractable marginal distribution of $q(x_t|x_0, y_0)$ at any timestep $t$:

$$q(x_t|x_0,y_0) = \mathcal{N}(x_t|x_0 + \eta_t e_0, \kappa^2\eta_t\mathbf{I}), t \in [1, T], \quad (4)$$

**Reverse process.** ResShift defines the reverse process as:

$$p_\theta(x_0|y_0) = \int p(x_T|y_0)\prod_{t=1}^T p_\theta(x_{t-1}|x_t,y_0)dx_{1:T} \quad (5)$$

Here $p(x_T|y_0) = \mathcal{N}(x_T|y_0, \kappa^2\eta_T I)$, and $p_\theta(x_{t-1}|x_t, y_0)$ is a Gaussian reverse transition kernel with parameters $\mu_\theta$ and $\Sigma_\theta$.

**Objective.** ResShift sets $\Sigma_\theta(x_t, y_0, t)$ to be independent of $x_t$ and $y_0$ and reparameterizes $\mu_\theta(x_t, y_0, t)$ as:

$$\mu_\theta(x_t,y_0,t) = \frac{\eta_{t-1}}{\eta_t}x_t + \frac{\alpha_t}{\eta_t}f_\theta(x_t,y_0,t), \quad (6)$$

where $f_\theta$ is a neural network that predicts $x_0$. The training objective of ResShift is as follows:

$$\min_\theta \sum_{t=1}^T \mathbb{E}_{p(x_0,y_0,x_t)}w_t\|f_\theta(x_t,y_0,t) - x_0\|^2, \quad (7)$$

where $w_t > 0$ and $p(x_0, y_0, x_t)$ are given by the ResShift forward process. We provide other details in Appendix J.

### 3.2. Residual Shifting Distillation (RSD)

We distill the ResShift *teacher* $f^*(x_t, y_0, t)$ into a stochastic one-step *student* generator $G_\theta$ mapping $y_0$ to $x_0$, which is parameterized as $\widehat{x}_0 = G_\theta(x_T, y_0, \epsilon)$ with $x_T \sim q(x_T|y_0)$ and $\epsilon \sim \mathcal{N}(0, I)$. We denote by $p_\theta(\widehat{x}_0|x_T, y_0)$ the distribution of $G_\theta$ produced for given $y_0, x_T$ and random $\epsilon$.

We train $G_\theta$ so that a **ResShift model $f_{G_\theta}$ trained on its outputs matches the teacher** $f^*$, assuming $f_{G_\theta} \approx f^*$ implies that the generated and real (LR, HR) distributions match:

$$f_{G_\theta} \approx f^* \Rightarrow p_\theta(y_0, x_0) \approx p_{\text{data}}(y_0, x_0) \quad (8)$$

We show that this assumption holds under mild conditions in Appendix L. Based on this assumption, we align student $G_\theta$ by producing data from the same distribution of (LR, HR) pairs as those in the training datasets for the teacher

network $f^*$:

$$\min_\theta \mathcal{L}_\theta,$$

$$\mathcal{L}_\theta \stackrel{\text{def}}{=} \sum_{t=1}^T w_t \mathbb{E}_{p_\theta(\widehat{x}_0,y_0,x_t)}\|f_{G_\theta}(x_t,y_0,t) - f^*(x_t,y_0,t)\|_2^2 \quad (9)$$

where $p_\theta(\widehat{x}_0, y_0, x_t)$ is induced by $\widehat{x}_0 = G_\theta(x_T, y_0, \epsilon)$, and the posterior $q(x_t|\widehat{x}_0, y_0)$ (4). In turn, $f_{G_\theta}(x_t, y_0, t)$ is the ResShift trained on the generator data $p_\theta(\widehat{x}_0|y_0)$. $\nabla_\theta\mathcal{L}_\theta$ includes $\nabla_\theta f_{G_\theta}(x_t, y_0, t)$, which is not tractable since back-propagation through the whole learning of the ResShift $f_{G_\theta}(x_t, y_0, t)$ is computationally infeasible, as we show in Appendix L. To solve the problem, we propose an equivalent expression of $\mathcal{L}_\theta$, which can be used for its evaluation:

**Proposition 3.1.** *Given a teacher model $f^*$, loss in Equation* (9) *can be evaluated in a tractable form:*

$$\mathcal{L}_\theta = -\min_\phi\Big\{\sum_t w_t\mathbb{E}_{p_\theta(\widehat{x}_0,y_0,x_t)}\Big(-\|f^*(x_t,y_0,t)\|_2^2+$$

$$\underbrace{\|f_\phi(x_t,y_0,t)\|_2^2 - 2\langle f_\phi(x_t,y_0,t) - f^*(x_t,y_0,t), \widehat{x}_0\rangle}_{\text{This objective } \mathcal{L}_{\text{fake}} \text{ is equivalent to training a fake model } f_\phi \text{ with objective (7)}}\Big)\Big\}$$

$$(10)$$

*Here, $f_\phi$ is an additional ResShift trained to optimize $\mathcal{L}_{\text{fake}}$ in Equation* (10) *for the estimation of $\mathcal{L}_\theta$. Furthermore, minimizing the loss in Equation* (10) *over $\phi$ is equivalent to training a "fake" ResShift using data generated by $G_\theta$.*

Thus, we solve the intractable gradient problem in Equation (9) by incorporating the fake ResShift model training into $\mathcal{L}_\theta$ (9). The proof of Proposition 3.1 is in Appendix L. In Appendix A.1, we compare the RSD loss and the VSD objective used in OSEDiff and show that $\mathcal{L}_\theta$ is equal to:

$$\mathcal{L}_\theta = \mathbb{E}_{p(y_0)}\mathcal{D}_{\text{KL}}\left(p(x_{0:T}|y_0)\,\|\,p^*(x_{0:T}|y_0)\right) \quad (11)$$

### 3.3. Multistep RSD training

To further improve the quality of RSD, we consider multi-step generator training (Yin et al., 2024a; Zhou et al., 2024). We fix a subset of $N$ timesteps $1 < t_1 < \cdots < t_N = T$ and append additional time conditioning to $G_\theta(x_t, t, y_0, \epsilon)$. We denote $\widehat{x}_0^{t_n}$ as an output of $G_\theta(x_t, t, y_0, \epsilon)$ for timestep $t = t_n$. In this setup, the generator $G_\theta$ approximates the distributions $p_\theta(\widehat{x}_0|x_{t_n}, y_0) \approx q(x_0|x_{t_n}, y_0)$ for all $t_n$ from the set instead of only $t = T$. We generate training input data $q(x_{t_n}|x_0, y_0)$ using ground-truth pairs $p_{\text{data}}(x_0, y_0)$ and the forward marginal (4), and train $G_\theta$ jointly over all $t_n$ using Proposition 3.1. Inference remains a single-step for speed; multistep training improves robustness and performance (Table 5). For consistency, we denote the output of the single-step network at the timestep $T$ by $\widehat{x}_0$.

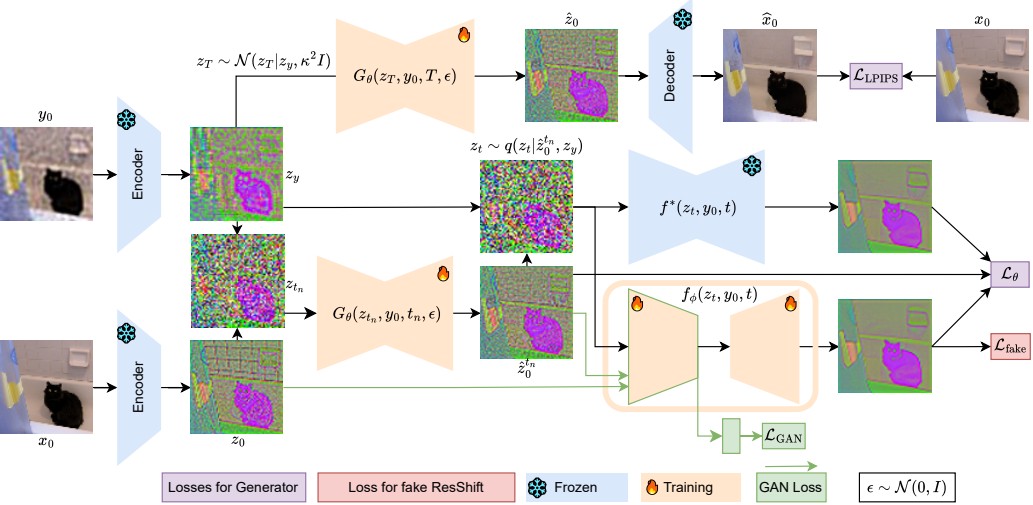

*Figure 2.* **The training framework of RSD.** First, we encode the (LR, HR) pair $(y_0, x_0)$ into latents $(z_y, z_0)$. Then, we obtain $z_{t_n}$ via the forward process (4), generate $\widehat{z}_0^{t_n}$ and sample $z_t$ using sampling (4). We process $z_t$ using both the fake and teacher ResShift models and compute the distillation losses $\mathcal{L}_\theta$ and $\mathcal{L}_{\text{fake}}$ from Proposition 3.1. For $\mathcal{L}_{\text{LPIPS}}$, we sample $z_T$ from $z_y$, generate $\widehat{z}_0$ from timestep $T$ and decode it back to obtain $\widehat{x}_0$ in pixel space. For $\mathcal{L}_{\text{GAN}}$, we use encoder features of $f_\phi$ with an additional discriminator head.

### 3.4. Supervised losses

Since teacher predictions may be biased by approximation errors in estimating $x_0$, we add supervised losses in RSD.

**LPIPS loss.** Inspired by OSEDiff, we use LPIPS ($\mathcal{L}_{\text{LPIPS}}$ (Zhang et al., 2018a)) to compare student output with HR ground truth in perceptual feature space, improving the recovery of textures and structural details beyond teacher guidance. Although OSEDiff also used MSE loss for better fidelity alignment, we found that it did not help in our setup.

**GAN loss.** Inspired by DMD2 (Yin et al., 2024a), we add a GAN loss to better match the HR distribution, using a small discriminator head on features from the fake ResShift bottleneck (Figure 2). Unlike DMD2, which compares the marginals of noised data and generator outputs, we find it more effective to compare $p_{\text{data}}(x_0|y_0)$ with $p_\theta(\widehat{x}_0^{t_n}|y_0)$ at each $t_n$:

$$\mathcal{L}_{\text{GAN}} =$$
$$\max_D \left[ \underset{p_{\text{data}}(x_0|y_0)}{\mathbb{E}} \log D\big(x_0|y_0\big) - \underset{p_\theta(\widehat{x}_0^{t_n}|y_0)}{\mathbb{E}} \log D\big(\widehat{x}_0^{t_n}|y_0\big) \right]$$
(12)

### 3.5. Putting everything together

**Translation into a latent space.** Although described in the image space ($x$), ResShift was trained in the latent space ($z$). We therefore compute $\mathcal{L}_\theta$ and $\mathcal{L}_{\text{GAN}}$ in the latent space to avoid extra encode/decode, while keeping $\mathcal{L}_{\text{LPIPS}}$ in the image space since the LPIPS network was trained there.

**Final algorithm.** The final loss function for each $t_n$ is:

$$\mathcal{L}_\theta + \lambda_1 \mathcal{L}_{\text{LPIPS}} + \lambda_2 \mathcal{L}_{\text{GAN}} \qquad (13)$$

The complete RSD algorithm with the respective notation is given in Appendix B, with an illustration in Figure 2.

### 3.6. Difference between RSD, VSD and ADD

We note that the proposed RSD loss (9) differs from the VSD loss. Specifically, we show that our method is different from VSD conceptually and computationally, and discuss the relation of the RSD loss (9) with the VSD loss (see Eq. 5 in VSD (Wang et al., 2023b)) in Appendix A.1. The key difference is that the VSD loss (Eq. (18)) aligns the marginal distributions at each timestep $t$ between the teacher's and fake's distributions, while the RSD loss (11) matches the joint distribution across all $t$. In Section 4.2, we discuss the results of RSD and VSD for the SR. In Appendix I, we discuss the difference between ADD (Sauer et al., 2025), its SR extension, AddSR (Xie et al., 2024), and RSD.

## 4. Experiments

In this section, we pursue two main goals: **(1)** to demonstrate that our proposed *distillation* method outperforms existing *distillation* methods under the same experimental setup. We chose the ResShift setup to be consistent with our teacher model and SinSR. We show our improvements compared to the current SOTA ResShift distillation method (SinSR), and we also implement and compare with VSD-based method applied to ResShift, called **ResShift-VSD** (see Appendix A.1); **(2)**, to show that RSD achieves competitive perceptual

*Table 1.* Results on real-world RealSR and RealSet65 datasets. The best and second best results are highlighted in **bold** and underline.

| Methods | T2I prior | NFE | Datasets | | | | | | |
|---|---|---|---|---|---|---|---|---|---|
| | | | *RealSR* | | | | | *RealSet65* | |
| | | | PSNR↑ | SSIM↑ | LPIPS↓ | CLIPIQA↑ | MUSIQ↑ | CLIPIQA↑ | MUSIQ↑ |
| SUPIR | yes, > 450M params | 50 | 24.38 | 0.698 | 0.331 | 0.5449 | 63.676 | 0.6133 | 66.460 |
| OSEDiff | | 1 | 25.25 | 0.737 | 0.299 | 0.6772 | 67.602 | 0.6836 | 68.853 |
| AdcSR | | 1 | 25.63 | 0.735 | 0.300 | 0.7033 | 67.550 | 0.7044 | 69.185 |
| PiSA-SR | | 1 | 25.59 | 0.750 | **0.271** | 0.6678 | 67.993 | 0.7062 | 70.208 |
| TSD-SR | | 1 | 24.88 | 0.723 | 0.281 | 0.7336 | **69.871** | 0.7263 | **70.958** |
| ResShift | no, < 180M params | 15 | **26.49** | 0.754 | 0.360 | 0.5958 | 59.873 | 0.6537 | 61.330 |
| CTMSR | | 1 | 26.18 | **0.765** | 0.294 | 0.6449 | 64.796 | 0.6893 | 67.173 |
| SinSR (distill only) | | 1 | 26.14 | 0.732 | 0.357 | 0.6119 | 57.118 | 0.6822 | 61.267 |
| SinSR | | 1 | 25.83 | 0.717 | 0.365 | 0.6887 | 61.582 | 0.7150 | 62.169 |
| ResShift-VSD (Appendix A.1) | | 1 | 23.96 | 0.616 | 0.466 | 0.7479 | 63.298 | **0.7606** | 66.701 |
| RSD (**Ours**, distill only) | | 1 | 24.92 | 0.696 | 0.355 | **0.7518** | 66.430 | 0.7534 | 68.383 |
| RSD (**Ours**) | | 1 | 25.91 | 0.754 | 0.273 | 0.7060 | 65.860 | 0.7267 | 69.172 |

*Table 2.* Results on ImageNet-Test (Yue et al., 2023). The best and second best results are highlighted in **bold** and underline.

| Methods | T2I prior | NFE | PSNR↑ | SSIM↑ | LPIPS↓ | CLIPIQA↑ | MUSIQ↑ | FID↓ |
|---|---|---|---|---|---|---|---|---|
| SUPIR | yes, > 450M params | 50 | 22.56 | 0.574 | 0.302 | **0.786** | 60.487 | 24.70 |
| OSEDiff | | 1 | 23.02 | 0.619 | 0.253 | 0.677 | 60.755 | 23.13 |
| AdcSR | | 1 | 22.99 | 0.615 | 0.252 | 0.711 | 63.218 | 34.61 |
| PiSA-SR | | 1 | 24.29 | 0.670 | 0.213 | 0.629 | 62.137 | **19.34** |
| TSD-SR | | 1 | 23.58 | 0.645 | 0.197 | 0.673 | **65.299** | 20.55 |
| ResShift | no, < 180M params | 15 | **25.01** | **0.677** | 0.231 | 0.592 | 53.660 | 30.34 |
| CTMSR | | 1 | 24.73 | 0.666 | 0.197 | 0.691 | 60.142 | 24.19 |
| SinSR (distill only) | | 1 | 24.69 | 0.664 | 0.222 | 0.607 | 53.316 | 32.13 |
| SinSR | | 1 | 24.56 | 0.657 | 0.221 | 0.611 | 53.357 | 25.85 |
| ResShift-VSD (Appendix A.1) | | 1 | 23.69 | 0.624 | 0.230 | 0.665 | 58.630 | 32.22 |
| RSD (**Ours**, distill only) | | 1 | 23.97 | 0.643 | 0.217 | 0.660 | 57.831 | 28.93 |
| RSD (**Ours**) | | 1 | 24.31 | 0.657 | **0.193** | 0.681 | 58.947 | 25.46 |

performance with recent T2I-based SR methods such as OSEDiff and SUPIR while requiring much smaller computational training and inference resources. These goals are supported by evaluations using the SinSR and OSEDiff setups. We present two types of models: RSD (**Ours**, distill only), where we used only distillation loss during training, and RSD (**Ours**), where we used additional losses (§3.4). Appendix C provides all relevant experimental details. We also compare RSD with very recent diffusion Real-ISR baselines, including CTMSR, AdcSR, PiSA-SR, and TSD-SR.

### 4.1. Experimental setup

**Training and evaluation details.** For a fair comparison, we follow the *training setup* of SinSR and ResShift, using $256 \times 256$ HR images randomly cropped from ImageNet (Deng et al., 2009) and generating LR images via the Real-ESRGAN degradations (Wang et al., 2021) with $\times 4$ SR factor. We also adopt the ResShift teacher used in SinSR. *For the evaluation*, we follow two different protocols from SinSR and OSEDiff ($\times 4$ SR factor). Following SinSR,

we use the following datasets: (1) for real-world degradations, we use full-size images from RealSR (Cai et al., 2019) and RealSet65 (Yue et al., 2023); (2) for synthetic degradations, we use ImageNet-Test (Yue et al., 2023). Following OSEDiff, we use test sets of $512 \times 512$ HR crops from StableSR (Wang et al., 2024a), including synthetic DIV2K-Val (Agustsson & Timofte, 2017) and real-world pairs from RealSR and DRealSR (Wei et al., 2020).

**Compared methods.** We consider two different experimental setups with different baseline comparisons, which follow (Wang et al., 2024b, Tables 1 and 2) and (Wu et al., 2024a, Table 1). We compare RSD against **diffusion** SR models in the main text: models with relatively small architectures (ResShift, SinSR, CTMSR) and recent T2I-based SR models, one-step OSEDiff and multistep SUPIR. We highlight that closely related models to RSD, such as ResShift, SinSR, and CTMSR, were only compared with **early T2I-based SR models**, namely LDM (Rombach et al., 2022) and StableSR. In addition to OSEDiff and SUPIR, we extend the comparison of diffusion SR methods without T2I prior to the very recent SOTA one-step T2I-based SR methods, namely Ad-

*Table 3.* Results on crops $512 \times 512$ from StableSR. The best and second best results are highlighted in **bold** and underline.

| Datasets | Methods | T2I prior | NFE | PSNR↑ | SSIM↑ | LPIPS↓ | DISTS↓ | NIQE↓ | MUSIQ↑ | MANIQA↑ | CLIPIQA↑ | FID↓ |
|---|---|---|---|---|---|---|---|---|---|---|---|---|
| DIV2K-Val | SUPIR | yes, | 50 | 22.13 | 0.5280 | 0.3923 | 0.2314 | 5.6758 | 63.82 | 0.5933 | **0.7147** | 31.46 |
| | OSEDiff | > 1.7B params | 1 | 23.72 | 0.6108 | 0.2941 | 0.1976 | **4.7097** | 67.97 | **0.6148** | 0.6683 | **26.32** |
| | ResShift | no, | 15 | 24.65 | 0.6181 | 0.3349 | 0.2213 | 6.8212 | 61.09 | 0.5454 | 0.6071 | 36.11 |
| | SinSR | < 180M params | 1 | 24.41 | 0.6018 | 0.3240 | 0.2066 | 6.0159 | 62.82 | 0.5386 | 0.6471 | 35.57 |
| | CTMSR | | 1 | **24.88** | **0.6265** | 0.3026 | 0.2040 | 5.1146 | 65.62 | 0.5165 | 0.6601 | 34.15 |
| | RSD (**Ours**) | | 1 | 23.91 | 0.6042 | **0.2857** | **0.1940** | 5.1987 | **68.05** | 0.5937 | 0.6967 | 34.84 |
| DRealSR | SUPIR | yes, | 50 | 24.93 | 0.6360 | 0.4263 | 0.2823 | 7.4336 | 59.39 | 0.5537 | 0.6799 | 164.86 |
| | OSEDiff | > 1.7B params | 1 | 27.92 | **0.7835** | **0.2968** | **0.2165** | 6.4902 | **64.65** | **0.5899** | 0.6963 | **135.30** |
| | ResShift | no, | 15 | 28.46 | 0.7673 | 0.4006 | 0.2656 | 8.1249 | 50.60 | 0.4586 | 0.5342 | 172.26 |
| | SinSR | < 180M params | 1 | 28.36 | 0.7515 | 0.3665 | 0.2485 | 6.9907 | 55.33 | 0.4884 | 0.6383 | 170.57 |
| | CTMSR | | 1 | **28.65** | 0.7834 | 0.3238 | 0.2358 | **6.1828** | 59.78 | 0.4861 | 0.6497 | 163.63 |
| | RSD (**Ours**) | | 1 | 27.40 | 0.7559 | 0.3042 | 0.2343 | 6.2577 | 62.03 | 0.5625 | **0.7019** | 167.47 |
| RealSR | SUPIR | yes, | 50 | 23.61 | 0.6606 | 0.3589 | 0.2492 | 5.8877 | 63.21 | 0.5895 | 0.6709 | 128.35 |
| | OSEDiff | > 1.7B params | 1 | 25.15 | 0.7341 | 0.2921 | **0.2128** | 5.6476 | **69.09** | **0.6326** | 0.6693 | **123.49** |
| | ResShift | no, | 15 | **26.31** | 0.7421 | 0.3421 | 0.2498 | 7.2365 | 58.43 | 0.5285 | 0.5442 | 141.71 |
| | SinSR | < 180M params | 1 | 26.28 | 0.7347 | 0.3188 | 0.2353 | 6.2872 | 60.80 | 0.5385 | 0.6122 | 135.93 |
| | CTMSR | | 1 | 25.98 | **0.7546** | 0.2897 | 0.2208 | **5.5546** | 64.26 | 0.5270 | 0.6318 | 135.35 |
| | RSD (**Ours**) | | 1 | 25.61 | 0.7420 | **0.2675** | 0.2205 | 5.7500 | 66.02 | 0.5930 | **0.6793** | 138.23 |

cSR, PiSA-SR, and TSD-SR, on SinSR evaluation datasets. In Appendix E, we provide quantitative results of other baselines, including GANs for SR, multistep T2I-based SR models, InvSR (Yue et al., 2025), and CCSR (Sun et al., 2024).

**Metrics.** Each setup employs different evaluation metrics, which we adopt from SinSR and OSEDiff. For the SinSR protocol, we report no-reference CLIPIQA and MUSIQ metrics. For the OSEDiff protocol, we report fidelity (PSNR, SSIM), full-reference perceptual metrics (LPIPS, DISTS (Ding et al., 2020)), and no-reference metrics (NIQE (Zhang et al., 2015), MANIQA-PIPAL (Yang et al., 2022), MUSIQ, CLIPIQA). PSNR and SSIM are computed on the Y channel in the YCbCr space, which follows SinSR and OSEDiff. We also report the distribution alignment metric (FID (Heusel et al., 2017)) in Tables 2-3, since ImageNet-Test and DIV2K-Val have 3k pairs, while other datasets have $\leq 100$.

### 4.2. Experimental results

**Quantitative comparisons**. The key quantitative results are summarized in Table 1 , Table 2 , and Table 3 . We group methods into (i) diffusion SR models with compact architectures (ResShift, SinSR, CTMSR, RSD) and (ii) T2I-based SR models with heavy architectures (SUPIR, OSEDiff, AdcSR, PiSA-SR, TSD-SR). We observe the following:

**(1)** RSD outperforms the teacher ResShift and our closest competitor, SinSR, by a large margin on all **perceptual** metrics (LPIPS, CLIPIQA, MUSIQ) and **all test datasets** while training on the same data. Moreover, RSD shows comparable or even better results than ResShift distilled with VSD loss, ResShift-VSD (Appendix A.1). CTMSR is the recent

one-step diffusion SR method, which also used ImageNet for training and, therefore, can be fairly compared to RSD. RSD achieves better results in **all real-world** datasets in most perceptual metrics (LPIPS, CLIPIQA, MUSIQ) with a noticeable improvement in MANIQA in Table 3.

**(2)** Compared to T2I-based OSEDiff and SUPIR models on Real-ISR benchmarks, RSD achieves the best value of the latest image-quality CLIPIQA and top-1 or top-2 results in terms of MUSIQ. RSD has a worse CLIPIQA than SUPIR for synthetic datasets but better than OSEDiff. However, SUPIR also produces excessive details, which leads to poor consistency with the LR image, as seen by the PSNR, SSIM, and LPIPS metrics. We highlight that RSD, even with slightly worse MUSIQ, achieves fidelity metrics that are much better than SUPIR and comparable to or better than OSEDiff for most setups while using a much smaller number of parameters and GPU memory, as shown in Table 4. Compared to very recent SOTA 1-step diffusion T2I-based SR methods (AdcSR, PiSA-SR, TSD-SR), RSD is capable of achieving competitive perceptual (LPIPS, CLIPIQA) and fidelity (PSNR, SSIM) quality in Tables 1-2.

**(3)** In Table 3 , we show that RSD achieves top-2 or top-1 perceptual metrics compared to OSEDiff and all DMs, which were trained on ImageNet. We highlight different training HR resolutions of RSD and OSEDiff - we used HR crops of the size $256 \times 256$ as in the teacher ResShift, while OSEDiff used HR crops of the size $512 \times 512$ for training on LSDIR (Li et al., 2023), which aligns with the crop size in Table 3 . Due to space limitations, we provide quantitative results for the recent SOTA models

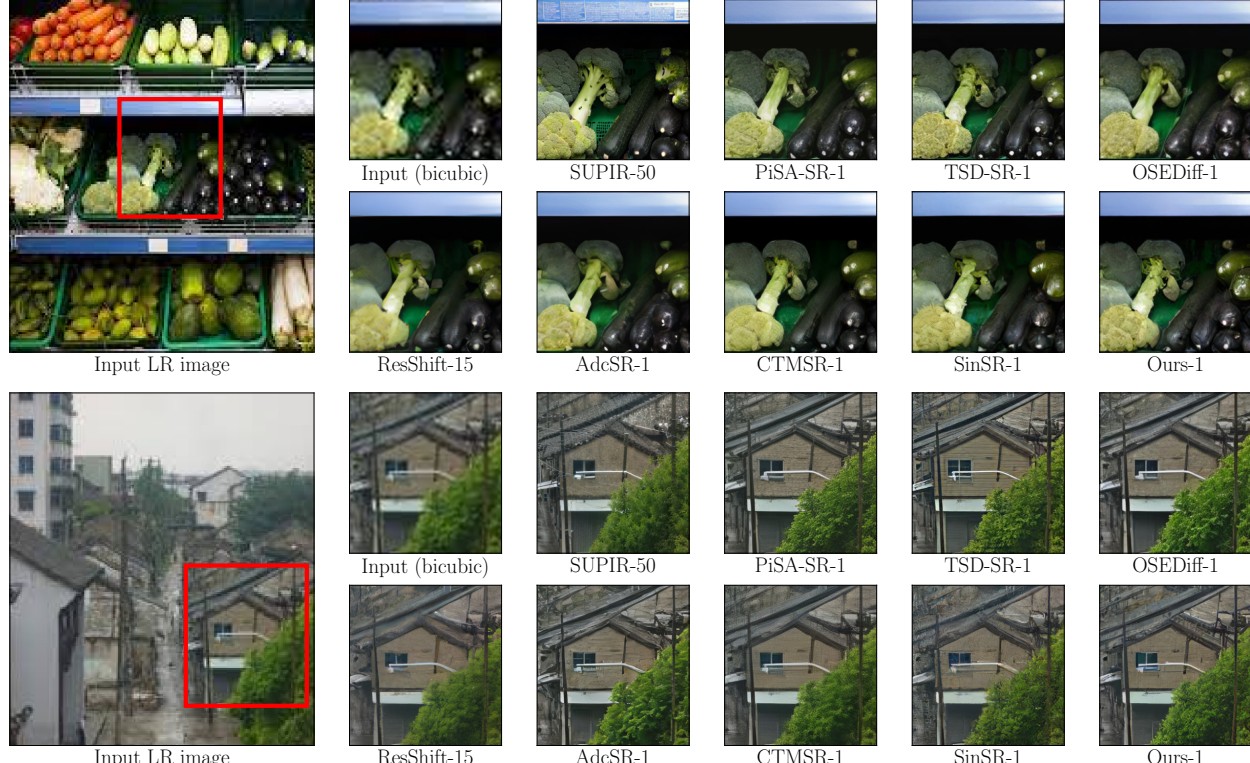

*Figure 3.* Comparison on RealSet65 (Yue et al., 2023) for diffusion SR models. Bottom images: ResShift, AdcSR, CTMSR, SinSR, and the proposed RSD. Top images: bicubic LR, SUPIR, PiSA-SR, TSD-SR, and OSEDiff. Please zoom in ×5 times for a better view.

*Table 4.* Inference complexity (NVIDIA A100, $64 \times 64$ LR, SR factor $\times 4$) and training budget. The best values are highlighted in **bold**.

| T2I prior | Yes | | | | | No | | | |
|---|---|---|---|---|---|---|---|---|---|
| Methods | SUPIR | OSEDiff | AdcSR | PiSA-SR | TSD-SR | ResShift | SinSR | CTMSR | RSD (**Ours**) |
| Inference Step (NFE) | 50 | **1** | **1** | **1** | **1** | 15 | **1** | **1** | **1** |
| Inference Time (s) | 17.704 | 0.075 | **0.024** | 0.089 | 0.074 | 0.643 | 0.060 | 0.059 | 0.059 |
| # Total Param (M) | 4801 | 1775 | 456 | 1290 | 2207 | 174 | 174 | **172** | 174 |
| Maximum GPU memory (MB) | 52535 | 3651 | 3940 | 4771 | 4611 | 1167 | 570 | 904 | **539** |
| Training time (hours / # GPU) | 240 / 64 A6000 | 24 / 4 A100 | 124 / 8 A100 | 5.5 / 4 A100 | 96 / 8 V100 | 110 / 1 A100 | 60 / 1 A100 | 58 / 4 A100 | **5 / 4 A100** |

of AdcSR, PiSA-SR, and TSD-SR for Table 3 and the Real-ISR benchmarks of RealLR200 (Wu et al., 2024b) and RealLQ250 (Ai et al., 2024) in Appendix E. We also discuss the RSD results trained on $512 \times 512$ HR images in Appendix G.

**Qualitative comparisons**. We visually compare RSD with SinSR, OSEDiff, ResShift, SUPIR, CTMSR, AdcSR, PiSA-SR, and TSD-SR on test images from RealSet65 in Figure 3. As illustrated in the top image, SUPIR tends to produce rich details that semantically do not correspond to the LR image (zoom in for excessive broccoli). ResShift, SinSR, and CTMSR produce conservative images, which may struggle with severely blurred details, like the roof of the house in the bottom image. OSEDiff may hallucinate excessive details, as can be seen in the panda's nose in Figure 1. RSD compromises between the good details of OSEDiff and SUPIR and the high fidelity of ResShift and SinSR. The

SOTA T2I-based SR models of AdcSR, PiSA-SR, and TSD-SR produce highly realistic results with rich textures that are usually better than OSEDiff. However, we found that these models can still hallucinate (Appendix F).

**Complexity comparisons**. We compare the complexity of competing diffusion SR models in Table 4, including NFE, inference time, total number of parameters, and maximum required GPU memory during inference. We also report training time and GPU usage information from the original papers. All methods are tested on an NVIDIA A100 GPU with $256 \times 256$ HR inputs following SinSR (Wang et al., 2024b, Table 3) and CTMSR (You et al., 2025, Table 3) complexity evaluation setups. RSD and SinSR use at least ×5 less GPU memory and $\times 2.5 - 10$ fewer parameters depending on the T2I baseline, indicating substantially lower compute budgets. We also note the training efficiency of RSD compared to SinSR: RSD is a **simulation-free method**.

SinSR runs the ResShift teacher for training all 15 steps (Eq. 5–6 in (Wang et al., 2024b)), while RSD avoids it (Algorithm 1, Appendix B). In Appendix C, we discuss that SinSR empirically converges roughly 3 times slower than RSD. Although CTMSR is a distillation-free method, we show in Appendix F that its total training time is longer than the total training time of the ResShift teacher and its distillation with RSD. Despite strong perceptual quality, recent one-step T2I-based SR methods (AdcSR, PiSA-SR, TSD-SR) require substantially larger compute budgets than RSD. For example, compared to TSD-SR, RSD training is $\times 19$ faster, using $\times 2$ fewer GPUs, while TSD-SR uses $\times 13$ more parameters and $\times 8$ more GPU memory for inference. More discussion of performance-efficiency trade-off for RSD and other SOTA methods is provided in Appendix F.

### 4.3. Ablation study

**Multistep training.** We ablate multistep training (§3.3) across timestep configurations. As shown in Table 5, we compare various numbers of timesteps $N$ ranging from 1 to 15 with the maximum number matching that of ResShift; timesteps are evenly placed. We choose $N = 4$ for the best perception–distortion trade-off (Blau & Michaeli, 2018).

*Table 5.* Impact of multistep training of our RSD on RealSR. The best and second best results are highlighted in **bold** and underline.

| $N$ | PSNR↑ | LPIPS↓ | CLIPIQA↑ | MUSIQ↑ |
|---|---|---|---|---|
| 1 | 24.82 | 0.4052 | 0.7444 | 64.290 |
| 2 | 24.77 | 0.3772 | **0.7523** | 65.760 |
| 4 | 24.92 | 0.3552 | 0.7518 | 66.430 |
| 8 | 25.63 | 0.3199 | 0.7286 | **66.445** |
| 15 | **25.91** | **0.2940** | 0.6857 | 65.689 |

**Supervised losses.** Table 6 examines the impact of incorporating supervised losses, as discussed in §3.4. Our results show that adding these losses significantly enhances quality in PSNR and LPIPS while introducing compromised yet acceptable changes in no-reference metrics (CLIPIQA, MUSIQ). In all evaluations, we use full-size images with real-world degradations from RealSR.

*Table 6.* Effect of incorporating supervised losses on RealSR. The best and second best results are highlighted in **bold** and underline.

| Method | PSNR↑ | LPIPS↓ | CLIPIQA↑ | MUSIQ↑ |
|---|---|---|---|---|
| $\lambda_{1,2} = 0$ | 24.92 | 0.3552 | **0.7518** | 66.430 |
| $\lambda_1 \neq 0$ | **26.01** | **0.2708** | 0.7089 | 65.178 |
| $\lambda_2 \neq 0$ | 24.98 | 0.3064 | 0.6970 | **67.615** |
| **Ours** | 25.91 | 0.2726 | 0.7060 | 65.860 |

We provide additional ablation studies on the number of updates for the fake model per student update and the training stability of RSD in Appendix H.

## 5. Conclusion and Future Work

In this work, we propose RSD, a novel approach to distill the ResShift model into a student network with a single inference step. Our model is computationally efficient thanks to its ResShift framework but remains constrained by its teacher capacity issue, as validated in Appendix G. A more advanced teacher, such as a T2I-based model, could improve performance and enable the application of our method at higher resolutions. We discuss the limitations of RSD and failure cases in Appendix K.

## Impact Statement

Our proposed distillation method for the diffusion-based image SR model, RSD, presents potential societal impacts, both positive and negative. On the positive side, the practical effects of the techniques developed to improve the quality and efficiency of the real-world SR model range from enhancing medical imaging for diagnostic purposes and assisting in disaster response to improving remote sensing and autonomous driving performance. However, there are concerns regarding the generation of fake content. Our model is generative and may facilitate misleading image enhancement or manipulation. Potential risks depend on the deployment context and safeguards. We note that our model was trained using only one dataset, ImageNet (Deng et al., 2009), which is known to be standard in diffusion SR research (Yue et al., 2023; Wang et al., 2024b; You et al., 2025; Gushchin et al., 2025). Thus, we do not expect any high risk of misuse of the trained model as long as the training data do not contain unsafe images.

## Acknowledgements

The work was supported by the grant for research centers in the field of AI provided by the Ministry of Economic Development of the Russian Federation in accordance with the agreement 000000C313925P4F0002 and the agreement №139-10-2025-033.

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

# Appendix

We organize the structure of the appendix as follows:

1. Appendix A discusses the relation of RSD to relevant methods that involve training an auxiliary "fake" model - variational score distillation (VSD (Yin et al., 2024b; Wu et al., 2024a; Wang et al., 2023b)), score identity distillation (SiD (Zhou et al., 2024)), Flow Generator Matching (FGM (Huang et al., 2024)), and inverse bridge matching distillation (IBMD (Gushchin et al., 2025)). Appendix A.1 includes the derivation of the variational score distillation for ResShift and its comparison with our RSD loss $\mathcal{L}_\theta$. Appendix A.2 discusses the relation of RSD to SiD and FGM. Appendix A.3 discusses the relation of RSD to IBMD and their quantitative comparison. In Appendix A.4, we also discuss the generalization of the RSD method for other diffusion models.

2. Appendix B details the implementation with the pseudocode of RSD and the notation used in the paper. We also present the pseudocode for ResShift-VSD, introduced in Appendix A

3. Appendix C consists of experimental details for the implementation of RSD and baselines.

4. Appendix D describes the details of LLM usage in the paper.

5. Appendix E consists of full quantitative results, including additional baselines and results on full-size DRealSR (Wei et al., 2020), RealLR200 (Wu et al., 2024b), and RealLQ250 (Ai et al., 2024), which have not been shown in the main text due to space limitations.

6. Appendix F provides a comparison of performance and efficiency for RSD and state-of-the-art diffusion SR models: PiSA-SR (Sun et al., 2025), TSD-SR (Dong et al., 2025), AdcSR (Chen et al., 2025), CTMSR (You et al., 2025), InvSR (Yue et al., 2025), and CCSR (Sun et al., 2024).

7. Appendix G provides the quantitative comparison between RSD, SinSR, and ResShift when all these models are trained on HR cropped images with a resolution $512 \times 512$ from the LSDIR dataset (Li et al., 2023), which follows the training setup of OSEDiff (Wu et al., 2024a).

8. Appendix H provides an additional discussion of ablation studies on hyperparameter $K$ and the training stability of RSD.

9. Appendix I discusses the qualitative and quantitative comparison between RSD and AddSR (Xie et al., 2024).

10. Appendix J includes additional details of ResShift theory, which have not been shown in the main text due to space limitations.

11. Appendix K discusses the limitations of RSD and failure cases.

12. Appendix L discusses the motivation for the assumption of Equation (8), presents the proof of Proposition 3.1, and explains the computational issues of the original problem in Equation (9).

## A. Relation of RSD to VSD, SiD, FGM and IBMD

### A.1. Derivation of VSD objective for ResShift (ResShift-VSD) and comparative analysis with our objective

In this section, our objective is to:

1. Derive the VSD loss in the ResShift framework to compare it with our distillation loss under the same experimental conditions (see Table 1 and Table 2 );

2. Explain the main differences between our approach and the VSD loss.

To achieve this, we consider a generator $G_\theta$ with parameters $\theta$ and seek an update rule for them. We use a fake ResShift model to solve the following problem:

$$\arg\min_f \sum_{t=1}^T w_t \mathbb{E}_{p_\theta(\widehat{x}_0, y_0, x_t)} \big[ \|f(x_t, y_0, t) - \widehat{x}_0\|_2^2 \big], \tag{14}$$

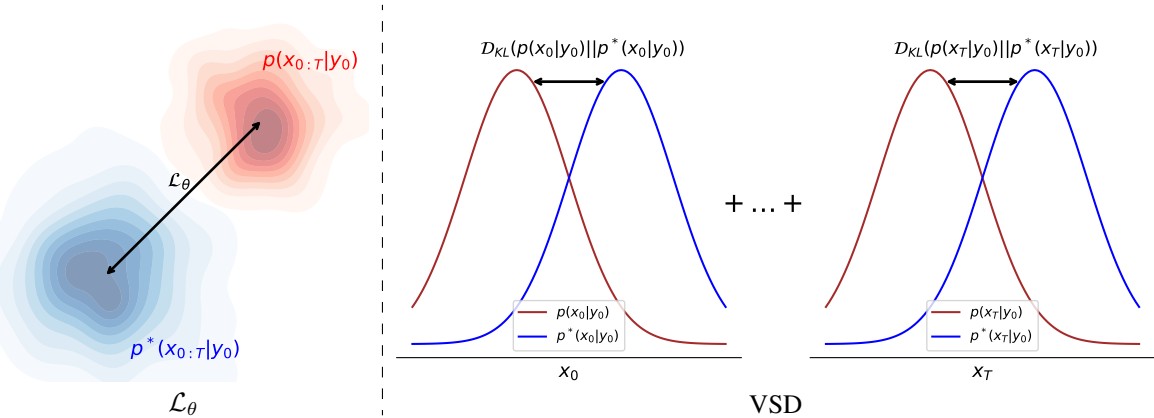

*Figure 4.* Illustration of the distinct distribution alignment strategies employed by the RSD $\mathcal{L}_\theta$ (**Ours**) and VSD loss functions. We denote by $p^*(x_{0:T}|y_0)$ reverse process of teacher ResShift model and by $p(x_{0:T}|y_0)$ reverse process of ResShift trained on generator $G_\theta$ data. The $\mathcal{L}_\theta$ loss enforces alignment of the joint distributions $p^*(x_{0:T}|y_0)$ and $p(x_{0:T}|y_0)$ across **all timesteps**, whereas the VSD loss aligns the marginal distributions at **each timestep** $t$ **simultaneously** between distributions of teacher ResShift and ResShift trained on generator $G_\theta$ data. For formal derivations, see Equations (24) and (18).

Since it is the optimization with MSE function, the solution is given by the conditional expectation:

$$f_{G_\theta}(x_t, y_0, t) = \mathbb{E}_{p_\theta(\widehat{x}_0|y_0, x_t)}[\widehat{x}_0] \tag{15}$$

**Notation.** Further we will use the following notation:

- $f^*$ – teacher ResShift.

- $x_{t_1:t_2} \overset{\text{def}}{=} (x_{t_1}, x_{t_1+1}, \ldots, x_{t_2})$ and $dx_{t_1:t_2} \overset{\text{def}}{=} \prod_{t=t_1}^{t_2} dx_t$ for any integer $t_1 < t_2$.

- The joint distribution across all timesteps is defined as follows:

$$p(x_{0:T}|y_0) \overset{\text{def}}{=} p(x_T|y_0) \prod_{t=1}^{T} p(x_{t-1}|x_t, y_0). \tag{16}$$

The transition probabilities are determined using Equation (2) and Equation (6):

$$p(x_{t-1}|x_t, y_0) = \mathcal{N}\left(x_{t-1} | \frac{\eta_{t-1}}{\eta_t} x_t + \frac{\alpha_t}{\eta_t} f_{G_\theta}(x_t, y_0, t), \kappa^2 \frac{\eta_{t-1}}{\eta_t} \alpha_t \mathbf{I}\right) \tag{17}$$

In the same way we define $p^*(x_{0:T}|y_0) \overset{\text{def}}{=} p^*(x_T|y_0) \prod_{t=1}^{T} p^*(x_{t-1}|x_t, y_0)$, where the transition probabilities are determined using $f^*$.

- $p^*(x_t|y_0) \overset{\text{def}}{=} \int p^*(x_{0:T}|y_0) dx_{0:t-1} dx_{t+1:T}$ and $p(x_t|y_0) \overset{\text{def}}{=} \int p(x_{0:T}|y_0) dx_{0:t-1} dx_{t+1:T}$ are marginal distributions.

**Derivation of the VSD loss for ResShift (ResShift-VSD).** Initially, the main objective of the VSD loss (Yin et al., 2024b; Wu et al., 2024a; Wang et al., 2023b) is:

$$\mathcal{L}_{\text{VSD}} = \mathbb{E}_{p(y_0)} \left[ \sum_{t=1}^{T} w_t \mathcal{D}_{\text{KL}}\Big(p(x_t|y_0) || p^*(x_t|y_0)\Big) \right] \tag{18}$$

We can get another expression for this loss using the reparametrization based on Equation (4):

$$\mathcal{L}_{\text{VSD}} = \sum_{t=1}^{T} w_t \mathbb{E}_{p(y_0)} \Big[ \mathcal{D}_{\text{KL}} \Big( p(x_t|y_0) || p^*(x_t|y_0) \Big) \Big] = \sum_{t=1}^{T} w_t \mathbb{E}_{p(y_0)p(x_t|y_0)} \log \frac{p(x_t|y_0)}{p^*(x_t|y_0)} =$$

$$\sum_{t=1}^{T} w_t \mathbb{E}_{p(y_0)} \mathop{\mathbb{E}}_{\substack{x_t=(1-\eta_t)\widehat{x}_0+\eta_t y_0+\kappa\sqrt{\eta_t}\epsilon' \\ \widehat{x}_0=G_\theta(y_0,\epsilon) \\ \epsilon',\epsilon\sim\mathcal{N}(0;\mathbf{I})}} \log \frac{p(x_t|y_0)}{p^*(x_t|y_0)} \tag{19}$$

Initially, this loss is intractable because it requires computing probability densities, which are not available in practice. However, taking the gradient with the chain rule facilitates its computation:

$$\nabla_\theta \mathcal{L}_{\text{VSD}} =$$

$$-\sum_{t=1}^{T} w_t \mathbb{E}_{p(y_0)} \mathop{\mathbb{E}}_{\substack{x_t=(1-\eta_t)\widehat{x}_0+\eta_t y_0+\kappa\sqrt{\eta_t}\epsilon' \\ \widehat{x}_0=G_\theta(y_0,\epsilon) \\ \epsilon',\epsilon\sim\mathcal{N}(0;\mathbf{I})}} \left[ (\nabla_{x_t} \log p^*(x_t|y_0) - \nabla_{x_t} \log p(x_t|y_0)) \frac{dx_t}{d\theta} \right] =$$

$$-\sum_{t=1}^{T} w_t \mathbb{E}_{p(y_0)} \mathop{\mathbb{E}}_{\substack{x_t=(1-\eta_t)\widehat{x}_0+\eta_t y_0+\kappa\sqrt{\eta_t}\epsilon' \\ \widehat{x}_0=G_\theta(y_0,\epsilon) \\ \epsilon',\epsilon\sim\mathcal{N}(0;\mathbf{I})}} \left[ (\nabla_{x_t} \log p^*(x_t|y_0) - \nabla_{x_t} \log p(x_t|y_0)) \frac{dx_t}{d\widehat{x}_0} \frac{d\widehat{x}_0}{d\theta} \right] =$$

$$-\sum_{t=1}^{T} w'_t \mathbb{E}_{p(y_0)} \mathop{\mathbb{E}}_{\substack{x_t=(1-\eta_t)\widehat{x}_0+\eta_t y_0+\kappa\sqrt{\eta_t}\epsilon' \\ \widehat{x}_0=G_\theta(y_0,\epsilon) \\ \epsilon',\epsilon\sim\mathcal{N}(0;\mathbf{I})}} \left[ (\nabla_{x_t} \log p^*(x_t|y_0) - \nabla_{x_t} \log p(x_t|y_0)) \frac{d\widehat{x}_0}{d\theta} \right], \tag{20}$$

where $w'_t \stackrel{\text{def}}{=} w_t \frac{dx_t}{d\widehat{x}_0} = w_t(1-\eta_t)$.

The expression $\nabla_{x_t} \log p(x_t|y_0)$ can be utilized as follows (Zheng et al., 2024):

$$\nabla_{x_t} \log p(x_t|y_0) = \frac{\nabla_{x_t} p(x_t|y_0)}{p(x_t|y_0)} = \frac{\nabla_{x_t} \int q(x_t|y_0,x_0)p(x_0|y_0)dx_0}{p(x_t|y_0)} =$$

$$\frac{\int p(x_0|y_0)\nabla_{x_t} q(x_t|y_0,x_0)dx_0}{p(x_t|y_0)} = \frac{\int p(x_0|y_0)q(x_t|y_0,x_0)\nabla_{x_t} \log q(x_t|y_0,x_0)dx_0}{p(x_t|y_0)} =$$

$$\int \frac{p(x_0|y_0)q(x_t|y_0,x_0)}{p(x_t|y_0)} \nabla_{x_t} \log q(x_t|y_0,x_0)dx_0 = \int p(x_0|x_t,y_0)\nabla_{x_t} \log q(x_t|y_0,x_0)dx_0$$

$$= \mathop{\mathbb{E}}_{p(x_0|x_t,y_0)} \Big[ \nabla_{x_t} \log q(x_t|y_0,x_0) \Big] \tag{21}$$

Since $q(x_t|y_0,x_0) = \mathcal{N}(x_t|x_0+\eta_t e_0, \kappa^2\eta_t\mathbf{I})$ (See Equation (4)), we get:

$$\nabla_{x_t} \log p(x_t|y_0) = -\mathop{\mathbb{E}}_{p(x_0|x_t,t,y_0)} \left[ \frac{x_t - \eta_t y_0 - (1-\eta_t)x_0}{\kappa^2\eta_t} \right], \tag{22}$$

which leads to:

$$\nabla_\theta \mathcal{L}_{\text{VSD}} = -\sum_{t=1}^{T} w''_t \mathbb{E}_{p(y_0)} \mathop{\mathbb{E}}_{\substack{x_t=(1-\eta_t)\widehat{x}_0+\eta_t y_0+\kappa\sqrt{\eta_t}\epsilon' \\ \widehat{x}_0=G_\theta(y_0,\epsilon) \\ \epsilon',\epsilon\sim\mathcal{N}(0;\mathbf{I})}} \left[ (f^*(x_t,y_0,t) - f_{G_\theta}(x_t,y_0,t)) \frac{d\widehat{x}_0}{d\theta} \right] \tag{23}$$

where $w''_t \stackrel{\text{def}}{=} w'_t \frac{1-\eta_t}{\kappa^2\eta_t}$. As a result, this loss can be implemented to match the gradients with $\nabla_\theta \mathcal{L}_{\text{VSD}}$ (see Algorithm 2). We call this model ResShift-VSD.

**Reformulation of our $\mathcal{L}_\theta$ loss**. We can express our RSD loss function as follows:

$$\mathcal{L}_\theta = \mathbb{E}_{p(y_0)}\Big[\mathcal{D}_{\mathrm{KL}}\Big(p(x_{0:T}|y_0)\,\|\,p^*(x_{0:T}|y_0)\Big)\Big], \tag{24}$$

Recalling that the joint probability distribution can be factorized (see Equation (16)), the above loss can be decomposed as:

$$\mathcal{L}_\theta = \mathbb{E}_{p(y_0)}\Big[\mathcal{D}_{\mathrm{KL}}\Big(p(x_{0:T}|y_0)\,\|\,p^*(x_{0:T}|y_0)\Big)\Big] = \mathbb{E}_{p(y_0)}\Big[\underbrace{\mathcal{D}_{\mathrm{KL}}\Big(p(x_T|y_0)\,\|\,p^*(x_T|y_0)\Big)}_{=0 \text{ since } p(x_T|y_0)=p^*(x_T|y_0) \text{ from Equation (4)}}\Big] +$$

$$\mathbb{E}_{p(y_0)}\Big[\sum_{t=1}^T \mathbb{E}_{p(x_t|y_0)}\mathcal{D}_{\mathrm{KL}}\Big(p(x_{t-1}|x_t,y_0)\,\|\,p^*(x_{t-1}|x_t,y_0)\Big)\Big] \tag{25}$$

Using Equation (17), the KL divergence inside the expectation reduces to the KL divergence between Gaussian distributions, which can be computed in closed form. Consequently, we obtain the following:

$$\mathcal{L}_\theta = \sum_{t=1}^T \mathbb{E}_{p(y_0)}\mathbb{E}_{p(x_t|y_0)}\mathcal{D}_{\mathrm{KL}}\Big(p(x_{t-1}|x_t,y_0)\,\|\,p^*(x_{t-1}|x_t,y_0)\Big)$$

$$= \sum_{t=1}^T \mathbb{E}_{p(y_0)}\mathbb{E}_{p(x_t|y_0)}\underbrace{\frac{1}{2\kappa^2}\frac{\alpha_t}{\eta_t\eta_{t-1}}}_{\overset{\text{def}}{=}w_t}\|f_{G_\theta}(x_t,y_0,t) - f^*(x_t,y_0,t)\|_2^2$$

$$= \sum_{t=1}^T w_t\mathbb{E}_{p(y_0)}\mathbb{E}_{p(x_t|y_0)}\|f_{G_\theta}(x_t,y_0,t) - f^*(x_t,y_0,t)\|_2^2 \tag{26}$$

Since the distribution $p(x_t|y_0)$ is generally intractable, we instead use the tractable distribution $q(x_t|\widehat{x}_0,y_0)$, which is known to satisfy $p(x_t|y_0) = \mathbb{E}_{\widehat{x}_0\sim p_\theta(\widehat{x}_0|y_0)}q(x_t|\widehat{x}_0,y_0)$. Thus, we have:

$$\mathcal{L}_\theta = \sum_{t=1}^T w_t\mathbb{E}_{p(y_0)}\mathbb{E}_{p_\theta(\widehat{x}_0|y_0)}\mathbb{E}_{q(x_t|\widehat{x}_0,y_0)}\|f_{G_\theta}(x_t,y_0,t) - f^*(x_t,y_0,t)\|_2^2 \tag{27}$$

Noting that the integrand is independent of $\widehat{x}_0$ and we can use $p_\theta(\widehat{x}_0,y_0)$ instead of $p(y_0)$, since $\int p_\theta(\widehat{x}_0|y_0)d\widehat{x}_0 = 1$, and therefore we obtain the following:

$$\mathcal{L}_\theta = \sum_{t=1}^T w_t\mathbb{E}_{q(x_t|\widehat{x}_0,y_0)\,p_\theta(\widehat{x}_0,y_0)}\|f_{G_\theta}(x_t,y_0,t) - f^*(x_t,y_0,t)\|_2^2 \tag{28}$$

Finally, recognizing that the joint distribution $p_\theta(\widehat{x}_0,y_0,x_t)$ is defined as

$$p_\theta(\widehat{x}_0,y_0,x_t) \overset{\text{def}}{=} q(x_t|\widehat{x}_0,y_0)\,p_\theta(\widehat{x}_0,y_0),$$

we arrive at the final form of the RSD loss:

$$\mathcal{L}_\theta = \sum_{t=1}^T w_t\mathbb{E}_{p_\theta(\widehat{x}_0,y_0,x_t)}\|f_{G_\theta}(x_t,y_0,t) - f^*(x_t,y_0,t)\|_2^2 \tag{29}$$

This derivation demonstrates that the loss function in Equation (24) reconstructs the initial objective presented in Equation (9).

**Conceptual comparison of VSD and our RSD $\mathcal{L}_\theta$ losses.** The key difference between VSD and $\mathcal{L}_\theta$ losses lies in how they match distributions. For a clearer intuitive explanation, one can see the formulations of losses with $\mathcal{D}_{\mathrm{KL}}$ for VSD (Equation (18)) and $\mathcal{L}_\theta$ (Equation (24)). The VSD loss aligns the marginal distributions at each timestep $t$ between the teacher's and

fake's distributions. In contrast, the $\mathcal{L}_\theta$ loss matches the joint distribution in all timesteps. This difference is illustrated in Figure 4, where the $\mathcal{L}_\theta$ loss enforces joint distribution alignment, while the VSD loss aligns the marginal distributions separately and then sums them.

Unlike VSD, which aligns marginal distributions at each timestep separately, RSD captures temporal dependencies more effectively. This joint alignment is particularly beneficial for SR tasks, where maintaining consistency and accuracy across all image details and features is crucial for high-quality resolution. The loss of RSD, which considers the entire distribution across multiple timesteps, leads to more precise and stable SR performance, as we validated in Section 4.2.

**Computational analysis of VSD and $\mathcal{L}_\theta$ losses**. As was shown in Proposition 3.1, our loss can be evaluated via:

$$\mathcal{L}_\theta = -\min_\phi \left\{ \sum_{t=1}^{T} w_t \mathbb{E}_{p_\theta(\widehat{x}_0, x_t, y_0)} \left[ \|f_\phi(x_t, y_0, t)\|_2^2 - \|f^*(x_t, y_0, t)\|_2^2 + \right. \right.$$
$$\left. \left. 2\langle f^*(x_t, y_0, t) - f_\phi(x_t, y_0, t), \widehat{x}_0 \rangle \right] \right\} \tag{30}$$

Using Equation (15) we can rewrite it and make reparameterization:

$$\mathcal{L}_\theta = -\sum_{t=1}^{T} w_t \mathbb{E}_{p_\theta(\widehat{x}_0, x_t, y_0)} \left[ \|f_{G_\theta}(x_t, y_0, t)\|_2^2 - \|f^*(x_t, y_0, t)\|_2^2 + \right.$$
$$\left. 2\langle f^*(x_t, y_0, t) - f_{G_\theta}(x_t, y_0, t), \widehat{x}_0 \rangle \right] =$$
$$-\sum_{t=1}^{T} w_t \mathop{\mathbb{E}}_{\substack{x_t=(1-\eta_t)\widehat{x}_0+\eta_t y_0+\kappa\sqrt{\eta_t}\epsilon' \\ \widehat{x}_0 = G_\theta(y_0,\epsilon) \\ \epsilon',\epsilon \sim \mathcal{N}(0;\mathbf{I})}} \left[ \|f_{G_\theta}(x_t, y_0, t)\|_2^2 - \|f^*(x_t, y_0, t)\|_2^2 + \right.$$
$$\left. 2\langle f^*(x_t, y_0, t) - f_{G_\theta}(x_t, y_0, t), \widehat{x}_0 \rangle \right] \tag{31}$$

To compare it with the VSD loss, we can take the gradient of $\mathcal{L}_\theta$ loss and get:

$$\frac{d\mathcal{L}_\theta}{d\theta} = -\sum_{t=1}^{T} w_t \mathop{\mathbb{E}}_{\substack{x_t=(1-\eta_t)\widehat{x}_0+\eta_t y_0+\kappa\sqrt{\eta_t}\epsilon' \\ \widehat{x}_0 = G_\theta(y_0,\epsilon) \\ \epsilon',\epsilon \sim \mathcal{N}(0;\mathbf{I})}} \left[ \frac{d\|f_{G_\theta}(x_t, y_0, t)\|_2^2}{d\theta} - \frac{d\|f^*(x_t, y_0, t)\|_2^2}{d\theta} + \right.$$
$$2\langle \frac{df^*(x_t, y_0, t)}{d\theta} - \frac{df_{G_\theta}(x_t, y_0, t)}{d\theta}, \widehat{x}_0 \rangle + 2\langle f^*(x_t, y_0, t) - f_{G_\theta}(x_t, y_0, t), \frac{d\widehat{x}_0}{d\theta} \rangle \Big] =$$
$$-\sum_{t=1}^{T} w_t \mathop{\mathbb{E}}_{\substack{x_t=(1-\eta_t)\widehat{x}_0+\eta_t y_0+\kappa\sqrt{\eta_t}\epsilon' \\ \widehat{x}_0 = G_\theta(y_0,\epsilon) \\ \epsilon',\epsilon \sim \mathcal{N}(0;\mathbf{I})}} \left[ \frac{d\|f_{G_\theta}(x_t, y_0, t)\|_2^2}{d\theta} - \frac{d\|f^*(x_t, y_0, t)\|_2^2}{d\theta} + \right.$$
$$2\langle \frac{df^*(x_t, y_0, t)}{d\theta} - \frac{df_{G_\theta}(x_t, y_0, t)}{d\theta}, \widehat{x}_0 \rangle \Big]$$
$$\underbrace{-\sum_{t=1}^{T} w_t \mathop{\mathbb{E}}_{\substack{x_t=(1-\eta_t)\widehat{x}_0+\eta_t y_0+\kappa\sqrt{\eta_t}\epsilon' \\ \widehat{x}_0 = G_\theta(y_0,\epsilon) \\ \epsilon',\epsilon \sim \mathcal{N}(0;\mathbf{I})}} \left[ 2\langle f^*(x_t, y_0, t) - f_{G_\theta}(x_t, y_0, t), \frac{d\widehat{x}_0}{d\theta} \rangle \right]}_{=2\cdot\nabla_\theta \mathcal{L}_{\text{VSD}} \text{ up to weighting term } w_t \text{ (see Equation(23))}} \tag{32}$$

Consequently, the gradients of our RSD $\mathcal{L}_\theta$ loss function encompass those of VSD loss, scaled by a constant factor of 2 and modulated by the time-dependent weighting term $w_t$. These scaling factors do not affect the optimal solution of the loss. However, $\mathcal{L}_\theta$ additionally incorporates gradient contributions from both the teacher and fake models. To reduce $\mathcal{L}_\theta$ to the standard VSD formulation, the application of a stop-gradient operator is required to suppress the influence of these auxiliary gradient terms. For a detailed implementation, refer to Algorithm 2.

## A.2. Relation of RSD to SiD and FGM

The loss function $\mathcal{L}_\theta$ in Equation (10) can be reformulated as follows:

$$
\begin{aligned}
\mathcal{L}_\theta = \max_\phi \Big\{ & \sum_{t=1}^{T} w_t \mathbb{E}_{p_\theta(\widehat{x}_0, y_0, x_t)} \Big( \|f^*(x_t, y_0, t)\|_2^2 - \|f_\phi(x_t, y_0, t)\|_2^2 + \\
& 2\langle f_\phi(x_t, y_0, t) - f^*(x_t, y_0, t), \widehat{x}_0 \rangle \Big) \Big\} \\
= \max_\phi \Big\{ & \sum_{t=1}^{T} w_t \mathbb{E}_{p_\theta(\widehat{x}_0, y_0, x_t)} \Big( \|f^*(x_t, y_0, t)\|_2^2 - \|f_\phi(x_t, y_0, t)\|_2^2 + \\
& 2\langle f_\phi(x_t, y_0, t) - f^*(x_t, y_0, t), \widehat{x}_0 \rangle \textcolor{red}{\pm 2\langle f^*(x_t, y_0, t), f_\phi(x_t, y_0, t)\rangle \pm \|f_\phi(x_t, y_0, t)\|_2^2} \Big) \Big\} \\
= \max_\phi \Big\{ & \sum_{t=1}^{T} w_t \mathbb{E}_{p_\theta(\widehat{x}_0, y_0, x_t)} \Big( \|f^*(x_t, y_0, t) - f_\phi(x_t, y_0, t)\|_2^2 + \\
& 2\langle f^*(x_t, y_0, t) - f_\phi(x_t, y_0, t), f_\phi(x_t, y_0, t)\rangle + 2\langle f_\phi(x_t, y_0, t) - f^*(x_t, y_0, t), \widehat{x}_0 \rangle \Big) \Big\} \\
= \max_\phi \Big\{ & \sum_{t=1}^{T} w_t \mathbb{E}_{p_\theta(\widehat{x}_0, y_0, x_t)} \Big( \|f^*(x_t, y_0, t) - f_\phi(x_t, y_0, t)\|_2^2 + \\
& 2\langle f^*(x_t, y_0, t) - f_\phi(x_t, y_0, t), f_\phi(x_t, y_0, t) - \widehat{x}_0 \rangle \Big) \Big\}
\end{aligned}
\tag{33}
$$

One can see that our objective can be reformulated in a manner similar to the formulations used in SiD (Zhou et al., 2024, Equation 23 with $\alpha = 0.5$) and FGM (Huang et al., 2024, Equations 4.11-4.12) with up to time weighting $w_t$. However, in both SiD (where $\alpha = 1.0, 1.2$ were used in the experiments) and FGM, the authors either omitted the quadratic term or assigned it a negative coefficient in the image experiments due to numerical instabilities. In contrast to these approaches, we retain the complete original loss formulation as prescribed by theory, without discarding or modifying any of its components.

Furthermore, it is important to emphasize that SiD and FGM were primarily developed for image generation tasks, whereas our proposed RSD framework is specifically tailored for image restoration, with a focus on reconstructing high-resolution images from their low-resolution counterparts. To this end, we adopt a dedicated ResShift architecture that integrates both VAE and U-Net components, along with a diffusion process specifically designed for the super-resolution task. Additionally, we incorporate supervised loss terms tailored to the super-resolution objective (see Section 3.5). These task-specific design choices are in contrast to SiD and FGM, which lack such adaptations for image restoration scenarios.

## A.3. Relation of RSD to IBMD

In this section, we compare qualitatively and quantitatively the RSD and IBMD (Gushchin et al., 2025) methods for the Real-ISR problem. Our goal is to support the practical contribution of RSD superiority for this problem, which was claimed in Section 1.

**Conceptual comparison**. The loss function $\mathcal{L}_\theta$ in Equation (10) can be equivalently reformulated as follows:

$$
\begin{aligned}
\mathcal{L}_\theta = \max_\phi \Big\{ & \sum_{t=1}^{T} w_t \mathbb{E}_{p_\theta(\widehat{x}_0, y_0, x_t)} \Big( \|f^*(x_t, y_0, t)\|_2^2 - \|f_\phi(x_t, y_0, t)\|_2^2 + \\
& 2\langle f_\phi(x_t, y_0, t) - f^*(x_t, y_0, t), \widehat{x}_0 \rangle \Big) \Big\} \\
= \max_\phi \Big\{ & \sum_{t=1}^{T} w_t \mathbb{E}_{p_\theta(\widehat{x}_0, y_0, x_t)} \Big( \|f^*(x_t, y_0, t)\|_2^2 - \|f_\phi(x_t, y_0, t)\|_2^2 + \\
& 2\langle f_\phi(x_t, y_0, t), \widehat{x}_0 \rangle - 2\langle f^*(x_t, y_0, t), \widehat{x}_0 \rangle \textcolor{red}{\pm \|\widehat{x}_0\|_2^2} \Big) \Big\} \\
= \max_\phi \Big\{ & \sum_{t=1}^{T} w_t \mathbb{E}_{p_\theta(\widehat{x}_0, y_0, x_t)} \Big( \|f^*(x_t, y_0, t) - \widehat{x}_0\|_2^2 - \|f_\phi(x_t, y_0, t) - \widehat{x}_0\|_2^2 \Big) \Big\}
\end{aligned}
\tag{34}
$$

It should be noted that our RSD loss, denoted by $\mathcal{L}_\theta$, can be interpreted as a discrete variant of the inverse bridge matching distillation (IBMD) loss (Gushchin et al., 2025, Equation 10), originally proposed for conditional Bridge Matching models. From a theoretical perspective, one of our main contributions is the development of a discretized form of the IBMD-conditional loss, which can offer practical benefits for complex tasks such as the Real-ISR problem.

Although the IBMD framework has been applied to a broad range of problems, including image restoration, its experimental setup relied on relatively simplistic degradation processes, such as bicubic and pool. In contrast, we have tailored our objective specifically for image restoration by integrating it into the ResShift paradigm, incorporating additional supervised losses explicitly designed for real-world super-resolution (see Section 3.5), and utilizing a more challenging and realistic degradation model based on Real-ESRGAN. We also note that ResShift and RSD use VAE and a diffusion process in the latent space, while IBMD operates in the pixel space, which makes RSD more efficient and scalable for handling images of varying resolutions due to the reduced computational complexity and memory requirements in the latent domain. Another relevant difference between IBMD and ResShift implementations for the SR problem is the larger architecture of IBMD. IBMD for super-resolution uses the ADM architecture (Dhariwal & Nichol, 2021) following the I2SB (Liu et al., 2023b) with 552M parameters, while RSD follows the ResShift architecture with 174M parameters.

**Quantitative comparison**. Thus, in addition to our theoretical contribution, we extend the implementation of the IBMD loss to more severe and practically relevant degradation settings. These task-specific modifications differentiate our approach from the original IBMD formulation, which does not account for such adaptations in the context of real-world image restoration scenarios. To support this claim quantitatively and qualitatively, we conducted the following numerical experiments to show that both the teacher model of I2SB and the distilled student model of IBMD do not provide sufficient perceptual quality in Real-ISR problems compared to ResShift and RSD, respectively.

**Step 1: training the I2SB teacher using Real-ESRGAN degradations**. For a fair comparison with ResShift, we trained an I2SB model on ImageNet using Real-ESRGAN degradations following the training setup of ResShift and RSD detailed in Section 4.1. The model was trained using the same hyperparameters as the original I2SB model trained on bicubic degradations with 4000 iterations. We used the official I2SB implementation published in the respective GitHub repository, which is provided by the I2SB authors:



https://github.com/NVlabs/I2SB



**Step 2: distillation of I2SB with IBMD**. We used the official IBMD implementation published in the respective GitHub repository, which is provided by the IBMD authors:



https://github.com/ngushchin/IBMD



We adapt the provided implementation with the replacement of Real-ESRGAN degradations instead of original bicubic degradations and use the same hyperparameters, which were used for the training of IBMD model for bicubic degradations (first line in Table 7 of IBMD). We distill the trained I2SB teacher with Real-ESRGAN degradations using IBMD method into a one-step student model with 1500 gradient updates for the student model, which we found enough for the convergence on the ImageNet-Test dataset.

For a fair comparison with ResShift and RSD, we evaluated the trained teacher I2SB with NFE = 15 and 1 and the trained IBMD student with NFE = 1, which follows the inference NFE of ResShift and RSD, respectively. We report on the results of their evaluation on the ImageNet-Test dataset, which follows the RSD evaluation setup reported in Table 2, and compare them with ResShift, RSD with supervised losses (RSD (**Ours**)), and RSD with only distillation loss (RSD (**Ours**, distill only)) in Table 7. We also extend the complexity comparison in our Table 4 of RSD with IBMD by providing the training time of both methods in Table 8.

**Comparison between teachers, I2SB and ResShift**. ResShift achieves better results on the ImageNet dataset with complex Real-ESRGAN degradations compared to I2SB **in all evaluation metrics** (PSNR, SSIM, LPIPS, MUSIQ, CLIPIQA) with a great improvement in perceptual metrics (LPIPS, MUSIQ, CLIPIQA). Our results in Table 7 are consistent with the analysis for comparison between ResShift and I2SB on simpler bicubic degradations, which is given in Appendix B.2 of ResShift. The same conclusion is quantitatively validated in Table 5 and visually supported in Figure 8 of ResShift, respectively. The results of Table 5 in ResShift also show a significant improvement in terms of perceptual quality for ResShift compared to I2SB for bicubic degradations with the same NFE = 15. We explain these results by the specific design of ResShift, which applies the diffusion process in discrete time in the latent space of VAE and uses the non-uniform geometric noise schedule (Section 2 in ResShift).

**Comparison between students, IBMD and RSD**. For a fair comparison, we compare our RSD model without supervised losses and the IBMD model, because IBMD originally is a data-free distillation method (see contribution 3 in IBMD). The results show that RSD is better in **all evaluation metrics even without supervised losses** (PSNR, SSIM, LPIPS, MUSIQ, CLIPIQA) with a significant improvement in perceptual metrics (LPIPS, MUSIQ, CLIPIQA). The addition of supervised losses in RSD even more increases the margin between all evaluation metrics. In practice, we also found that the IBMD model requires a significant computational budget, which is in line with the complexity reported in Table 9 of IBMD. We trained the IBMD model on 8 A100 for 23 hours, which is $> 4$ times more than the training time of RSD. IBMD also has $> 3$ times bigger the number of parameters and $> 8$ times bigger the required GPU memory for inference.

**Summary**. For a quantitative comparison, the RSD model without supervised losses outperforms the IBMD model in all evaluation metrics. For a computational comparison, the RSD model has a much faster distillation training time, requires fewer parameters, GPU memory, and has a faster generation time during inference compared to IBMD. Thus, task-specific features of RSD used for Real-ISR problems are essential for high perceptual quality compared to the diffusion distillation method of IBMD, which is developed for general image-to-image translation problems. It supports our claim in practical contributions in Section 1. We also visually observed that HR predictions for both I2SB and its IBMD distillation models struggle with severe blur artifacts, which explains low perceptual metrics (LPIPS, CLIPIQA, MUSIQ) and support higher fidelity metrics of PSNR and SSIM for I2SB. Due to the limitations of the file size of the submission, we do not provide visual results for Table 7 since Figure 8 of ResShift already shows the superiority of ResShift compared to I2SB for the SR problem.

*Table 7.* Comparison on ImageNet-Test between I2SB (Liu et al., 2023b), ResShift (Yue et al., 2023), and their 1-step distillation versions, IBMD (Gushchin et al., 2025) and RSD, respectively. The best and second best results are highlighted in **bold** and underline.

| Methods | Distillation model | NFE | PSNR↑ | SSIM↑ | LPIPS↓ | CLIPIQA↑ | MUSIQ↑ |
|---|---|---|---|---|---|---|---|
| I2SB (Liu et al., 2023a) | No | 15 | 24.80 | 0.663 | 0.302 | 0.444 | 49.584 |
| I2SB (Liu et al., 2023a) | No | 1 | **25.52** | **0.690** | 0.412 | 0.405 | 34.439 |
| ResShift (Yue et al., 2023) | No | 15 | 25.01 | 0.677 | 0.231 | 0.592 | 53.660 |
| IBMD (Gushchin et al., 2025) | Yes | 1 | 23.91 | 0.619 | 0.284 | 0.505 | 54.667 |
| RSD (**Ours**, distill only) | Yes | 1 | 23.97 | 0.643 | 0.217 | 0.660 | 57.831 |
| RSD (**Ours**) | Yes | 1 | 24.31 | 0.657 | **0.193** | **0.681** | **58.947** |

*Table 8.* Training and inference complexity between RSD and IBMD (Gushchin et al., 2025). All methods are tested with an LR image of size $64 \times 64$ for SR factor $\times 4$, and the inference is done on an NVIDIA A100 GPU. The best values are highlighted in **bold**.

| Methods | RSD (**Ours**) | IBMD |
|---|---|---|
| Inference Step (NFE) | **1** | 1 |
| Inference Time (s) | **0.059** | 0.077 |
| # Total Param (M) | **174** | 553 |
| Maximum GPU memory (MB) | **539** | 4676 |
| Training time (hours / # GPU) | **5 / 4 A100** | 23 / 8 A100 |

### A.4. Generalization of RSD to other methods.

It should be noted that the proof of Proposition 3.1 (see Appendix L), as well as the formulation of our loss function, does not rely on a specific form of processes used in the ResShift model. The only difference from other approaches is how they sample the joint distribution $p_\theta(\widehat{x}_0, y_0, x_t)$ during the training procedure. Usually, $p_\theta(\widehat{x}_0, y_0, x_t)$ is written in the following way:

$$p_\theta(\widehat{x}_0, y_0, x_t) = q(x_t \mid \widehat{x}_0, y_0)p_\theta(\widehat{x}_0 \mid y_0)p(y_0), \qquad (35)$$

which show that the only thing that is different for each method is the used distribution $q$. Thus, to adopt our approach to other processes, one only needs to change the distribution $q$. For example, to train I2SB or LDM models using the RSD formulation, one can use their discrete formulations for both models (Liu et al., 2023a, Equation 11) in I2SB or (Rombach et al., 2022, Equation 4) in LDM and plug them into $\mathcal{L}_\theta$.

# B. Algorithms of RSD, ResShift-RSD and Used Notation

The pseudocode for our RSD training algorithm is presented in Algorithm 1.

---

**Algorithm 1:** Residual Shifting Distillation (RSD).

---

**Input:**
Training dataset $p_{\text{data}}(x_0, y_0)$;
Pretrained ResShift Teacher model $f^*$; frozen encoder and decoder of VAE: Enc, Dec;
Number of fake ResShift ($f_\phi$) training iterations $K$, $f_\phi^{\text{encoder}}$ - encoder part of fake ResShift model, $N$ - amount of evenly spaced
  timesteps for multistep training;

**Output:**
A trained generator $G_\theta$;

**func** SampleEverything()
  $\quad$ Sample $(x_0, y_0) \sim p_{\text{data}}(x_0, y_0)$
  $\quad$ $z_y \leftarrow \text{Enc}(\text{upsample}(y_0)); \quad z_0 \leftarrow \text{Enc}(x_0)$
  $\quad$ Sample $t_n \sim \mathcal{U}\{t_1, \ldots, t_N\},\ z_{t_n} \sim q(z_{t_n}|z_0, z_y),\ \epsilon \sim \mathcal{N}(0, \mathbf{I})$ // Eq.  (4)
  $\quad$ $\widehat{z}_0^{t_n} \leftarrow G_\theta(z_{t_n}, y_0, t_n, \epsilon)$
  $\quad$ Sample $t \sim \mathcal{U}\{1, \ldots, T\},\ z_t \sim q(z_t|\widehat{z}_0^{t_n}, z_y)$ // Eq.  (4)
  $\quad$ **return** $(x_0, y_0, z_0, z_y, t_n, z_{t_n}, \widehat{z}_0^{t_n}, t, z_t)$
// Initialize generator from pretrained model
// Initialize fake ResShift from pretrained model and GAN discriminator head randomly
$G_\theta \leftarrow \text{copyWeightsAndUnfreeze}(f^*)$;
$f_\phi \leftarrow \text{copyWeightsAndUnfreezeAndAddNoiseChannels}(f^*)$ // See Appendix C
$D_\psi \leftarrow \text{randomInitOfDiscriminatorHead}()$

**while** *train* **do**
  $\quad$ // Train fake ResShift model
  $\quad$ **for** $k \leftarrow 1$ **to** $K$ **do**
  $\quad\quad$ $(x_0, y_0, z_0, z_y, t_n, z_{t_n}, \widehat{z}_0^{t_n}, t, z_t) \leftarrow \text{SampleEverything}()$ // Generate training data
  $\quad\quad$ $\mathcal{L}_{\text{fake}} \leftarrow w_t \|f_\phi(z_t, y_0, t) - \widehat{z}_0^{t_n}\|_2^2$ // Eq.  (7)
  $\quad\quad$ $\mathcal{L}_{\text{GAN}} \leftarrow \text{calcGANLossD}(D_\psi(f_\phi^{\text{encoder}}(\widehat{z}_0^{t_n}, y_0, 0)), D_\psi(f_\phi^{\text{encoder}}(z_0, y_0, 0)))$ // Eq.  (12)
  $\quad\quad$ $\mathcal{L}_\phi^{\text{total}} \leftarrow \mathcal{L}_{\text{fake}} + \lambda_2 \mathcal{L}_{\text{GAN}}$ // Eq.  (13)
  $\quad\quad$ Update $\phi$ by using $\frac{\partial \mathcal{L}_\phi^{\text{total}}}{\partial \phi}$
  $\quad\quad$ Update $\psi$ by using $\frac{\partial \mathcal{L}_{\text{GAN}}}{\partial \psi}$
  $\quad$ **end for**
  $\quad$ // Train generator model
  $\quad$ $(x_0, y_0, z_0, z_y, t_n, z_{t_n}, \widehat{z}_0^{t_n}, t, z_t) \leftarrow \text{SampleEverything}()$ // Generate training data

  $\quad$ $\mathcal{L}_\theta \leftarrow \text{calcThetaLoss}(f^*(z_t, y_0, t), f_\phi(z_t, y_0, t), \widehat{z}_0^{t_n})$ // Compute $\mathcal{L}_\theta$ loss with Eq.  (10)

  $\quad$ Sample $z_T \sim \mathcal{N}(z_T|z_y, \kappa^2\mathbf{I}); \quad \widehat{z}_0 \leftarrow G_\theta(z_T, y_0, T, \epsilon)$ // Eq.  (4)
  $\quad$ $\mathcal{L}_{\text{LPIPS}} \leftarrow \text{LPIPS}(x_0, \text{Dec}(\widehat{z}_0))$ // Compute $\mathcal{L}_{\text{LPIPS}}$ loss

  $\quad$ // Compute generator $\mathcal{L}_{\text{GAN}}$ loss
  $\quad$ $\mathcal{L}_{\text{GAN}} \leftarrow \text{calcGANLossG}(D_\psi(f_\phi^{\text{encoder}}(\widehat{z}_0^{t_n}, y_0, 0)))$ // Eq.  (12)

  $\quad$ $\mathcal{L}_\theta^{\text{total}} \leftarrow \mathcal{L}_\theta + \lambda_1 \mathcal{L}_{\text{LPIPS}} + \lambda_2 \mathcal{L}_{\text{GAN}}$ // Eq.  (13)
  $\quad$ Update $\theta$ by using $\frac{\partial \mathcal{L}_\theta^{\text{total}}}{\partial \theta}$
**end while**

---

The pseudocode for the baseline ResShift-VSD training algorithm is presented in Algorithm 2, while the foundational theoretical framework is detailed in Appendix A.1. To ensure a fair comparison with the distillation loss in OSEDiff (Wu et al., 2024a), specifically the VSD loss, under an identical experimental setup (i.e., ResShift), we adapted it to the ResShift framework using the same implementation details. In Table 9 we provide a detailed explanation of the notation used in Algorithms 1 and 2.

---

**Algorithm 2:** ResShift-VSD.

---

**Input:**
Training dataset $p_{\text{data}}(x_0, y_0)$;
Pretrained ResShift Teacher model $f^*$; frozen encoder and decoder of VAE: Enc, Dec;
Number of fake ResShift ($f_\phi$) training iterations $K$;

**Output:**
A trained generator $G_\theta$;

**func** SampleEverything()
   Sample $(x_0, y_0) \sim p_{\text{data}}(x_0, y_0)$;
   $z_y \leftarrow \text{Enc}(\text{upsample}(y_0))$
   Sample $z_T \sim \mathcal{N}(z_y, \kappa^2 \eta_T \mathbf{I})$ // Eq.   (4)
   $\widehat{z}_0 \leftarrow G_\theta(z_T, y_0, T)$
   Sample $t \sim \mathcal{U}\{1, \ldots, T\}$, $z_t \sim q(z_t|\widehat{z}_0, z_y)$ // Eq.   (4)
   **return** $(y_0, t, z_t, \widehat{z}_0)$

```
// Initialize generator from pretrained model
// Initialize fake ResShift from pretrained model
```
$G_\theta \leftarrow \text{copyWeightsAndUnfreeze}(f^*)$;
$f_\phi \leftarrow \text{copyWeightsAndUnfreeze}(f^*)$;

**while** *train* **do**
   `// Train fake ResShift model`
   **for** $k \leftarrow 1$ **to** $K$ **do**
      $(y_0, t, z_t, \widehat{z}_0) \leftarrow \text{SampleEverything}()$ `// Generate training data`
      $\mathcal{L}_{\text{fake}} \leftarrow w_t \|f_\phi(z_t, y_0, t) - \widehat{z}_0\|_2^2$ // Eq.   (7)
      Update $\phi$ by using $\frac{\partial \mathcal{L}_{\text{fake}}}{\partial \phi}$
   **end for**

   `// Train generator model`
   $(y_0, t, z_t, \widehat{z}_0) \leftarrow \text{SampleEverything}()$ `// Generate training data`
   $\mathcal{L}_\theta \leftarrow \text{calcThetaLoss}(\text{stopgrad}(f^*(z_t, y_0, t)), \text{stopgrad}(f_\phi(z_t, y_0, t)), \widehat{z}_0)$ // Eq.   (10)
   Update $\theta$ by using $\frac{\partial \mathcal{L}_\theta}{\partial \theta}$
**end while**

---

*Table 9.* Notation used in our paper. Pixel-space refers to the image domain, while latent-space refers to the internal representation domain.

| Symbol | Description | Space |
|:---:|:---|:---:|
| $x_0$ | Original high-resolution image | Pixel-space |
| $\hat{x}_0$ | Reconstructed high-resolution image from $\hat{z}_0$ | Pixel-space |
| $y_0$ | Low-resolution input image | Pixel-space |
| $z_0$ | Latent representation of $x_0$ | Latent-space |
| $z_y$ | Latent representation of $y_0$ | Latent-space |
| $z_{t_n}$ | Noised latent sampled from $z_0$ | Latent-space |
| $z_T$ | Noised latent sampled from $\mathcal{N}(z_T|z_y, \kappa^2 I)$ | Latent-space |
| $\hat{z}_0$ | Denoised latent output of generator $G_\theta(z_T, y_0, T, \epsilon)$ | Latent-space |
| $z_t$ | Noised latent sampled from $q(z_t|\hat{z}_0^{t_n}, z_y)$ | Latent-space |
| $\hat{z}_0^{t_n}$ | Generator output from $G_\theta(z_{t_n}, y_0, t_n, \epsilon)$ | Latent-space |
| $f^*(z_t, y_0, t)$ | Frozen teacher network output | Latent-space |
| $f_\phi(z_t, y_0, t)$ | Student network output (trained) | Latent-space |
| $\epsilon$ | Noise variable sampled from $\mathcal{N}(0, I)$ | Latent-space |

# C. Experimental Details

**Noise condition**. By default, fake ResShift and generator models are initialized with teacher weights. Furthermore, for noise conditioning, as described in §3.2, we implement an additional convolutional channel to expand the generator's first convolutional layer to accept noise as an additional input. The noise is concatenated with the encoded low-resolution image and is processed by a separate zero-initialized convolutional layer.

**Training hyperparameters**. We use the same hyperparameters as SinSR for training, including batch size, EMA rate, and optimizer type. To achieve smoother convergence, we replace the learning rate scheduler with a constant learning rate of $5 \times 10^{-5}$, which corresponds to the base learning rate of SinSR. Additionally, we adjust the AdamW (Loshchilov & Hutter, 2019) optimizer's $\beta$ parameters to $[0.9, 0.95]$ to further stabilize training. To ensure controlled adaptation between the generator and the fake ResShift models, we update the generator's weights once for every $K = 5$ updates of the fake model, following the strategy of DMD2 (Yin et al., 2024a). The influence of the hyperparameter $K$ on the training stability of RSD and its results is validated in Section 4.3. Furthermore, we adopt the loss normalization technique proposed in SiD (Zhou et al., 2024) to improve the stability of the training. In the final loss function (Equation 13), we set $\lambda_1 = 2$ and $\lambda_2 = 3 \cdot 10^{-3}$ following OSEDiff (Wu et al., 2024a) and DMD2, respectively.

**Training time**. The complete RSD training process, performed on 4 NVIDIA A100 GPUs, takes approximately 5 hours. During this time, the student model undergoes around 3000 gradient update iterations, while the fake model completes 15000 iterations. In practice, we found that SinSR (Wang et al., 2024b) requires around 60 hours on a single NVIDIA A100 GPU for 30000 iterations (2.57 days in Table 7 of SinSR (Wang et al., 2024b)) and SinSR converges roughly 3 times slower than RSD. We explain this difference by **simulation-free property** of RSD, which SinSR does not have. We recall that SinSR is a knowledge distillation method, which runs a full teacher ResShift model for all $T = 15$ steps during training according to Equations 5 and 6 in SinSR (Wang et al., 2024b):

$$x_{t-1} = k_t f^*(x_t, y_0, t) + m_t x_t + j_t y_0, \quad t \in \{T, T-1, \ldots, 2, 1\}, \tag{36}$$

$$F(x_T, y_0) = x_0, \quad x_T = y_0 + \kappa\sqrt{\eta_T}\epsilon, \quad \epsilon \sim \mathcal{N}(0, \mathbf{I}), \tag{37}$$

$$\mathcal{L}_{distill,SinSR} = L_{MSE}(f_\theta(x_T, y_0, T), F(x_T, y_0)), \tag{38}$$

where $f_\theta(x_T, y_0, T)$ is the student network in SinSR predicting the HR image in only one step, $F(x_T, y_0)$ represents the deterministic sampling with the ResShift teacher model using Equation (36), and $f^*(x_t, y_0, t)$ is the teacher model in ResShift. During training, RSD does not require full teacher simulation as SinSR does in Equation (36). However, in the RSD training, additional $K = 5$ updates for the fake model are required, while SinSR does not have any fake model. Thus, RSD achieves an acceleration of around $\times 3$ for training compared to SinSR.

**Codebase**. Our method is implemented based on the original SinSR repository (Wang et al., 2024b), which serves as the primary source of code for our experiments. We build our method on this framework to implement our training Algorithm 1, which is given in Appendix B.

**Teacher checkpoint**. Following SinSR repository (Wang et al., 2024b), we also distill the same ResShift checkpoint `resshift_realsrx4_s15_v1.pth`, which was trained with 300k iterations.

**Datasets and baselines**. Table 10 lists details on the datasets used for training and testing, including their sources and download links. Table 11 provides the associated licenses for the used datasets. Table 12 lists the models used for training and quality comparison and includes links to access them.

**Evaluation of metrics for SR models**. For calculating SR metrics, we use the PyTorch Toolbox for Image Quality Assessment and the `pyiqa` package (Chen & Mo, 2022). We also used the image quality assessment script provided in the OSEDiff GitHub repository.

# D. Statement on LLM Usage

The authors used the large language model (LLM) only to improve the writing and grammar of the text. All the results from the LLM were checked by the authors.

*Table 10.* The used datasets and their sources

| Name | URL | Citation |
|------|-----|----------|
| RealSR-V3 | GitHub Link | (Cai et al., 2019) |
| RealSet65 | GitHub Link | (Yue et al., 2023) |
| DRealSR | GitHub Link | (Wei et al., 2020) |
| ImageNet | Website Link | (Deng et al., 2009) |
| ImageNet-Test | Google Drive Link | (Yue et al., 2023) |
| DIV2K-Val-512 | Hugging Face Link | (Agustsson & Timofte, 2017; Wang et al., 2024a) |
| DRealSR-512 | Hugging Face Link | (Wang et al., 2024a; Wei et al., 2020) |
| RealSR-512 | Hugging Face Link | (Wang et al., 2024a; Cai et al., 2019) |
| RealLR200 | Google Drive Link | (Wu et al., 2024b) |
| RealLQ250 | Google Drive Link | (Ai et al., 2024) |

*Table 11.* The used datasets and their licenses

| Name | License |
|------|---------|
| RealSR-V3 | NTU S-Lab License 1.0 |
| DRealSR | Unknown |
| ImageNet | Custom (research, non-commercial) |
| ImageNet-Test | NTU S-Lab License |
| DIV2K-Val-512 | NTU S-Lab License |
| DRealSR-512 | NTU S-Lab License |
| RealSR-512 | NTU S-Lab License |
| RealLR200 | Apache 2.0 License |
| RealLQ250 | Apache 2.0 License |

*Table 12.* Baselines used for comparison. In each case, we used original code from GitHub repositories and model weights.

| Name | URL | Citation | License |
|------|-----|----------|---------|
| Real-ESRGAN | GitHub Link | (Wang et al., 2021) | BSD 3-Clause License |
| BSRGAN | GitHub Link | (Zhang et al., 2021) | Apache-2.0 license |
| SwinIR | GitHub Link | (Liang et al., 2021) | Apache-2.0 license |
| ResShift | GitHub Link | (Yue et al., 2023) | NTU S-Lab License 1.0 |
| SinSR | GitHub Link | (Wang et al., 2024b) | CC BY-NC-SA 4.0 |
| SUPIR | GitHub Link | (Yu et al., 2024) | SUPIR Software License |
| OSEDiff | GitHub Link | (Wu et al., 2024a) | Apache License 2.0 |
| AdcSR | GitHub Link | (Chen et al., 2025) | Apache License 2.0 |
| PiSA-SR | GitHub Link | (Sun et al., 2025) | Apache License 2.0 |
| TSD-SR | GitHub Link | (Dong et al., 2025) | Apache License 2.0 |
| CTMSR | GitHub Link | (You et al., 2025) | MIT License |
| CCSR | GitHub Link | (Sun et al., 2024) | Apache License 2.0 |
| InvSR | GitHub Link | (Yue et al., 2025) | NTU S-Lab License 1.0 |

# E. Additional Quantitative Results

We present an additional set of quantitative results, including more baselines and evaluations on full-size DRealSR (Wei et al., 2020), RealLR200 (Wu et al., 2024b), and RealLQ250 (Ai et al., 2024), which were not included in the main text due to space limitations:

- Table 13 provides results on full-size images from the DRealSR dataset (Wei et al., 2020).

- Table 14 provides non-reference results on full-size images from the RealLR200 (Wu et al., 2024b) and RealLQ250 (Ai et al., 2024) datasets.

- Table 15 presents an extended version of Table 1 on the RealSR (Cai et al., 2019) and RealSet65 (Yue et al., 2023) datasets, with additional baselines.

- Table 16 presents an extended version of Table 2 on the ImageNet-Test dataset (Yue et al., 2023) with additional baselines.

- Table 17 presents an extended version of Table 3 on crops from DIV2K (Agustsson & Timofte, 2017), RealSR, and DRealSR used in StableSR (Wang et al., 2024a) with additional baselines.

**Table 13**. We evaluated the following models for Table 13 and followed their official implementations listed in Table 12:

 1. **Diffusion-based SR models**. We ran pre-trained models of ResShift (Yue et al., 2023), SinSR (Wang et al., 2024b), OSEDiff (Wu et al., 2024a), and SUPIR (Yu et al., 2024) as representative members of diffusion-based SR models. We used the following checkpoints from the respective official repositories (Table 12): `resshift_realsrx4_s15_v1.pth`, `SinSR_v2.pth`, `osediff.pkl`, and `SUPIR-v0Q.ckpt`. Due to the high demands for GPU memory for the SUPIR model, we ran it with tiled VAE using the flag `--use_tile_vae`. For FluxSR (Li et al., 2025a), we used the results provided in their Google Drive Link and borrowed the results from their Tables 1 and 2.

 2. **State-of-the-art diffusion-based one-step SR models**. In addition to the ResShift, SinSR, OSEDiff, and SUPIR models, we also ran pre-trained, recent state-of-the-art one-step diffusion SR models, including TSD-SR (Dong et al., 2025), PiSA-SR (Sun et al., 2025), CTMSR (You et al., 2025), CCSR (Sun et al., 2024), and InvSR (Yue et al., 2025). We used the following checkpoints from the respective repositories listed in Table 12: 1) TSD-SR - LoRA weights from the folder `checkpoint/tsdsr-mse`, embedding weights from the folder `dataset/default`, and the teacher SD3-medium model from the Hugging Face Link; 2) PiSA-SR - `pisa_sr.pkl`; 3) CTMSR - `CTMSR.pth`; 4) InvSR - `noise_predictor_sd_turbo_v5_diftune.pth`; 5) CCSR - to follow the CCSR GitHub repository, we used ControlNet weights from the Google Drive Link, VAE weights from the Google Drive Link, and pre-trained ControlNet weights from the Google Drive Link, and Dino models from the Google Drive Link, respectively.

 3. **Non-diffusion SR models**. We ran pre-trained GAN-based SR models of Real-ESRGAN (Wang et al., 2021) and BSRGAN (Zhang et al., 2021) with the checkpoint names `RealESRGAN_x4plus.pth` and `BSRGAN.pth`, which are provided in the respective GitHub repositories listed in Table 12. We ran the pre-trained SwinIR model (Liang et al., 2021) with the checkpoint name `003_realSR_BSRGAN_DFOWMFC_s64w8_SwinIR-L_x4_GAN.pth` as the representative model from transformer-based SR models using the respective GitHub repository listed in Table 12.

We compute the same set of metrics as in Table 3 : PSNR, SSIM, LPIPS, CLIPIQA, MUSIQ, DISTS, NIQE, and MANIQA-PIPAL.

**Table 14**. We evaluated RSD and the following diffusion models on real-world benchmarks, namely RealLR200 (Wu et al., 2024b) and RealLQ250 (Ai et al., 2024), using no-reference perceptual metrics (CLIPIQA, MUSIQ, NIQE, MANIQA), which follow the evaluation protocol of SeeSR (Wu et al., 2024a, Table 2) and DreamClear (Ai et al., 2024, Table 1):

 1. **Diffusion-based SR models without T2I models**. We evaluated methods that were trained on the ImageNet data, namely ResShift, SinSR, CTMSR, and RSD.

 2. **T2I-based diffusion SR models**. We evaluated SUPIR, OSEDiff, AdcSR, PiSA-SR, TSD-SR, InvSR, and CCSR. For AdcSR (Chen et al., 2025), we used the weights from the checkpoint `net_params_200.pkl` from the respective GitHub repository.

*Table 13.* Quantitative results of models on full size images from DRealSR (Wei et al., 2020). The best and second best results are highlighted in **bold** and underline.

| Methods | Model class | NFE | PSNR↑ | SSIM↑ | LPIPS↓ | CLIPIQA↑ | MUSIQ↑ | DISTS↓ | NIQE↓ | MANIQA↑ |
|---|---|---|---|---|---|---|---|---|---|---|
| BSRGAN (Zhang et al., 2021) | GANs | 1 | 28.34 | 0.8206 | 0.2929 | 0.5704 | 35.500 | 0.1636 | 4.6811 | 0.4682 |
| Real-ESRGAN (Wang et al., 2021) | GANs | 1 | 27.91 | 0.8249 | 0.2818 | 0.5180 | 35.255 | 0.1464 | 4.7142 | 0.4756 |
| SwinIR (Liang et al., 2021) | Transformer | 1 | 28.31 | **0.8272** | **0.2741** | 0.5072 | 35.826 | **0.1387** | 4.6665 | 0.4617 |
| SUPIR (Yu et al., 2024) | Diffusion model, used T2I prior | 50 | 25.73 | 0.7224 | 0.3906 | 0.5862 | 36.089 | 0.1944 | 4.4685 | 0.5720 |
| OSEDiff (Wu et al., 2024a) | Diffusion model, used T2I prior | 1 | 26.67 | 0.7922 | 0.3123 | 0.7264 | 37.761 | 0.1617 | 4.1768 | 0.5883 |
| PiSA-SR (Sun et al., 2025) | Diffusion model, used T2I prior | 1 | 27.43 | 0.8119 | 0.2844 | 0.6878 | 35.060 | 0.1537 | 4.4783 | 0.5615 |
| TSD-SR (Dong et al., 2025) | Diffusion model, used T2I prior | 1 | 26.53 | 0.7637 | 0.3084 | **0.7517** | 37.395 | 0.1567 | **3.6624** | 0.5549 |
| InvSR (Yue et al., 2025) | Diffusion model, used T2I prior | 1 | 26.06 | 0.7455 | 0.3578 | 0.7485 | 33.878 | 0.1838 | 3.7279 | **0.5928** |
| CCSR (Sun et al., 2024) | Diffusion model, used T2I prior | 1 | 27.71 | 0.8022 | 0.3208 | 0.7104 | 35.716 | 0.1816 | 4.3081 | 0.5720 |
| FluxSR (Li et al., 2025a) | Diffusion model, used T2I prior | 1 | 25.92 | 0.7592 | 0.3618 | 0.7347 | 37.287 | 0.1928 | 4.6947 | 0.5566 |
| ResShift (Yue et al., 2023) | Diffusion model, no T2I prior | 15 | **28.76** | 0.7863 | 0.4310 | 0.5838 | 32.042 | 0.2314 | 6.6335 | 0.4297 |
| CTMSR (You et al., 2025) | Diffusion model, no T2I prior | 1 | 28.28 | 0.8017 | 0.3355 | 0.6821 | 33.206 | 0.1946 | 4.7795 | 0.4702 |
| SinSR (Wang et al., 2024b) | Diffusion model, no T2I prior | 1 | 27.32 | 0.7233 | 0.4452 | 0.7223 | 32.800 | 0.2368 | 5.5748 | 0.4757 |
| RSD (**Ours**) | Diffusion model, no T2I prior | 1 | 27.66 | 0.7864 | 0.3105 | 0.7398 | **38.340** | 0.1868 | 4.6098 | 0.5314 |

*Table 14.* Quantitative results of diffusion models on RealLR200 (Wu et al., 2024b) and RealLQ250 (Ai et al., 2024) datasets. The best and second best results are highlighted in **bold** and underline.

| Methods | T2I prior | NFE | RealLR200 CLIPIQA↑ | RealLR200 MUSIQ↑ | RealLR200 NIQE↓ | RealLR200 MANIQA↑ | RealLQ250 CLIPIQA↑ | RealLQ250 MUSIQ↑ | RealLQ250 NIQE↓ | RealLQ250 MANIQA↑ |
|---|---|---|---|---|---|---|---|---|---|---|
| SUPIR (Yu et al., 2024) | yes, > 450M params | 50 | 0.6188 | 64.79 | 4.1862 | 0.6120 | 0.5746 | 65.72 | **3.6607** | 0.5969 |
| OSEDiff (Wu et al., 2024a) | yes, > 450M params | 1 | 0.6728 | 69.45 | 4.0506 | 0.6153 | 0.6724 | 69.56 | 3.9682 | 0.5889 |
| AdcSR (Chen et al., 2025) | yes, > 450M params | 1 | 0.7047 | 70.35 | 3.8792 | 0.6174 | 0.6889 | 69.98 | 3.7181 | 0.5944 |
| PiSA-SR (Sun et al., 2025) | yes, > 450M params | 1 | 0.7039 | 70.90 | 3.9594 | 0.6419 | 0.7054 | 71.25 | 3.9162 | **0.6190** |
| TSD-SR (Dong et al., 2025) | yes, > 450M params | 1 | **0.7335** | **72.06** | 3.8352 | 0.6248 | 0.7368 | **73.22** | 3.6996 | 0.6037 |
| InvSR (Yue et al., 2025) | yes, > 450M params | 1 | 0.6774 | 68.15 | 4.0378 | **0.6461** | 0.6499 | 64.77 | 4.6505 | 0.5810 |
| CCSR (Sun et al., 2024) | yes, > 450M params | 1 | 0.6937 | 70.49 | 4.3108 | 0.6319 | 0.6850 | 70.80 | 4.4760 | 0.6021 |
| FluxSR (Li et al., 2025a) | yes, > 450M params | 1 | 0.7101 | 71.60 | 5.1905 | 0.6117 | **0.7374** | 72.65 | 5.3973 | 0.5901 |
| ResShift (Yue et al., 2023) | no, < 180M params | 15 | 0.6368 | 61.80 | 5.7016 | 0.5436 | 0.6348 | 61.99 | 5.7622 | 0.5364 |
| SinSR (Wang et al., 2024b) | no, < 180M params | 1 | 0.7089 | 64.90 | 5.3329 | 0.5561 | 0.7142 | 65.29 | 5.4630 | 0.5294 |
| CTMSR (You et al., 2025) | no, < 180M params | 1 | 0.6754 | 67.63 | 4.2943 | 0.5426 | 0.6701 | 68.07 | 4.5831 | 0.5130 |
| RSD (**Ours**) | no, < 180M params | 1 | 0.7151 | 68.66 | 4.7074 | 0.5949 | 0.7252 | 69.63 | 4.5531 | 0.5826 |

**Table 15** . We report an extended version of Table 1 with additional baselines used in the ResShift and SinSR papers:

1. **Non-diffusion SR models**. We evaluated Real-ESRGAN (Wang et al., 2021) and BSRGAN (Zhang et al., 2021) on RealSR and RealSet65. We also evaluated SwinIR on RealSR and RealSet65.

2. **State-of-the-art diffusion-based one-step SR models**. We also evaluated InvSR and CCSR using the same pre-trained models as for Table 13. For $D^3SR$ (Li et al., 2025b), we borrow the results from their Tables 1 and 2.

**Table 16** . We report an extended version of Table 2 with additional baselines used in the ResShift and SinSR papers:

1. **Diffusion-based SR models**. We borrow the results of Table 2 from SinSR for LDM-15 and LDM-30 (Rombach et al., 2022) and SinSR (Wang et al., 2024b). We borrow the results of Table 3 from (Yue et al., 2023) for ResShift. In addition to TSD-SR, PiSA-SR, CTMSR, and AdcSR, we also evaluated InvSR and CCSR using the same pre-trained models as for Table 13.

2. **Non-diffusion SR models**. We borrow the results of Table 2 from SinSR for ESRGAN (Wang et al., 2019), RealSR-JPEG (Ji et al., 2020), Real-ESRGAN (Wang et al., 2021), and BSRGAN (Zhang et al., 2021). We also borrow the results of Table 2 from SinSR for DASR (Liang et al., 2022b) and SwinIR (Liang et al., 2021).

*Table 15.* Extended quantitative results of models on two real-world datasets, RealSR (Cai et al., 2019) and RealSet65 (Yue et al., 2023). The best and second best results are highlighted in **bold** and underline.

| Methods | Model class | NFE | | | RealSR | | | RealSet65 | |
|---|---|---|---|---|---|---|---|---|---|
| | | | PSNR↑ | SSIM↑ | LPIPS↓ | CLIPIQA↑ | MUSIQ↑ | CLIPIQA↑ | MUSIQ↑ |
| BSRGAN (Zhang et al., 2021) | GANs | 1 | **26.51** | 0.775 | 0.269 | 0.5439 | 63.586 | 0.6163 | 65.582 |
| Real-ESRGAN (Wang et al., 2021) | | 1 | 25.85 | 0.773 | 0.273 | 0.4898 | 59.678 | 0.5995 | 63.220 |
| SwinIR (Liang et al., 2021) | Transformer | 1 | 26.43 | **0.786** | **0.251** | 0.4654 | 59.636 | 0.5782 | 63.822 |
| SUPIR (Yu et al., 2024) | Diffusion model, used T2I prior | 50 | 24.38 | 0.698 | 0.331 | 0.5449 | 63.676 | 0.6133 | 66.460 |
| OSEDiff (Wu et al., 2024a) | | 1 | 25.25 | 0.737 | 0.299 | 0.6772 | 67.602 | 0.6836 | 68.853 |
| AdcSR (Chen et al., 2025) | | 1 | 25.63 | 0.735 | 0.300 | 0.7033 | 67.550 | 0.7044 | 69.185 |
| PiSA-SR (Sun et al., 2025) | | 1 | 25.59 | 0.750 | 0.271 | 0.6678 | 67.993 | 0.7062 | 70.208 |
| TSD-SR (Dong et al., 2025) | | 1 | 24.88 | 0.723 | 0.281 | 0.7336 | **69.871** | 0.7263 | **70.958** |
| InvSR (Yue et al., 2025) | | 1 | 24.73 | 0.731 | 0.275 | 0.6798 | 66.403 | 0.6990 | 67.770 |
| CCSR (Sun et al., 2024) | | 1 | 25.99 | 0.752 | 0.287 | 0.6656 | 67.991 | 0.7150 | 70.731 |
| FluxSR (Li et al., 2025a) | | 1 | 24.83 | 0.718 | 0.320 | 0.6490 | 68.950 | - | 70.750 |
| D³SR (Li et al., 2025b) | | 1 | 24.11 | 0.715 | 0.296 | 0.5647 | 68.230 | 0.5481 | 70.250 |
| ResShift (Yue et al., 2023) | Diffusion model, no T2I prior | 15 | 26.49 | 0.754 | 0.360 | 0.5958 | 59.873 | 0.6537 | 61.330 |
| CTMSR (You et al., 2025) | | 1 | 26.18 | 0.765 | 0.294 | 0.6449 | 64.796 | 0.6893 | 67.173 |
| SinSR (distill only) (Wang et al., 2024b) | | 1 | 26.14 | 0.732 | 0.357 | 0.6119 | 57.118 | 0.6822 | 61.267 |
| SinSR (Wang et al., 2024b) | | 1 | 25.83 | 0.717 | 0.365 | 0.6887 | 61.582 | 0.7150 | 62.169 |
| ResShift-VSD (Appendix A) | | 1 | 23.96 | 0.616 | 0.466 | 0.7479 | 63.298 | **0.7606** | 66.701 |
| RSD (**Ours**, distill only) | | 1 | 24.92 | 0.696 | 0.355 | **0.7518** | 66.430 | 0.7534 | 68.383 |
| RSD (**Ours**) | | 1 | 25.91 | 0.754 | 0.273 | 0.7060 | 65.860 | 0.7267 | 69.172 |

*Table 16.* Extended quantitative results of models on ImageNet-Test (Yue et al., 2023). The best and second best results are highlighted in **bold** and underline.

| Methods | Model class | NFE | PSNR↑ | SSIM↑ | LPIPS↓ | CLIPIQA↑ | MUSIQ↑ |
|---|---|---|---|---|---|---|---|
| ESRGAN (Wang et al., 2019) | GANs | 1 | 20.67 | 0.448 | 0.485 | 0.451 | 43.615 |
| Real-ESRGAN (Wang et al., 2021) | | 1 | 24.04 | 0.665 | 0.254 | 0.523 | 52.538 |
| RealSR-JPEG (Ji et al., 2020) | | 1 | 23.11 | 0.591 | 0.326 | 0.537 | 46.981 |
| BSRGAN (Zhang et al., 2021) | | 1 | 24.42 | 0.659 | 0.259 | 0.581 | 54.697 |
| SwinIR (Liang et al., 2021) | Transformer | 1 | 23.99 | 0.667 | 0.238 | 0.564 | 53.790 |
| DASR (Liang et al., 2022b) | Mixture of experts | 1 | 24.75 | 0.675 | 0.250 | 0.536 | 48.337 |
| LDM (Rombach et al., 2022) | Diffusion model, used T2I prior | 30 | 24.49 | 0.651 | 0.248 | 0.572 | 50.895 |
| LDM (Rombach et al., 2022) | | 15 | 24.89 | 0.670 | 0.269 | 0.512 | 46.419 |
| SUPIR (Yu et al., 2024) | | 50 | 22.56 | 0.574 | 0.302 | **0.786** | 60.487 |
| OSEDiff (Wu et al., 2024a) | | 1 | 23.02 | 0.619 | 0.253 | 0.677 | 60.755 |
| AdcSR (Chen et al., 2025) | | 1 | 22.99 | 0.615 | 0.252 | 0.711 | 63.218 |
| PiSA-SR (Sun et al., 2025) | | 1 | 24.29 | 0.670 | 0.213 | 0.629 | 62.137 |
| TSD-SR (Dong et al., 2025) | | 1 | 23.58 | 0.645 | 0.197 | 0.673 | **65.299** |
| InvSR (Yue et al., 2025) | | 1 | 21.31 | 0.604 | 0.293 | 0.641 | 54.870 |
| CCSR (Sun et al., 2024) | | 1 | 24.79 | **0.677** | 0.238 | 0.602 | 61.789 |
| ResShift (Yue et al., 2023) | Diffusion model, no T2I prior | 15 | **25.01** | **0.677** | 0.231 | 0.592 | 53.660 |
| CTMSR (You et al., 2025) | | 1 | 24.73 | 0.666 | 0.197 | 0.691 | 60.142 |
| SinSR (distill only) (Wang et al., 2024b) | | 1 | 24.69 | 0.664 | 0.222 | 0.607 | 53.316 |
| SinSR (Wang et al., 2024b) | | 1 | 24.56 | 0.657 | 0.221 | 0.611 | 53.357 |
| ResShift-VSD (Appendix A) | | 1 | 23.69 | 0.624 | 0.230 | 0.665 | 58.630 |
| RSD (**Ours**, distill only) | | 1 | 23.97 | 0.643 | 0.217 | 0.660 | 57.831 |
| RSD (**Ours**) | | 1 | 24.31 | 0.657 | **0.193** | 0.681 | 58.947 |

**Table 17** . We report an extended version of Table 3 with additional baselines used in (Wu et al., 2024a, Table 1).

*Table 17.* Extended quantitative results of models on crops from StableSR (Wang et al., 2024a). The best and second best results are highlighted in **bold** and underline.

| Datasets | Methods | Model class | NFE | PSNR↑ | SSIM↑ | LPIPS↓ | DISTS↓ | NIQE↓ | MUSIQ↑ | MANIQA↑ | CLIPIQA↑ | FID↓ |
|---|---|---|---|---|---|---|---|---|---|---|---|---|
| DIV2K-Val | BSRGAN (Zhang et al., 2021) | GANs | 1 | 24.58 | 0.6269 | 0.3351 | 0.2275 | 4.7518 | 61.20 | 0.5071 | 0.5247 | 44.23 |
| | Real-ESRGAN (Wang et al., 2021) | | 1 | 24.29 | **0.6371** | 0.3112 | 0.2141 | 4.6786 | 61.06 | 0.5501 | 0.5277 | 37.64 |
| | LDL (Liang et al., 2022a) | | 1 | 23.83 | 0.6344 | 0.3256 | 0.2227 | 4.8554 | 60.04 | 0.5350 | 0.5180 | 42.29 |
| | FeMASR (Chen et al., 2022) | | 1 | 23.06 | 0.5887 | 0.3126 | 0.2057 | 4.7410 | 60.83 | 0.5074 | 0.5997 | 35.87 |
| | StableSR (Wang et al., 2024a) | Diffusion model, used T2I prior | 200 | 23.26 | 0.5726 | 0.3113 | 0.2048 | 4.7581 | 65.92 | 0.6192 | 0.6771 | **24.44** |
| | DiffBIR (Lin et al., 2025) | | 50 | 23.64 | 0.5647 | 0.3524 | 0.2128 | 4.7042 | 65.81 | 0.6210 | 0.6704 | 30.72 |
| | SeeSR (Wu et al., 2024b) | | 50 | 23.68 | 0.6043 | 0.3194 | 0.1968 | 4.8102 | 68.67 | 0.6240 | 0.6936 | 25.90 |
| | PASD (Yang et al., 2025) | | 20 | 23.14 | 0.5505 | 0.3571 | 0.2207 | 4.3617 | 68.95 | **0.6483** | 0.6788 | 29.20 |
| | SUPIR (Yu et al., 2024) | | 50 | 22.13 | 0.5280 | 0.3923 | 0.2314 | 5.6758 | 63.82 | 0.5933 | 0.7147 | 31.46 |
| | OSEDiff (Wu et al., 2024a) | | 1 | 23.72 | 0.6108 | 0.2941 | 0.1976 | 4.7097 | 67.97 | 0.6148 | 0.6683 | 26.32 |
| | AdcSR (Chen et al., 2025) | | 1 | 23.74 | 0.6017 | 0.2853 | 0.1899 | 4.3579 | 68.00 | 0.6073 | 0.6764 | 25.52 |
| | PiSA-SR (Sun et al., 2025) | | 1 | 23.87 | 0.6058 | 0.2823 | 0.1934 | 4.5565 | 69.68 | 0.6375 | 0.6928 | 25.09 |
| | TSD-SR (Dong et al., 2025) | | 1 | 23.02 | 0.5808 | **0.2673** | 0.1821 | 4.3244 | **71.69** | 0.6192 | **0.7416** | 29.16 |
| | InvSR (Yue et al., 2025) | | 1 | 23.10 | 0.5985 | 0.3045 | 0.1985 | 4.7056 | 68.43 | 0.6385 | 0.7117 | 28.45 |
| | CCSR (Sun et al., 2024) | | 1 | 24.30 | 0.6283 | 0.2979 | 0.2020 | 5.3367 | 69.52 | 0.6145 | 0.6752 | 30.86 |
| | D³SR (Li et al., 2025b) | | 1 | 22.05 | 0.6031 | 0.3556 | **0.1500** | **3.2950** | 68.51 | 0.5795 | 0.5370 | - |
| | ResShift (Yue et al., 2023) | Diffusion model, no T2I prior | 15 | 24.65 | 0.6181 | 0.3349 | 0.2213 | 6.8212 | 61.09 | 0.5454 | 0.6071 | 36.11 |
| | SinSR (Wang et al., 2024b) | | 1 | 24.41 | 0.6018 | 0.3240 | 0.2066 | 6.0159 | 62.82 | 0.5386 | 0.6471 | 35.57 |
| | CTMSR (You et al., 2025) | | 1 | **24.88** | 0.6265 | 0.3026 | 0.2040 | 5.1146 | 65.62 | 0.5165 | 0.6601 | 34.15 |
| | RSD (**Ours**) | | 1 | 23.91 | 0.6042 | 0.2857 | 0.1940 | 5.1987 | 68.05 | 0.5937 | 0.6967 | 34.84 |
| DRealSR | BSRGAN (Zhang et al., 2021) | GANs | 1 | **28.75** | 0.8031 | 0.2883 | 0.2142 | 6.5192 | 57.14 | 0.4878 | 0.4915 | 155.63 |
| | Real-ESRGAN (Wang et al., 2021) | | 1 | 28.64 | 0.8053 | 0.2847 | **0.2089** | 6.6928 | 54.18 | 0.4907 | 0.4422 | 147.62 |
| | LDL (Liang et al., 2022a) | | 1 | 28.21 | **0.8126** | **0.2815** | 0.2132 | 7.1298 | 53.85 | 0.4914 | 0.4310 | 155.53 |
| | FeMASR (Chen et al., 2022) | | 1 | 26.90 | 0.7572 | 0.3169 | 0.2235 | 5.9073 | 53.74 | 0.4420 | 0.5464 | 157.78 |
| | StableSR (Wang et al., 2024a) | Diffusion model, used T2I prior | 200 | 28.03 | 0.7536 | 0.3284 | 0.2269 | 6.5239 | 58.51 | 0.5601 | 0.6356 | 148.98 |
| | DiffBIR (Lin et al., 2025) | | 50 | 26.71 | 0.6571 | 0.4557 | 0.2748 | 6.3124 | 61.07 | 0.5930 | 0.6395 | 166.79 |
| | SeeSR (Wu et al., 2024b) | | 50 | 28.17 | 0.7691 | 0.3189 | 0.2315 | 6.3967 | 64.93 | 0.6042 | 0.6804 | 147.39 |
| | PASD (Yang et al., 2025) | | 20 | 27.36 | 0.7073 | 0.3760 | 0.2531 | **5.5474** | 64.87 | 0.6169 | 0.6808 | 156.13 |
| | SUPIR (Yu et al., 2024) | | 50 | 24.93 | 0.6360 | 0.4263 | 0.2823 | 7.4336 | 59.39 | 0.5537 | 0.6799 | 164.86 |
| | OSEDiff (Wu et al., 2024a) | | 1 | 27.92 | 0.7835 | 0.2968 | 0.2165 | 6.4902 | 64.65 | 0.5899 | 0.6963 | 135.30 |
| | AdcSR (Chen et al., 2025) | | 1 | 28.10 | 0.7726 | 0.3046 | 0.2200 | 6.4467 | 66.27 | 0.5916 | 0.7049 | 134.05 |
| | PiSA-SR (Sun et al., 2025) | | 1 | 28.32 | 0.7804 | 0.2960 | 0.2169 | 6.1766 | 66.11 | 0.6161 | 0.6968 | **130.61** |
| | TSD-SR (Dong et al., 2025) | | 1 | 27.77 | 0.7559 | 0.2967 | 0.2136 | 5.9131 | **66.62** | 0.5874 | **0.7343** | 134.98 |
| | InvSR (Yue et al., 2025) | | 1 | 25.79 | 0.7176 | 0.3471 | 0.2381 | 5.8627 | 64.92 | **0.6212** | 0.7185 | 166.51 |
| | CCSR (Sun et al., 2024) | | 1 | 28.24 | 0.7818 | 0.3201 | 0.2327 | 6.7901 | 66.28 | 0.6056 | 0.6632 | 157.23 |
| | ResShift (Yue et al., 2023) | Diffusion model, no T2I prior | 15 | 28.46 | 0.7673 | 0.4006 | 0.2656 | 8.1249 | 50.60 | 0.4586 | 0.5342 | 172.26 |
| | SinSR (Wang et al., 2024b) | | 1 | 28.36 | 0.7515 | 0.3665 | 0.2485 | 6.9907 | 55.33 | 0.4884 | 0.6383 | 170.57 |
| | CTMSR (You et al., 2025) | | 1 | 28.65 | 0.7834 | 0.3238 | 0.2358 | 6.1828 | 59.78 | 0.4861 | 0.6497 | 163.63 |
| | RSD (**Ours**) | | 1 | 27.40 | 0.7559 | 0.3042 | 0.2343 | 6.2577 | 62.03 | 0.5625 | 0.7019 | 167.47 |
| RealSR | BSRGAN (Zhang et al., 2021) | GANs | 1 | **26.39** | 0.7654 | **0.2670** | 0.2121 | 5.6567 | 63.21 | 0.5399 | 0.5001 | 141.28 |
| | Real-ESRGAN (Wang et al., 2021) | | 1 | 25.69 | 0.7616 | 0.2727 | 0.2063 | 5.8295 | 60.18 | 0.5487 | 0.4449 | 135.18 |
| | LDL (Liang et al., 2022a) | | 1 | 25.28 | 0.7567 | 0.2766 | 0.2121 | 6.0024 | 60.82 | 0.5485 | 0.4477 | 142.71 |
| | FeMASR (Chen et al., 2022) | | 1 | 25.07 | 0.7358 | 0.2942 | 0.2288 | 5.7885 | 58.95 | 0.4865 | 0.5270 | 141.05 |
| | StableSR (Wang et al., 2024a) | Diffusion model, used T2I prior | 200 | 24.70 | 0.7085 | 0.3018 | 0.2288 | 5.9122 | 65.78 | 0.6221 | 0.6178 | 128.51 |
| | DiffBIR (Lin et al., 2025) | | 50 | 24.75 | 0.6567 | 0.3636 | 0.2312 | 5.5346 | 64.98 | 0.6246 | 0.6463 | 128.99 |
| | SeeSR (Wu et al., 2024b) | | 50 | 25.18 | 0.7216 | 0.3009 | 0.2223 | 5.4081 | 69.77 | 0.6442 | 0.6612 | 125.55 |
| | PASD (Yang et al., 2025) | | 20 | 25.21 | 0.6798 | 0.3380 | 0.2260 | 5.4137 | 68.75 | 0.6487 | 0.6620 | 124.29 |
| | SUPIR (Yu et al., 2024) | | 50 | 23.61 | 0.6606 | 0.3589 | 0.2492 | 5.8877 | 63.21 | 0.5895 | 0.6709 | 128.35 |
| | OSEDiff (Wu et al., 2024a) | | 1 | 25.15 | 0.7341 | 0.2921 | 0.2128 | 5.6476 | 69.09 | 0.6326 | 0.6693 | 123.49 |
| | AdcSR (Chen et al., 2025) | | 1 | 25.47 | 0.7301 | 0.2885 | 0.2128 | 5.3477 | 69.90 | 0.6353 | 0.6730 | 118.41 |
| | PiSA-SR (Sun et al., 2025) | | 1 | 25.50 | 0.7418 | 0.2672 | **0.2044** | 5.5046 | 70.15 | 0.6551 | 0.6696 | 124.09 |
| | TSD-SR (Dong et al., 2025) | | 1 | 24.81 | 0.7172 | 0.2743 | 0.2105 | **5.1266** | 71.18 | 0.6346 | **0.7160** | **114.45** |
| | InvSR (Yue et al., 2025) | | 1 | 24.30 | 0.7145 | 0.2775 | 0.2060 | 5.7168 | 67.31 | **0.6572** | 0.6734 | 129.52 |
| | CCSR (Sun et al., 2024) | | 1 | 25.92 | 0.7485 | 0.2799 | 0.2122 | 5.7324 | 69.18 | 0.6398 | 0.6336 | 122.98 |
| | ResShift (Yue et al., 2023) | Diffusion model, no T2I prior | 15 | 26.31 | 0.7421 | 0.3421 | 0.2498 | 7.2365 | 58.43 | 0.5285 | 0.5442 | 141.71 |
| | SinSR (Wang et al., 2024b) | | 1 | 26.28 | 0.7347 | 0.3188 | 0.2353 | 6.2872 | 60.80 | 0.5385 | 0.6122 | 135.93 |
| | CTMSR (You et al., 2025) | | 1 | 25.98 | 0.7546 | 0.2897 | 0.2208 | 5.5546 | 64.26 | 0.5270 | 0.6318 | 135.35 |
| | RSD (**Ours**) | | 1 | 25.61 | 0.7420 | 0.2675 | 0.2205 | 5.7500 | 66.02 | 0.5930 | 0.6793 | 138.23 |

# F. Performance-Efficiency Trade-Off for RSD and Recent State-of-the-Art One-Step Diffusion SR Methods

In this section, we discuss the comparison between RSD and very recent SOTA one-step diffusion SR methods - CTMSR (You et al., 2025) and T2I-based SR models, including PiSA-SR (Sun et al., 2025), TSD-SR (Dong et al., 2025), AdcSR (Chen et al., 2025), InvSR (Yue et al., 2025) and CCSR (Sun et al., 2024). We support the comparison with visual results of these models in Figure 5 for the RealLR200 dataset (Wu et al., 2024b) and Figure 6 for the RealLQ250 dataset (Ai et al., 2024), respectively.

## F.1. Comparison with CTMSR

**CTMSR method**. CTMSR proposed a distillation-free method for one-step diffusion SR, which is based on consistency training (Song et al., 2023; Song & Dhariwal, 2024). Their training scheme is split into two stages.

**Stage 1**. In the first stage, they formulate the ResShift forward stochastic diffusion process in Equation (1) as the deterministic trajectory of PF-ODE (Song et al., 2021b); see Equations 8 and 9 in the CTMSR paper. They trained the respective consistency model with 500k iterations using the proposed PF-ODE trajectories and the consistency loss $\mathcal{L}_{\mathrm{CT}}$ according to Equation 10 in the CTMSR paper.

**Stage 2**. After the first stage, CTMSR additionally optimizes the model with the proposed Distribution Trajectory Matching objective. Its idea is to minimize the Distribution Trajectory Distance ($\mathcal{L}_{\mathrm{DTD}}$ in Equations 15 and 16 in the CTMSR paper) between the end points of the real PF-ODE trajectory, which starts from the real HR images, and the fake PF-ODE trajectory, which starts from the predicted fake HR images; see Equations 11, 12, 13, 14 in the CTMSR paper. Computation of the gradient $\nabla_\theta \mathcal{L}_{\mathrm{DTD}}$ with respect to the original CTMSR parameters $\theta$ requires calculating the U-Net Jacobian term. Inspired by SDS (Poole et al., 2023) and VSD (Wang et al., 2023b), CTMSR omits this U-Net Jacobian term. The authors optimized the CTMSR model using the gradients $\nabla_\theta \mathcal{L}_{\mathrm{CT}} + \nabla_\theta \mathcal{L}_{\mathrm{DTD}}$ with additional 2k iterations.

CTMSR uses the same training scheme on the ImageNet dataset with Real-ESRGAN degradations, which is detailed in Section 4.1, as ResShift, SinSR and RSD. Thus, CTMSR can be fairly comparable with these models.

**Perceptual-fidelity comparison**. According to the quantitative results in Tables 1, 3, 13, and 14, RSD has a stable improvement over CTMSR for **all real-world datasets (RealSR, RealSet65, RealSR and DRealSR $512 \times 512$ crops, DRealSR, RealLR200, RealLQ250)** in **most perceptual metrics (LPIPS, DISTS, CLIPIQA, MUSIQ, MANIQA)**. We observe the most notable gaps between the perceptual quality of CTMSR and RSD on the following datasets:

1. RealSR and DRealSR $512 \times 512$ crops in Table 3 - improvement in MANIQA in 0.0660 and 0.0764, respectively, and in CLIPIQA in 0.0475 and 0.0522, respectively.

2. Full-size DRealSR in Table 13 - improvement in MUSIQ in 5.134 and MANIQA in 0.0612, respectively.

3. RealLR200 and RealLQ250 in Table 14 - improvement in MANIQA in 0.0523 and 0.0696, respectively, and in CLIPIQA in 0.0397 and 0.0551, respectively.

These results highlight the strong competitive perceptual performance of RSD among one-step diffusion SR models using the similar UNet architecture with Swin Transformer blocks (Liu et al., 2021) - ResShift, SinSR and CTMSR. However, CTMSR sometimes has better NIQE values and also achieves slightly better CLIPIQA and MUSIQ on the synthetic ImageNet-Test dataset from Table 2. We note that NIQE (Mittal et al., 2013) has been shown to have a worse correlation with human preference compared to recent IQA measures, including MUSIQ, MANIQA, and CLIPIQA, as evident in (Wang et al., 2023a, Tables 1 and 5), (Ke et al., 2021, Table 1), (Yang et al., 2022, Table 3). CTMSR also achieves better fidelity measures (PSNR and SSIM) compared to RSD, which are close to the results of SinSR. Unfortunately, this results in the blur problem of CTMSR, as we discuss below.

**Qualitative comparison**. We provide a visual comparison of RSD with CTMSR, as well as with SinSR, in Figure 5 for the RealLR200 dataset (Wu et al., 2024b) and Figure 6 for the RealLQ250 dataset (Ai et al., 2024), respectively. In Section 4.2, we observed in Figure 3 for RealSet65 (Yue et al., 2023) that CTMSR has blurry artifacts; see the roof of the house in Figure 3. We also observe this result on other images from RealSet65 (Yue et al., 2023); see the bear in Figure 7. Figure 5 shows that RSD has richer textures than CTMSR for the man (top image). For the image of the bird (bottom image), RSD is the only diffusion model without T2I prior, which provides some details of its eye. Similarly, in Figure 6 we observe blur in

the CTMSR images for the rose (top image) and the monkey character (bottom image). We hypothesize that the blur effect of CTMSR is inherited from its consistency training framework, which is based on deterministic sampling from ODE.

**Complexity comparison**. The CTMSR can be fairly compared to the RSD in computational complexity due to the similar architecture following ResShift. For inference complexity, we evaluated the pre-trained CTMSR model using its official implementation listed in Table 12 on the same setup as RSD in Table 4. According to Table 4, CTMSR has a similar number of parameters as ResShift, SinSR, and RSD (172 millions for CTMSR and 174 millions for RSD) and a similar inference time per LR image with resolution $64 \times 64$. However, we also found that CTMSR requires $> 1.5$ GPU memory during inference compared to RSD. As the distillation-free method, CTMSR states that ResShift and its distilled version, SinSR, are limited in two points:

1. **Considerable training costs**. Training SinSR requires the training of the teacher ResShift model and the student SinSR model, while CTMSR is able to train one-step diffusion SR model without an additional distillation stage.

2. **Limitations of the teacher model**. The performance of the student model is limited with the performance of the teacher model.

In Appendices K and G, we agree with the second statement. To verify the first statement, we trained CTMSR with 500k iterations for the first stage and additional 2k iterations for the second stage using their official training code on recommended GPUs (4 NVIDIA A100 GPUs). Following the pre-trained ResShift model, which we used for the distillation with RSD and SinSR, we also trained the ResShift model with 300k iterations using its official training code on the same 4 A100 GPUs to compare the CTMSR training time and the total training time of ResShift and RSD. The results are given in Table 18. Surprisingly, we found that the total training time for ResShift and its distillation with our RSD requires **less training time** than the training time for the distillation-free CTMSR method using the same resources. The training efficiency of the RSD model is supported by its simulation-free property, which is detailed in Appendix C. Compared to the training of the original teacher ResShift model, its distillation with our RSD method requires only the $\approx 15\%$ training time of ResShift, which leads to faster convergence than the distillation-free CTMSR even if we count the training time of ResShift. As noted in Appendix H, the training time of our RSD can be further halved using $K = 1$ without sacrificing quality. These results highlight the strong computational efficiency of RSD compared to CTMSR in both training and inference.

*Table 18.* Training and inference complexity for RSD, ResShift (Yue et al., 2023) and CTMSR (You et al., 2025). All models are trained on the same 4 NVIDIA A100 GPUs using the respective official training code and tested with an LR image of size $64 \times 64$ for SR factor $\times 4$. The inference is done on an NVIDIA A100 GPU. The best values are highlighted in **bold**.

| Methods | ResShift | RSD (**Ours**) | RSD with ResShift training | CTMSR |
|---|---|---|---|---|
| Inference Step (NFE) | 15 | **1** | **1** | **1** |
| Inference Time (s) | 0.643 | **0.059** | **0.059** | **0.059** |
| # Total Param (M) | 174 | 174 | 174 | **172** |
| Maximum GPU memory (MB) | 1167 | **539** | **539** | 904 |
| Training time (hours) | 36 | 5 | 41 | 58 |

### F.2. Comparison with PiSA-SR, TSD-SR, AdcSR, InvSR and CCSR

PiSA-SR (Sun et al., 2025), TSD-SR (Dong et al., 2025), AdcSR (Chen et al., 2025), InvSR (Yue et al., 2025), and CCSR (Sun et al., 2024) are recent SOTA T2I-based SR models, which attempt to resolve the limitations of the OSEDiff model in different aspects.

**Adjustable perception-distortion trade-off and slow training convergence**. OSEDiff does not provide perception-distortion control without re-training, while the training requires 24 hours on 4 NVIDIA A100 GPUs according to Section 4.1 in OSEDiff paper. PiSA-SR proposed a decoupled training approach to train pixel and semantic level LoRA modules (Hu et al., 2022), which allows to adjust the perception-distortion trade-off by different pixel-semantic guidance scales (Ho & Salimans, 2021) during inference without the need of re-training. PiSA-SR uses $\ell_2$ loss for the training of the LoRA module of pixel-level regression and CSD loss (Ho & Salimans, 2021) combined with the LPIPS loss for the training of the LoRA module of semantic level. The CSD loss is computed using the pre-trained Stable Diffusion 2.1-base model and does not require an additional fake model used in OSEDiff, leading to faster training (Ma et al., 2025) according to Figure 9 in the PiSA-SR paper.

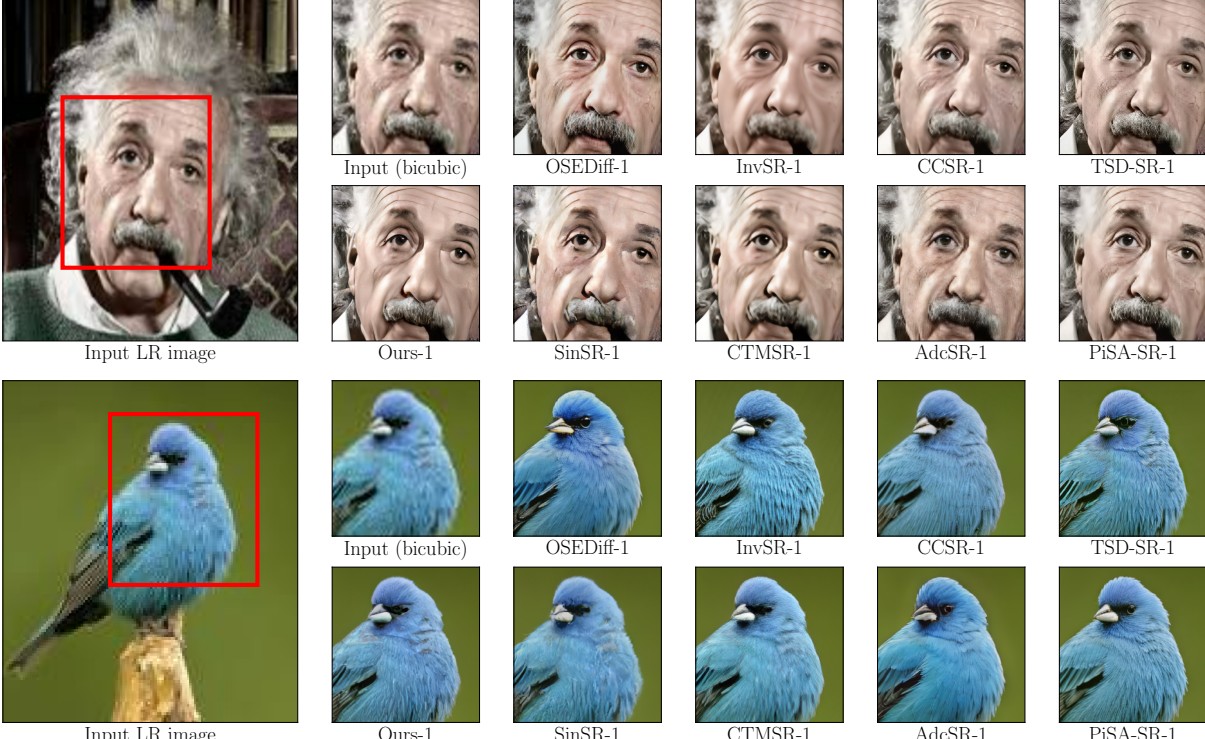

*Figure 5.* Visual results of recent one-step diffusion SR models (RSD, SinSR, CTMSR, OSEDiff, AdcSR, CCSR, InvSR, PiSA-SR, TSD-SR) on full-size images from RealLR200 (Wu et al., 2024b). Please zoom in for a better view.

**Limitations of the VSD objective**. As shown in the TSD-SR paper, the VSD objective used in OSEDiff has two limitations. The first limitation is that the guidance of the teacher is unreliable in scenarios where the initial SR outputs are suboptimal, as visualized in Figure 3 of TSD-SR. To solve this problem, TSD-SR proposed Target Score Matching, which aligns the predictions made by the teacher model on both synthetic and HR latents. The second limitation is that the matching of the score functions predicted by the teacher model and the LoRA model is inconsistent across different timesteps, which is shown in Figure 5 of TSD-SR. To address this issue, TSD-SR proposed the Distribution-Aware Sampling Module, which accumulates optimization gradients for earlier timestep samples in a single iteration, enabling the backpropagation of more gradients focused on detail optimization. TSD-SR initialized all training models from the Stable Diffusion 3 model (Esser et al., 2024).

**Large computational costs of T2I-based SR models**. As observed in AdcSR, the complexity of OSEDiff in terms of parameter number and inference time can still be too high for real deployments, especially on resource-limited edge devices. To reduce the complexity of OSEDiff while maintaining its high perceptual quality, AdcSR proposed an adversarial diffusion compression framework to OSEDiff. The idea of the framework is to train a smaller network after removing unnecessary OSEDiff modules and pruning the remaining modules. The training of AdcSR consists of two stages: 1) pretraining channel-pruned VAE decoder; 2) use of knowledge distillation for OSEDiff with adversarial loss to train a smaller student model.

**Extension to multistep models**. The OSEDiff approach is developed only for the one-step diffusion model, which limits its generation capacity and flexibility for varying perception-distortion requirements. InvSR and CCSR proposed different approaches to enable multistep diffusion models without retraining. InvSR introduces a trainable noise prediction network and reformulates the SR problem as diffusion inversion (Chihaoui et al., 2024). The noise predictor is trained to estimate the noise maps for multiple pre-selected steps via time embedding. To enable arbitrary-step inversion for the inference, InvSR uses the noise map prediction for the initialization the reverse sampling process, where the starting timestep can be freely chosen during inference, resulting in perception-distortion trade-off. CCSR achieves multistep diffusion modeling with another idea. Inspired by the StableSR (Wang et al., 2024a) perception-distortion analysis depending on the diffusion reverse time step in Figure 2 of the CCSR, it proposed to disentangle the SR process into structure generation and detail

enhancement by GAN and DM, respectively.

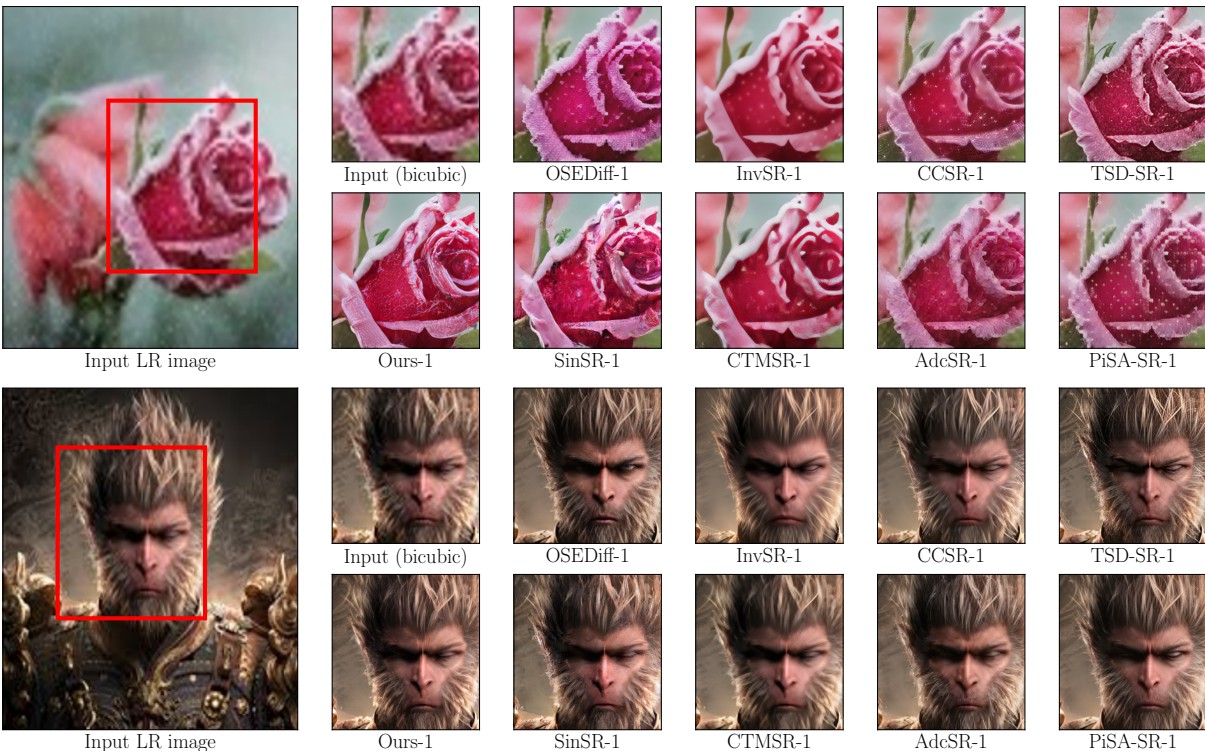

*Figure 6.* Visual results of recent one-step diffusion SR models (RSD, SinSR, CTMSR, OSEDiff, AdcSR, CCSR, InvSR, PiSA-SR, TSD-SR) on full-size images from RealLQ250 (Ai et al., 2024). Please zoom in for a better view.

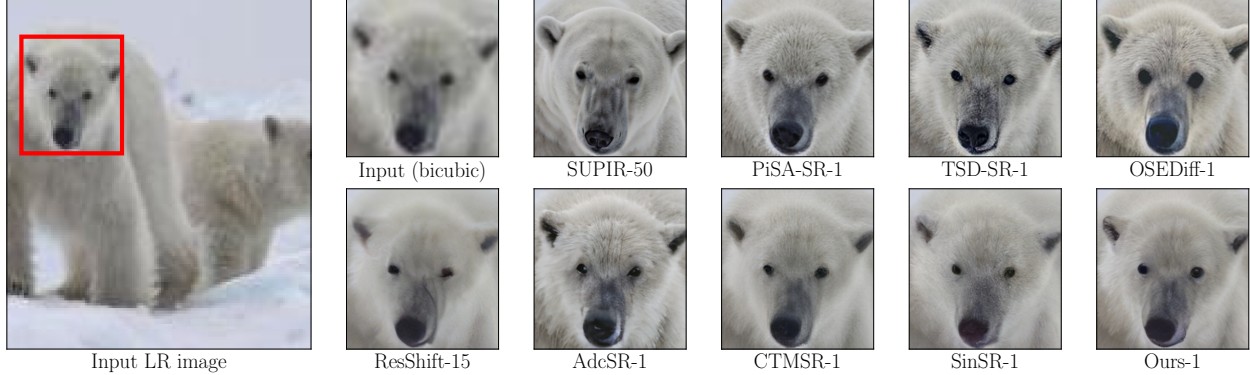

*Figure 7.* Additional comparison on RealSet65 (Yue et al., 2023) for diffusion SR models. Bottom images: ResShift, AdcSR, CTMSR, SinSR, and the proposed RSD. Top images: bicubic LR, SUPIR, PiSA-SR, TSD-SR, and OSEDiff. Please zoom in ×5 times for a better view.

**Quantitative comparison**. In Tables 13, 14, 15, 16, 17, we observe that our RSD combines the high fidelity of relatively small models (ResShift, SinSR, CTMSR) and the good perceptual quality of T2I-based SR models (TSD-SR, PiSA-SR, InvSR, CCSR, AdcSR). TSD-SR, PiSA-SR, InvSR, AdcSR and CCSR further develop T2I-based SR models to improve perceptual and fidelity quality compared to OSEDiff. Compared to these methods, RSD achieves mostly better fidelity consistency with HR images, which is evident by PSNR and SSIM metrics, with yet competitive perceptual metrics (LPIPS, CLIPIQA, MUSIQ).

**Qualitative comparison**. We provide a visual comparison of one-step T2I-based SR models with RSD in Figure 5 for the RealLR200 dataset (Wu et al., 2024b) and Figure 6 for the RealLQ250 dataset (Ai et al., 2024), respectively. Among

these models, the model with the smallest number of parameters, AdcSR, has sharper textures and a better level of detail on most images. However, as the distillation model for OSEDiff, AdcSR can hallucinate for the same images, where OSEDiff hallucinates, see the bear in Figure 7 with an unnatural blue nose for OSEDiff and unnatural fur for AdcSR. Although other SOTA one-step T2I-based SR models, such as PiSA-SR and TSD-SR, also generally have better perceptual quality than RSD, we observe for the rose in Figure 6 that failure cases with blurry effects or unrelated hallucination details occur even for them.

**Complexity comparison**. Despite the good perceptual performance of the TSD-SR, PiSA-SR, AdcSR, CCSR, and InvSR models, these T2I-based models require more computational costs compared to the other one-step T2I-based SR model, OSEDiff, and much more computational costs compared to RSD. In Table 4, we highlight that the T2I-based one-step SR models of PiSA-SR, AdcSR, and TSD-SR require much more GPU memory for inference and a much longer training time compared to RSD. This analysis supports our claim that RSD aims to compromise between fidelity, perceptual quality, and computational efficiency.

## G. Results of RSD Trained on Bigger Resolution

To compare the performance of the RSD, SinSR, and ResShift models for training on high resolution images, we followed the training setup of OSEDiff , which was trained on $512 \times 512$ HR images randomly cropped from LSDIR (Li et al., 2023) with LR images generated via the Real-ESRGAN degradations with the $\times 4$ SR factor. Since the original ResShift model was trained only on $256 \times 256$ HR images, we first trained the ResShift model on $512 \times 512$ HR random crops from LSDIR (Li et al., 2023) for 300k iterations using the source training ResShift code and then distilled it with the RSD and SinSR methods. For a fair comparison, we used the same hyperparameters for RSD and SinSR, which were used for their training on $256 \times 256$ HR images in Table 1 , Table 2 , Table 3 .

We evaluate the trained ResShift, SinSR, and RSD models on full-size images from RealSR (Cai et al., 2019) (the left part of Table 1 ) and provide quantitative results in Table 19, where we also show the results of Real-ESRGAN, BSRGAN, SUPIR, and OSEDiff from Table 15. We provide visual results of those models in Figure 8.

*Table 19.* Results on full-size images from RealSR (Cai et al., 2019). ResShift, SinSR and RSD were trained on $512 \times 512$ HR random crops from LSDIR (Li et al., 2023). The best and second best results are highlighted in **bold** and underline.

| Methods | NFE | PSNR↑ | SSIM↑ | LPIPS↓ | CLIPIQA↑ | MUSIQ↑ |
|---|---|---|---|---|---|---|
| Real-ESRGAN (Wang et al., 2021) | 1 | 25.85 | 0.773 | 0.273 | 0.4898 | 59.678 |
| BSRGAN (Zhang et al., 2021) | 1 | 26.51 | 0.775 | 0.269 | 0.5439 | 63.586 |
| SUPIR (Yu et al., 2024) | 50 | 24.38 | 0.698 | 0.331 | 0.5449 | 63.679 |
| OSEDiff (Wu et al., 2024a) | 1 | 25.25 | 0.737 | 0.299 | **0.6772** | **67.602** |
| ResShift (Yue et al., 2023) | 15 | **27.53** | **0.790** | 0.277 | 0.4988 | 58.034 |
| SinSR (Wang et al., 2024b) | 1 | 27.27 | 0.780 | 0.268 | 0.5503 | 59.478 |
| RSD (**Ours**) | 1 | 26.89 | 0.773 | **0.260** | 0.6103 | 64.987 |

**Comparison with SinSR**. We observe that the RSD achieves better perceptual results, especially in CLIPIQA and MUSIQ, with competitive PSNR and SSIM compared to SinSR. Although ResShift and SinSR have better fidelity metrics, the gap between the visual quality of these models compared with the RSD can be observed in image details, which are sharper for the RSD (compare the jackets in the top of Figure 8).

**Comparison with OSEDiff**. Compared to OSEDiff, RSD trained on the same images from the LSDIR dataset (Li et al., 2023) using HR crops of the same resolution $512 \times 512$ achieves better reference metrics (PSNR, SSIM, LPIPS) but worse no-reference metrics (CLIPIQA, MUSIQ). This is evident in Figure 8, where both OSEDiff and SUPIR models provide details that are not relevant to the LR image (see non-existing inscriptions in the bottom of Figure 8). We highlight that since we did not finetune the hyperparameters of the teacher ResShift model, the quality of RSD is limited by the quality of the ResShift model. The main goal of this study is to show that **RSD achieves better perceptual results compared with SinSR, even when trained on images with higher resolutions**. Improving the perceptual image quality of the RSD when trained on images with higher resolutions to make it closer to the visual quality of T2I-based SR models is promising future work.

**The performance of the RSD model closely mirrors that of its teacher model**. Notably, the RSD model trained at a resolution of $256 \times 256$ demonstrates better performance on no-reference image quality metrics, such as CLIPIQA and MUSIQ (see Table 1). In contrast, the RSD model trained at $512 \times 512$ resolution achieves better results on reference-based metrics, including PSNR, SSIM, and LPIPS (see Table 19). We hypothesize that the observed decline in no-reference metrics, alongside the improvement in reference-based metrics at higher resolutions, is primarily attributed to the behavior of the teacher model, ResShift. Specifically, the ResShift model trained on $256 \times 256$ images yields higher scores on no-reference perceptual quality metrics, whereas the model trained on $512 \times 512$ images performs better on reference-based metrics. The RSD model exhibits the same pattern, which explains the observed trade-off between the two types of evaluation metrics across resolutions.

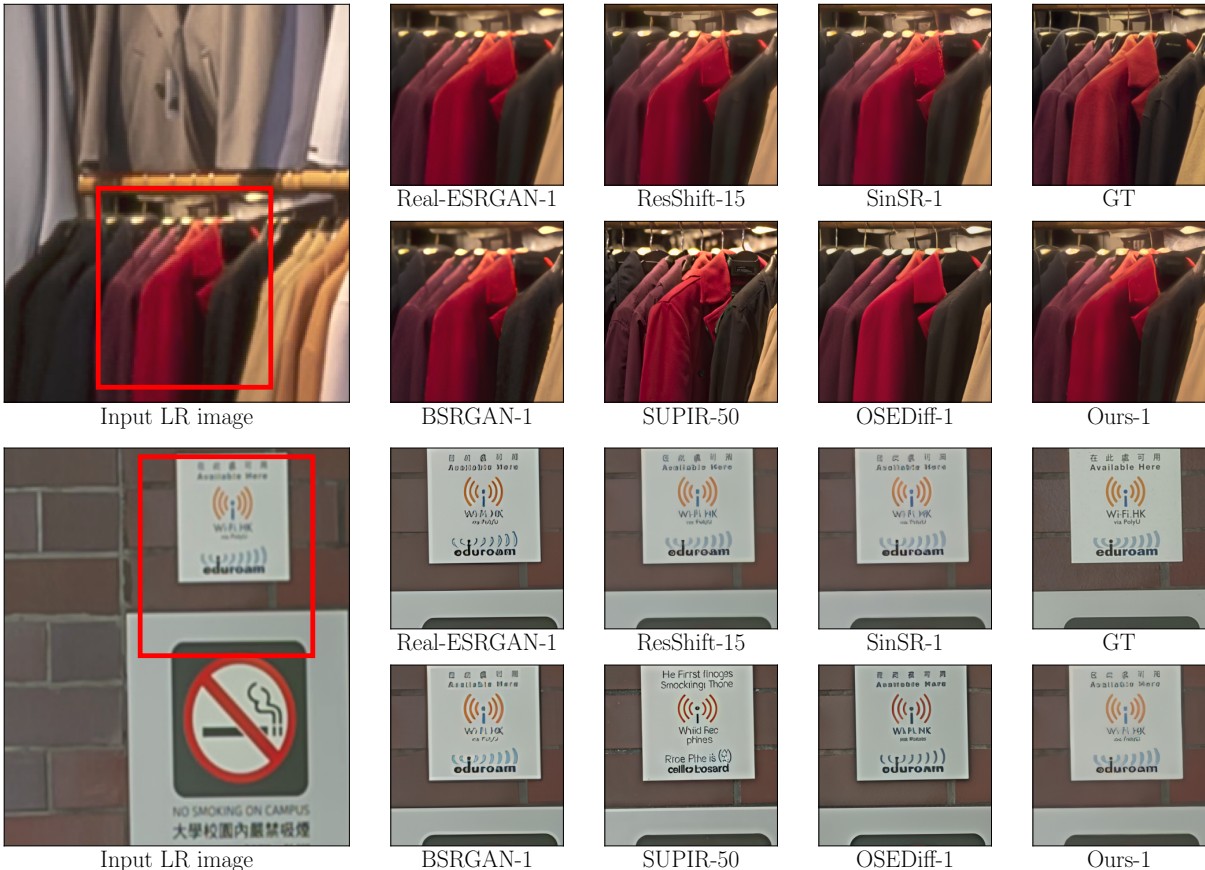

*Figure 8.* Visual results of RSD, ResShift, and SinSR models trained on $512 \times 512$ HR images from LSDIR dataset (Li et al., 2023) and other baselines (Real-ESRGAN, BSRGAN, SUPIR, OSEDiff) on full-size images from RealSR (Yue et al., 2023). Please zoom in for a better view.

## H. Additional Ablation Studies

**Ablation on the hyperparameter $K$.** To assess the sensitivity of RSD to the hyperparameter $K$ in Algorithm 1 for the number of fake ResShift updates per student update, we performed experiments with the final RSD configuration (Tables 1, 2, and 3), along with additional supervised losses for $K \in \{1, 3, 5, 10\}$. We evaluated the trained models on both the full-size RealSR dataset (Table 20, as in Section 4.3) and the ImageNet-Test dataset (Table 21, following Table 2). Across both datasets, all choices of $K$ yield very similar performance: $K = 1$ slightly improves or matches the metrics of $K = 5$, while roughly halving the training time due to fewer fake-model updates. These results indicate that, in the presence of additional ground-truth losses, RSD is largely insensitive to the exact choice of $K$ in this range, so $K$ can be used to optimize computation at the cost of only minor performance changes. We used $K = 5$ to follow the DMD2 strategy for the number of updates of the fake model per student update (Yin et al., 2024a); see Figure 9 in Appendix C of DMD2 for the analysis of the impact of $K$ on training stability for image generation problems. Our results show that RSD training with $K = 1$ and

supervised losses can also be beneficial for Real-ISR problems while not compromising the good performance of $K = 5$.

**Training stability of RSD**. To isolate the effect of $K$ on optimization stability, we further repeated the ablation in a *distill only* setup without supervised losses ((**Ours**, distill only) in Tables 1 and 2). Figure 9 shows the convergence of PSNR and CLIPIQA on the ImageNet-Test dataset for $K \in \{1, 3, 5, 10\}$ in this case. For $K = 1$, the training dynamics become highly unstable, with large oscillations and clearly degraded final metrics, whereas configurations with $K \geq 3$ converge smoothly to similar quality levels. This suggests that supervised losses play a stabilizing role when using small $K$, and that $K = 5$ remains a robust choice in more challenging or purely distillation-based settings, while $K = 1$ is a viable and more efficient alternative in the supervised Real-ISR configuration used in the main experiments. This behavior was also reported in Figure 9 of DMD2.

*Table 20.* Ablation on the hyperparameter $K$ on the RealSR validation set. All runs use the final RSD configuration with supervised losses.

| $K$ | PSNR↑ | SSIM↑ | LPIPS↓ | CLIPIQA↑ | MUSIQ↑ |
|---|---|---|---|---|---|
| 1 | 25.99 | 0.756 | 0.2713 | 0.7159 | 66.247 |
| 3 | 25.86 | 0.752 | 0.2701 | 0.7093 | 66.140 |
| 5 | 25.91 | 0.754 | 0.2726 | 0.7060 | 65.860 |
| 10 | 26.27 | 0.749 | 0.2732 | 0.7135 | 65.233 |

*Table 21.* Ablation on the hyperparameter $K$ on the ImageNet-Test dataset. All runs use the final RSD configuration with supervised losses.

| $K$ | PSNR↑ | SSIM↑ | LPIPS↓ | CLIPIQA↑ | MUSIQ↑ |
|---|---|---|---|---|---|
| 1 | 24.28 | 0.657 | 0.196 | 0.697 | 59.499 |
| 3 | 24.11 | 0.644 | 0.191 | 0.675 | 59.535 |
| 5 | 24.31 | 0.657 | 0.193 | 0.681 | 58.947 |
| 10 | 24.01 | 0.639 | 0.193 | 0.671 | 59.110 |

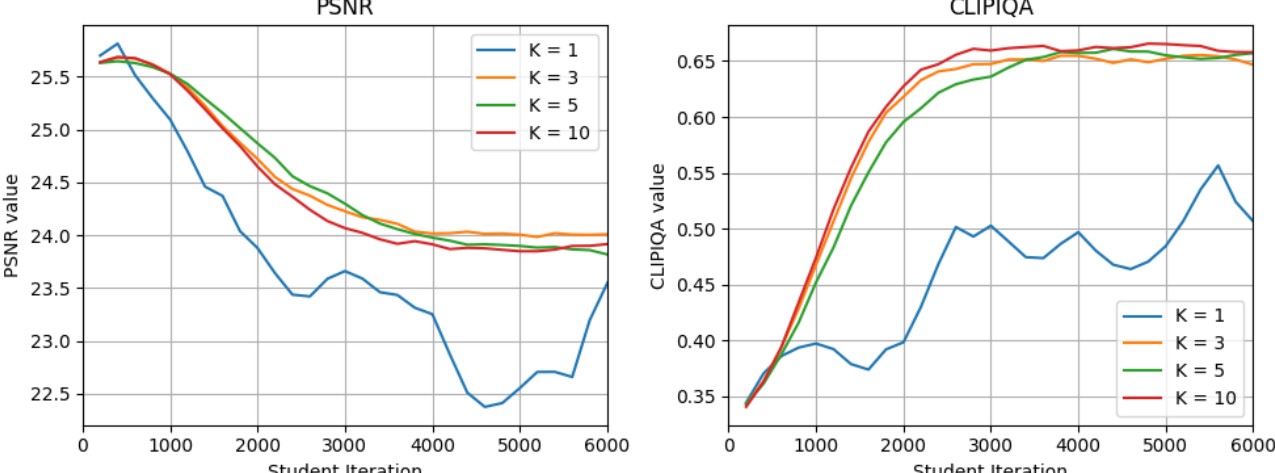

*Figure 9.* Convergence of PSNR and CLIPIQA on the ImageNet-Test dataset for $K \in \{1, 3, 5, 10\}$ when training *distill only* RSD. For $K = 1$, the optimization becomes unstable and fails to reach the quality of configurations with $K \geq 3$.

# I. Comparison with AddSR

It may seem that our primary distillation loss $\mathcal{L}_\theta$ (9) is similar to the SDS loss adopted in ADD (Sauer et al., 2025) $\mathcal{L}_{distill}$ and AddSR (Xie et al., 2024) (Appendix A, (Sauer et al., 2025) and $\mathcal{L}_{ta-dis}$, Equation 1, (Xie et al., 2024)). However, we show that this similarity is only on the surface.

**Conceptual difference in objective functions**. In our work, we propose the RSD loss (24), which is fundamentally different from SDS in both its formulation and practical implications. RSD introduces an auxiliary model to more accurately estimate the score function of the model distribution. This allows for a tighter and lower-variance approximation of the true KL gradient. Moreover, instead of treating each timestep independently as in SDS and VSD (i.e., using marginal KL divergences over $x_t$), our RSD loss is formulated over **the entire trajectory** $x_{0:T}$, leading to a more holistic distillation of the teacher's reverse process.

To facilitate a clear comparison, we summarize the key differences between the loss objectives used in SDS and our proposed RSD in Table 22. We denote by $p^*$ the reverse process of the teacher ResShift model and by $p$ the reverse process of ResShift trained on generator data.

*Table 22.* Comparison of distillation objectives between SDS (Poole et al., 2023), ADD (Sauer et al., 2025), AddSR (Xie et al., 2024) and RSD

| Methods | Objective Function |
|---|---|
| SDS (Poole et al., 2023), ADD (Xie et al., 2024), AddSR (Xie et al., 2024) | $\mathbb{E}_{p(y_0)}\left[\sum_{t=1}^{T} w_t \mathcal{D}_{\mathrm{KL}}\Big(p(x_t|y_0)||p^*(x_t|y_0)\Big)\right]$ |
| RSD (**Ours**, distill-only) | $\mathbb{E}_{p(y_0)}\left[\mathcal{D}_{\mathrm{KL}}\Big(p(x_{0:T}|y_0)\,\|\,p^*(x_{0:T}|y_0)\Big)\right]$ |

**Practical difference**. In addition to theoretical differences between RSD and ADD objectives discussed above, we list practical differences between implementations of RSD and AddSR for real-world SR.

**1. Objective implementation**. The implementation of objective in Eq. 1 in AddSR (Xie et al., 2024) is different from RSD objective in Eq. (13) in many aspects:

1. **RSD does not have any hyperparameters in the distillation loss**. The distillation loss of AddSR, $\mathcal{L}_{ta-dis}$ (Equation 1, (Xie et al., 2024)), requires a weighting function $d(s,t)$ (Equation 3, (Xie et al., 2024)), which is defined by two hyperparameters, $\mu$ and $\nu$. The choice $\mu$ and $\nu$ is based solely on empirical analysis of performance results, as shown in Table 7 and Table 8 of AddSR. In contrast, RSD loss in Eq. (10) relies only on weights $w_t$, which are used for the training of the ResShift model (Eq. 8 in (Yue et al., 2023)). These weights are derived from the theory of DDPM (Ho et al., 2020) and in practice are omitted by ResShift and RSD following the conclusion of DDPM (see Appendix J).

2. **Different supervised losses**. The adversarial loss of AddSR, $\mathcal{L}_{ta-dis}$, follows the hinge loss used in ADD (Sauer et al., 2025). We follow the adversarial loss of DMD2 (Yin et al., 2024a) and use the standard non-saturating loss. We also use LPIPS loss following OSEDiff (Wu et al., 2024a), while AddSR omits it.

3. **Fake model**. Contrary to AddSR, the objective of RSD involves a trainable fake model.

**Architecture**. The major architectural differences are as follows:

1. **RSD does not have any networks related to text-to-image models**. The architecture of generator and fake model in RSD is a UNet model (Ronneberger et al., 2015) following ResShift. We avoid ControlNet (Zhang et al., 2023) and other models used in AddSR.

2. **The sizes of the AddSR and RSD architectures differ by 1 order of magnitude**. In total, the architecture of AddSR requires 2.28B parameters, while RSD requires 174M parameters.

**Empirical results**. We show the comparison between results of 1-step AddSR and RSD in Table 23. It shows that RSD outperforms AddSR in most fidelity and perceptual metrics while having $\times 10$ much fewer parameters.

*Table 23.* Quantitative results of AddSR and RSD models on crops $512 \times 512$ from StableSR (Wang et al., 2024a). The best results are highlighted in **bold**.

| Datasets | Methods | PSNR↑ | SSIM↑ | LPIPS↓ | DISTS↓ | NIQE↓ | MUSIQ↑ | MANIQA↑ | CLIPIQA↑ |
|---|---|---|---|---|---|---|---|---|---|
| DIV2K-Val | AddSR (Xie et al., 2024) | 23.26 | 0.5902 | 0.3623 | 0.2123 | **4.7610** | 63.39 | 0.5657 | 0.5734 |
| | RSD (**Ours**) | **23.91** | **0.6042** | **0.2857** | **0.1940** | 5.1987 | **68.05** | **0.5937** | **0.6967** |
| DRealSR | AddSR (Xie et al., 2024) | **27.77** | **0.7722** | 0.3196 | **0.2242** | 6.9321 | 60.85 | 0.5490 | 0.6188 |
| | RSD (**Ours**) | 27.40 | 0.7559 | **0.3042** | 0.2343 | **6.2577** | **62.03** | **0.5625** | **0.7019** |
| RealSR | AddSR (Xie et al., 2024) | 24.79 | 0.7077 | 0.3091 | **0.2191** | **5.5440** | **66.18** | **0.6098** | 0.5722 |
| | RSD (**Ours**) | **25.61** | **0.7420** | **0.2675** | 0.2205 | 5.7500 | 66.02 | 0.5930 | **0.6793** |

## J. Details of ResShift

As part of the diffusion model class, ResShift can be described by specifying the forward (degradation) process, the parameterization of the reverse (restoration) process, and the objective of training the reverse process.

**Forward process**. Consider a pair of $(\text{LR}, \text{HR})$ images $(y_0, x_0) \sim p_{\text{data}}(y_0, x_0)$. For a residual $e_0 = y_0 - x_0$, ResShift proposes a transition from $x_0$ to $y_0$ with the Markov chain $\{x_t\}_{t=1}^{T}$ of length $T$ through the following Gaussian transition distribution:

$$q(x_t|x_{t-1}, y_0) = \mathcal{N}(x_t|x_{t-1} + \alpha_t e_0, \kappa^2 \alpha_t \mathbf{I}), \tag{39}$$

where:

- $\alpha_t = \eta_t - \eta_{t-1}$ for $t > 1$ and $\alpha_1 = \eta_1$ are defined by the shifting sequence $\{\eta_t\}_{t=1}^{T}$, and $\mathbf{I}$ denotes the identity matrix.

- $\kappa$ is a hyper-parameter controlling the noise variance, and the shifting sequence $\{\eta_t\}_{t=1}^{T}$ monotonically increases with the timestep $t$.

The shifting sequence satisfies $\eta_1 \approx 0$ and $\eta_T \approx 1$, which guarantees the convergence of the marginal distributions of $x_1$ and $x_T$ to approximate distributions of the HR image and the LR image, respectively. Notably, the posterior distribution $q(x_{t-1}|x_t, x_0, y_0)$ for the transition distribution (1) is tractable and can be derived using Bayes's rule:

$$q(x_{t-1}|x_t, x_0, y_0) = \mathcal{N}\left(x_{t-1} \Big| \frac{\eta_{t-1}}{\eta_t} x_t + \frac{\alpha_t}{\eta_t} x_0, \kappa^2 \frac{\eta_{t-1}}{\eta_t} \alpha_t \mathbf{I}\right). \tag{40}$$

**Reverse process.** ResShift suggests the construction of the reverse process to estimate the posterior distribution $p(x_0|y_0)$ in the following parameterized form:

$$p_\theta(x_0|y_0) = \int p(x_T|y_0) \prod_{t=1}^{T} p_\theta(x_{t-1}|x_t, y_0) dx_{1:T} \tag{41}$$

Here $p(x_T|y_0) \approx \mathcal{N}(x_T|y_0, \kappa^2 I)$ and $p_\theta(x_{t-1}|x_t, y_0)$ is the inverse transition kernel from $x_{t-1}$ to $x_t$ with learnable parameters $\theta$. Following DDPM (Ho et al., 2020), ResShift parametrizes this transition kernel with the Gaussian:

$$p_\theta(x_{t-1}|x_t, y_0) = \mathcal{N}(x_{t-1}|\mu_\theta(x_t, y_0, t), \Sigma_\theta(x_t, y_0, t)) \tag{42}$$

**Objective.** To derive the minimization objective for parameters $\theta$, ResShift applies the variational bound estimation on negative log-likelihood for the $p_\theta(x_0|y_0)$, as in DDPM:

$$\min_\theta \mathbb{E}_{(x_0, y_0)} \sum_{t=1}^{T} \mathbb{E}_{x_t \sim q(x_t|x_0, y_0)} \left[ \mathcal{D}_{KL}(q(x_{t-1}|x_t, x_0, y_0) || p_\theta(x_{t-1}|x_t, y_0)) \right] \tag{43}$$

Inspired by the tractable formula for the posterior $q(x_{t-1}|x_t, x_0, y_0)$ in (40), ResShift sets the variance parameter $\Sigma_\theta(x_t, y_0, t)$ to be independent of $x_t$ and $y_0$ and reparametrized the parameter $\mu_\theta(x_t, y_0, t)$ as follows:

$$\Sigma_\theta(x_t, y_0, t) = \kappa^2 \frac{\eta_{t-1}}{\eta_t} \alpha_t \mathbf{I} \tag{44}$$

$$\mu_\theta(x_t, y_0, t) = \frac{\eta_{t-1}}{\eta_t} x_t + \frac{\alpha_t}{\eta_t} f_\theta(x_t, y_0, t), \tag{45}$$

where $f_\theta$ is a deep neural network with parameter $\theta$, aiming to predict $x_0$. Given the Gaussian form of the distributions $q(x_{t-1}|x_t, x_0, y_0)$ (40) and $p_\theta(x_{t-1}|x_t, y_0)$ (42), the objective (43) simplifies as follows:

$$\min_\theta \left[ \sum_{t=1}^{T} w_t \mathbb{E}_{(x_0, y_0, x_t)} \| f_\theta(x_t, y_0, t) - x_0 \|^2 \right], \tag{46}$$

where $w_t = \frac{\alpha_t}{2\kappa^2 \eta_t \eta_{t-1}}$. Empirically, omitting the weight $w_t$ leads to a noticeable improvement in performance, which aligns with the conclusion in DDPM.

## K. Limitations and Failure Cases

Below, we present failure cases for image restoration for RSD and other models. Our method may produce images with mistakes since the teacher model is imperfect. However, we stress that T2I-based SR models with rich priors also face such problems. Specifically, in Figure 10 (top), we observe that the teacher model produces an indistinguishable image compared to the simple bicubic upsampling image. A similar issue occurs with OSEDiff, while all other methods, including ours, SinSR, SUPIR, and GAN-based models, produce images with visible artifacts. Another typical failure case of diffusion-based methods, ResShift, SinSR, and RSD, includes images with rich background details that are hard to predict due to insufficient contextual information in the LR image. As we show in Figure 10 (bottom), the hallucination properties of T2I-based methods, SUPIR and OSEDiff, provide realistic continuations of the road and cars with greater and richer details compared to the results of ResShift, SinSR, and RSD. In Figure 11, we show failure cases of the considered SR methods on the real-world RealSR benchmark (Cai et al., 2019) with available ground truth images. All methods struggle when running on images with many small details, like bush patterns. Hallucinations of diffusion-based methods do not coincide with the original HR image in small details.

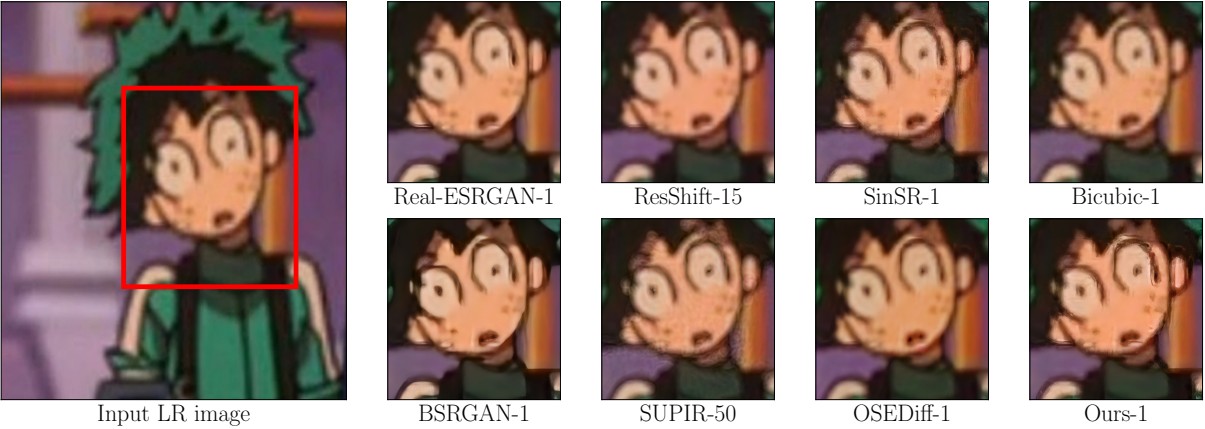

*Figure 10.* Failure cases on images from RealSet65 (Yue et al., 2023). Please zoom in for a better view.

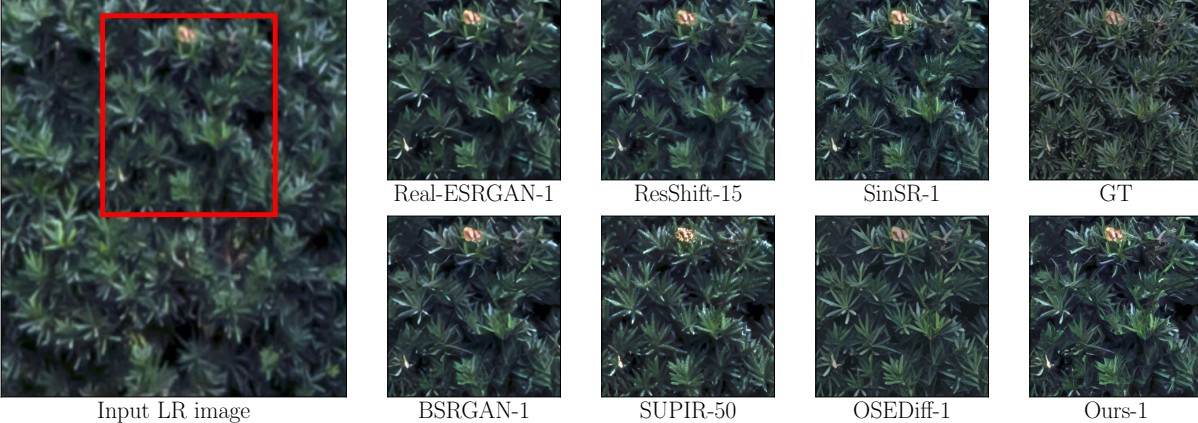

*Figure 11.* Failure cases on images from RealSR (Yue et al., 2023). Please zoom in for a better view.

## L. Proofs

*Motivation for the assumption in Equation* (8). . We prove that

$$f_{G_\theta} = f^* \quad \Rightarrow \quad \mathcal{L}_\theta = 0 \quad \Rightarrow \quad p_\theta(x_0, y_0) = p_{\text{data}}(x_0, y_0) \tag{47}$$

under the additional assumptions:

$$p^*(x_0|y_0) = p_{\text{data}}(x_0|y_0), \tag{48}$$

$$p(x_0|y_0) = p_\theta(x_0|y_0) \tag{49}$$

Practically, assumption in Equation (48) means that the teacher reverse process exactly recovers the data conditional distribution, assumption in Equation (49) means that the fake reverse process recovers the generator distribution ideally. From the definition of the loss $\mathcal{L}_\theta$ in Equation (9), exact matching of the fake and teacher models implies zero loss:

$$f_{G_\theta} = f^* \quad \Rightarrow \quad \mathcal{L}_\theta = 0 \tag{50}$$

From Equation (24), we obtain that $\mathcal{L}_\theta = 0$ leads to

$$p(x_{0:T}|y_0) = p^*(x_{0:T}|y_0) \tag{51}$$

for almost all $y_0$ with respect to $p(y_0)$. We integrate the statement in Equation (51) with respect to $x_{1:T}$:

$$p(x_0|y_0) = p^*(x_0|y_0) \tag{52}$$

holds for almost all $y_0$ with respect to $p(y_0)$. Using assumptions in Equations (48) and (49), we obtain that:

$$p_\theta(x_0|y_0) = p(x_0|y_0) = p^*(x_0|y_0) = p_{\text{data}}(x_0|y_0) \tag{53}$$

holds for almost all $y_0$ with respect to $p(y_0)$. We multiply Equation (53) on the same LR marginal $p(y_0)$ and derive:

$$p_\theta(x_0, y_0) = p_{\text{data}}(x_0, y_0), \tag{54}$$

which justifies the final statement in Equation (47) and motivates the assumption in Equation (8). $\qquad\square$

*Proof of Proposition 3.1.* **First stage.** We first prove that using objective $\mathcal{L}_{\text{fake}}$ (see Eq. (10)) is equivalent to training a fake model $f_\phi$ with objective (7). We recall the $\mathcal{L}_{\text{fake}}$ minimization objective:

$$\arg\min_\phi \mathcal{L}_{\text{fake}}, \tag{55}$$

where

$$\mathcal{L}_{\text{fake}} = \Big( \sum_{t=1}^{T} w_t \mathbb{E}_{p_\theta(\widehat{x}_0, y_0, x_t)} \Big\{ \|f_\phi(x_t, y_0, t)\|_2^2 - 2\langle f_\phi(x_t, y_0, t) + \underbrace{f^*(x_t, y_0, t)}_{\text{Does not depend on } \phi}, \widehat{x}_0 \rangle \Big\} \Big) \tag{56}$$

Then we prove:

$$\arg\min_{\phi} \Big( \underbrace{\sum_{t=1}^{T} w_t \mathbb{E}_{p_\theta(\widehat{x}_0,y_0,x_t)} \|f_\phi(x_t,y_0,t) - \widehat{x}_0\|_2^2}_{\text{Training a fake model } f_\phi \text{ with objective (7)}} \Big) =$$

$$\arg\min_{\phi} \Big( \sum_{t=1}^{T} w_t \mathbb{E}_{p_\theta(\widehat{x}_0,y_0,x_t)} \Big\{ \|f_\phi(x_t,y_0,t)\|_2^2 - 2\langle f_\phi(x_t,y_0,t), \widehat{x}_0\rangle \Big\} +$$

$$\underbrace{\sum_{t=1}^{T} w_t \mathbb{E}_{p_\theta(\widehat{x}_0,y_0,x_t)} \|\widehat{x}_0\|_2^2}_{\text{Does not depend on } \phi} \Big) =$$

$$\arg\min_{\phi} \Big( \sum_{t=1}^{T} w_t \mathbb{E}_{p_\theta(\widehat{x}_0,y_0,x_t)} \Big\{ \|f_\phi(x_t,y_0,t)\|_2^2 - 2\langle f_\phi(x_t,y_0,t), \widehat{x}_0\rangle \Big\} -$$

$$\underbrace{\sum_{t=1}^{T} w_t \mathbb{E}_{p_\theta(\widehat{x}_0,y_0,x_t)} \Big\{ 2\langle f^*(x_t,y_0,t), x_0\rangle \Big\}}_{\text{Does not depend on } \phi} \Big) =$$

$$\arg\min_{\phi} \Big( \sum_{t=1}^{T} w_t \mathbb{E}_{p_\theta(\widehat{x}_0,y_0,x_t)} \Big\{ \|f_\phi(x_t,y_0,t)\|_2^2 -$$

$$2\langle f_\phi(x_t,y_0,t) + \underbrace{f^*(x_t,y_0,t)}_{\text{Does not depend on } \phi}, \widehat{x}_0\rangle \Big\} \Big) =$$

$$\arg\min_{\phi} \mathcal{L}_{\text{fake}}. \tag{57}$$

**Second stage.** Now we prove that:

$$\underbrace{\sum_{t=1}^{T} w_t \mathbb{E}_{p_\theta(\widehat{x}_0,y_0,x_t)} \|f_{G_\theta}(x_t,y_0,t) - f^*(x_t,y_0,t)\|_2^2}_{\mathcal{L}_\theta} =$$

$$-\min_{\phi} \Big\{ \sum_{t=1}^{T} w_t \mathbb{E}_{p_\theta(\widehat{x}_0,y_0,x_t)} \Big( \|f_\phi(x_t,y_0,t)\|_2^2 - \|f^*(x_t,y_0,t)\|_2^2 +$$

$$2\langle f^*(x_t,y_0,t) - f_\phi(x_t,y_0,t), \widehat{x}_0\rangle \Big) \Big\} \tag{58}$$

Note, that since ResShift objective (46) is an MSE, the solution $f_{G_\theta}$ for the data produced by generator $G_\theta$ is given by the conditional expectation as:

$$f_{G_\theta}(x_t,y_0,t) = \mathbb{E}_{p_\theta(\widehat{x}_0|y_0,x_t)}[\widehat{x}_0]. \tag{59}$$

We start from the right part of (58) and transform it back to the left part:

$$-\min_{\phi}\Big\{\sum_{t=1}^{T}w_t\mathbb{E}_{p_\theta(\widehat{x}_0,y_0,x_t)}\Big(-\|f^*(x_t,y_0,t)\|_2^2+\|f_\phi(x_t,y_0,t)\|_2^2+$$

$$2\langle f^*(x_t,y_0,t)-f_\phi(x_t,y_0,t),\widehat{x}_0\rangle\Big)\Big\}=$$

$$\sum_{t=1}^{T}w_t\mathbb{E}_{p_\theta(\widehat{x}_0,y_0,x_t)}\Big\{\|f^*(x_t,y_0,t)\|_2^2-2\langle f^*(x_t,y_0,t),\widehat{x}_0\rangle\Big\}-$$

$$\min_{\phi}\Big\{\sum_{t=1}^{T}w_t\mathbb{E}_{p_\theta(\widehat{x}_0,y_0,x_t)}\Big(\|f_\phi(x_t,y_0,t)\|_2^2-2\langle f_\phi(x_t,y_0,t),\widehat{x}_0\rangle\Big)\Big\}=$$

$$\sum_{t=1}^{T}w_t\mathbb{E}_{p_\theta(y_0,x_t)}\Big(\|f^*(x_t,y_0,t)\|_2^2-2\langle f^*(x_t,y_0,t),\underbrace{\mathbb{E}_{p_\theta(\widehat{x}_0|y_0,x_t)}\widehat{x}_0}_{f_{G_\theta(x_t,y_0,t)}}\rangle\Big)-$$

$$\min_{\phi}\Big\{\sum_{t=1}^{T}w_t\mathbb{E}_{p_\theta(\widehat{x}_0,y_0,x_t)}\Big(\|f_\phi(x_t,y_0,t)\|_2^2-2\langle f_\phi(x_t,y_0,t),\widehat{x}_0\rangle\Big)\Big\}=$$

$$\sum_{t=1}^{T}w_t\mathbb{E}_{p_\theta(y_0,x_t)}\Big(\|f^*(x_t,y_0,t)\|_2^2-2\langle f^*(x_t,y_0,t),f_{G_\theta(x_t,y_0,t)}\rangle\Big)-$$

$$\sum_{t=1}^{T}w_t\mathbb{E}_{p_\theta(y_0,x_t)}\Big(\|f_{G_\theta}(x_t,y_0,t)\|_2^2-2\underbrace{\langle f_{G_\theta}(x_t,y_0,t),f_{G_\theta(x_t,y_0,t)}\rangle}_{\|f_{G_\theta}\|_2^2}\Big)=$$

$$\sum_{t=1}^{T}w_t\mathbb{E}_{p_\theta(y_0,x_t)}\Big(\|f^*(x_t,y_0,t)\|_2^2-2\langle f^*(x_t,y_0,t),f_{G_\theta(x_t,y_0,t)}\rangle+\|f_{G_\theta}(x_t,y_0,t)\|_2^2\Big)=$$

$$\sum_{t=1}^{T}w_t\mathbb{E}_{p_\theta(y_0,x_t)}\underbrace{\|f_{G_\theta}(x_t,y_0,t)-f^*(x_t,y_0,t)\|_2^2}_{\text{Does not depend on }\widehat{x}_0\text{ so we can add }\widehat{x}_0\text{ in expectation.}}=$$

$$\sum_{t=1}^{T}w_t\mathbb{E}_{p_\theta(\widehat{x}_0,y_0,x_t)}\|f_{G_\theta}(x_t,y_0,t)-f^*(x_t,y_0,t)\|_2^2.$$

$\square$

**Discussion**. We explain the intractability of the gradient for $\mathcal{L}_\theta$ in Equation (9) as follows. The gradient of $\mathcal{L}_\theta$ (9) over parameters $\theta$ of $G_\theta$ is given by the chain rule:

$$\frac{d\mathcal{L}_\theta}{d\theta}=\underbrace{\frac{\partial\mathcal{L}}{\partial\theta}}_{\text{direct}}+\frac{\partial\mathcal{L}}{\partial f_{G_\theta}}\cdot\underbrace{\frac{\partial f_{G_\theta}}{\partial\theta}}_{\text{implicit}} \tag{60}$$

$\frac{d\mathcal{L}_\theta}{d\theta}$ contains an implicit term, which requires a differentiation through the full ResShift training loop and is computationally infeasible (because one needs to backpropagate through all the gradient updates used to train $f_{G_\theta}$):

$$\frac{\partial f_{G_\theta}}{\partial\theta}=\frac{\partial}{\partial\theta}\underbrace{\Big[\arg\min_{\phi}[\mathcal{L}(\phi,\theta)]\Big]}_{\text{Training on the Generator outputs}} \tag{61}$$

In contrast, our Proposition 3.1 and Equation (10) resolve this by deriving the mathematically equivalent form of Equation (9), which does not require directly differentiating through the "re-training" step $f_{G_\theta}$ i.e., it does not contain terms with argmin operations.

