# OpenReview forum: "One-Step Residual Shifting Diffusion for Image Super-Resolution via Distillation"
_ICML.cc/2026/Conference — ICML 2026 regular_

### Official Review · Reviewer_CvfC · 2026-03-03

**Soundness:** 2
**Presentation:** 3
**Significance:** 2
**Originality:** 3
**Overall Recommendation:** 4
**Confidence:** 3

**Summary:**

This paper proposed RSD (Residual Shifting Distillation), which is a one-step super-resolution distillation approach.
The core contribution is the objective for 1-step distillation of diffusion SR models in discrete time and derive its tractable version.

RSD outperforms the teacher model (ResShift) by a noticeable margin in the perceptual metrics.

**Compliance With Llm Reviewing Policy:**

Affirmed.

**Final Justification:**

This paper presents a new discrete-time distillation objective (RSD Loss). From a model distillation viewpoint, this contribution is valuable. However, regarding practical image super-resolution and restoration, its real-world usefulness is limited.
The authors' rebuttal has addressed most of my concerns.

**Key Questions For Authors:**

Please see in the Weaknesses.

**Limitations:**

Yes

**Strengths And Weaknesses:**

Strengths:
1. The paper introduces a new discrete-time distillation objective (RSD Loss) and, through Proposition 3.1, derives a computable closed-form expression for it, which avoids the infeasible requirement of backpropagating through the entire ResShift training process.

2. The Fake ResShift framework is able to capture the teacher’s full diffusion behavior rather than relying on a single‑step approximation.

3. The paper presents comprehensive experiments with thorough comparisons, and the proposed method even surpasses the teacher model in perceptual scores.

Weaknesses:
1. The core challenge of Real-ISR is degradation mismatch, yet the paper does not evaluate unseen degradations.
2. The improvement in perceptual metrics may stem from the LPIPS and GAN losses rather than from the RSD Loss itself.
3. Limited practical relevance in the current landscape
Modern unified image-editing or T2I-based models (Qwen-image-edit, Flux.2) already cover super-resolution, restoration, and enhancement with stronger robustness and better real-world performance. The paper does not discuss this shift, and the practical value of a specialized SR diffusion model is increasingly limited.

---

> ### Author Rebuttal · Authors · 2026-03-30
>
> Dear Reviewer CvfC, thank you for your comments. Here are the answers to your questions and comments.
>
> **(1) ...The paper does not evaluate unseen degradations...**
>
> We highlight that we evaluate RSD and other methods in Tabs. 1, 3, and 14 **exclusively on unseen test degradations**, including 5 real-world datasets (RealSet65, RealSR, DRealSR, RealLR200, RealLQ250). They have become standard for the evaluation of real-world SR methods and our baselines; see lines 234-244. The degradation mismatch arises from training using only synthetic Real-ESRGAN degradations on ImageNet.
>
> **(2) ...The improvement in perceptual metrics may stem from the LPIPS and GAN losses rather than from the RSD Loss itself...**
>
> **Improvement in perceptual metrics using only RSD loss**. As we discuss in the answer to Question 1 of Reviewer D9nn, the proposed RSD objective for Real-ISR problems outperforms other diffusion distillation SR approaches, such as knowledge distillation with SinSR and VSD distillation of ResShift, for the same ResShift teacher in perceptual metrics **without any supervised losses**. Thus, the improvement comes from the new distillation formulation, not from supervised losses (lines 261-269).
>
> **Influence of supervised losses**. We evaluate the influence of supervised losses for the RSD model in Tab. 6 with a discussion in lines 408-415 and 1980-1987. As discussed in Section 3.4, we use supervised LPIPS and GAN losses because teacher predictions may be biased by approximation errors; this is consistent with the improvement in PSNR and LPIPS in Tab. 6. We highlight that the use of supervised losses based on HR images **is standard for diffusion distillation baselines**; see Eq. 5 in OSEDiff and Eq. 8 in SinSR.
>
> **(3) ...Limited practical relevance in the current landscape Modern unified image-editing or T2I-based models (Qwen-image-edit, Flux.2) already cover super-resolution, restoration, and enhancement with stronger robustness and better real-world performance. The paper does not discuss this shift, and the practical value of a specialized SR diffusion model is increasingly limited...**
>
> We discuss two aspects of modern text-conditioned image editing models that limit their applications to Real-ISR. Our observations for the Qwen-Image Edit model coincide with the conclusions of Section 3 of the work [1] for the Nano Banana Pro model [2].
>
> **Severe hallucinations and poor fidelity**. Tab. 2 and Figs. 6-8 in [1] show that Nano Banana Pro produces unexpected effects in Real-ISR, such as unintended field-of-view expansion (Fig. 6 in [1]); we observed similar issues for Qwen-Image Edit [3]. As a result, the SR results of these models lack pixel-level and perceptual alignment with the reference image.
>
> **High computational cost**. These models are too expensive for real-time and high-resolution inputs. Qwen-Image Edit (40 NFE) requires about 48 seconds and 57 GB on a $128 \times 128$ LR image using 1 NVIDIA A100 GPU, whereas RSD (1 NFE) requires only 0.12 seconds and 1.1 GB on the same setup. The problem for Qwen-Image Edit becomes even more severe for degraded images of higher resolution: the running time of this model on full size LR image from DRealSR of resolution $900 \times 1400$ is 45 minutes using the same resources.
>
> We compare RSD with Nano Banana Pro and Qwen-Image Edit-2511 on the same evaluation setup. For Nano Banana Pro, we use the results from Tab. 2 in [1] and evaluate Qwen-Image Edit with the same prompt on $512 \times 512$ HR crops from DIV2K-Val, DRealSR, and RealSR datasets:
>
> |Dataset|Method|PSNR↑|SSIM↑|LPIPS↓|MUSIQ↑|CLIPIQA↑|
> |---|---|---|---|---|---|---|
> |DIV2K-Val|RSD|23.91|0.6042|0.2857|68.05|0.6967|
> |DIV2K-Val|Qwen-Image Edit|20.46|0.4964|0.5516|47.30|0.3308|
> |DIV2K-Val|Nano Banana Pro|20.29|0.4720|0.3645|65.40|0.5257|
> |DRealSR|RSD|27.40|0.7559|0.3042|62.03|0.7019|
> |DRealSR|Qwen-Image Edit|22.72|0.6437|0.4387|51.26|0.3949|
> |DRealSR|Nano Banana Pro|23.97|0.6323|0.3809|59.00|0.5145|
> |RealSR|RSD|25.61|0.7420|0.2675|66.02|0.6793|
> |RealSR|Qwen-Image Edit|20.78|0.6138|0.3012|59.54|0.4220|
> |RealSR|Nano Banana Pro|23.56|0.6649|0.2978|60.18|0.5199|
>
> **Summary**. RSD outperforms Qwen-Image Edit and Nano Banana Pro in all considered metrics by a significant margin. Our specialized SR diffusion model, RSD, offers a better perception-distortion trade-off, is much faster, and has much lower computational requirements. We will add these results to the revised text.
>
> **References**.
> [1] Is Nano Banana Pro a Low-Level Vision All-Rounder?
> A Comprehensive Evaluation on 14 Tasks and 40 Datasets. Zuo et al., arXiv, 2025.
>
> [2] Gemini: a family of highly capable multimodal models. Gemini Team, 2023.
>
> [3] Qwen-Image Technical Report. Qwen Team, 2025.
>
> **Concluding remarks**. We would be grateful if you could let us know if our explanations have been satisfactory. If so, we kindly ask that you consider increasing your rating. We are also open to discussing any other questions you may have.

---

> > ### Author Rebuttal · Reviewer_CvfC · 2026-04-03
> >
> > Thank you for the rebuttal. I still have a few additional questions.
> >
> > What I mean by "unseen degradations" refers to testing in fully in-the-wild scenarios, such as certain AIGC images. Since the authors construct their training data using the RealESRGAN degradation model, existing datasets (e.g., RealSet65, RealSR) should already include RealESRGAN-based comparisons.
> >
> > **Regarding the fidelity issues of current editing models**: editing models may indeed behave inconsistently, possibly because they are not trained for a single dedicated task. However, if the same training data are used, a simple LoRA fine-tuning could potentially yield strong results. In addition, I personally feel that the current design of RSD is somewhat overly complex, which may hinder future follow-up work and large-scale adoption.
> >
> > **Regarding the computational cost of current editing models**: indeed, Qwen-Image-Edit is quite large. However, we tested Flux.2-Klein-4B (4-step version), and its runtime for a 1024×1024 image on an A100 is around 7 seconds, which is generally acceptable.
> >
> > Given the nature of image super-resolution, I recommend evaluating RSD at higher output resolutions, such as upscaling to 2048×2048.

---

> > > ### Author Response · Authors · 2026-04-05
> > >
> > > Dear Reviewer CvfC, here are the answers to your questions and comments.
> > >
> > > **Benchmarks in Tabs. 1, 13, and 14 are not synthetic datasets with Real-ESRGAN degradation**. The RealSR and DRealSR datasets are real-world in the sense that LR distortions arise from **real-world cameras** (lines 044-048). In both datasets, LR-HR pairs were obtained by **changing the focal length** of real-world DSLR cameras. ResShift introduces RealSet65, and SeeSR/DreamClear introduce RealLR200/RealLQ250 as additional **real-world LR benchmarks**. In these benchmarks, some LR images were taken from DPED and Real47 to capture degradations from real cameras, while other LR images were collected from the internet. For this reason, our evaluation is **standard** for the classical Real-ISR problem and follows prior work. We compare RSD with Real-ESRGAN and other GAN-based models in Tabs. 13 and 15-17.
> > >
> > > **AIGC evaluation is a different setting**. Our paper studies a **classical Real-ISR**, i.e., the restoration of degraded LR observations from real DSLR cameras. As we understand your question, AIGC means that LR images are produced by generative models, such as text-to-image models. The evaluation of SR for AIGC LR images significantly depends on models, prompts, and noise realizations. We could not find any reproducible AIGC benchmark for SR with specific LR image sets during the rebuttal. Without HR images, the evaluation should use only no-reference metrics, as in our Tabs. 1 (RealSet65) and 14 (RealLR200 and RealLQ250).
> > >
> > > **Comparison with LoRA finetuned models**. We highlight that Tabs. 1–3 already include multiple LoRA-adapted T2I-based SR methods (lines 137-146): OSEDiff, PiSA-SR, AdcSR (SD-2.1 base), and TSD-SR (SD3). In our response to Reviewer Qp4R, we further include S3Diff (SD-Turbo with LoRA).  Results in Tab. 17 justify the statement that these models achieve much better fidelity compared to Qwen-Image Edit and Nano Banana Pro without LoRA finetuning. We also acknowledge that recent T2I-based methods such as PiSA-SR and TSD-SR provide strong perceptual results (lines 400-403 and 1800-1803). However, they remain computationally heavy and still exhibit failure cases (blur or unrelated hallucinations, Fig. 6). As discussed in our responses to Reviewers Qp4R and D9nn, RSD provides a better fidelity–perception balance with much lower computational costs: higher PSNR on ImageNet-Test and RealSR, improved LPIPS on ImageNet-Test, and competitive no-reference perceptual metrics on real-world data (lines 368-383 and 1790-1795). Moreover, even with optimizations such as pruning (AdcSR) and quantization (PassionSR),  these models still require much more resources: RSD uses x7 times less GPU memory than AdcSR (Tab. 4) and 27\% fewer parameters than PassionSR (our response to Reviewer Qp4R) while maintaining competitive quality.
> > >
> > > **Comparison with Flux models**. We thank Reviewer CvfC for pointing out Flux.2-Klein-4B model (4 NFE).
> > > While its inference is indeed faster than Qwen-Image Edit, it remains significantly more expensive than RSD: 6 s and 8 GB vs. 0.12 s and 1.1 GB on a single A100 (for $128\times128$ LR input). We evaluate the Flux.2-Klein-4B model using the same prompt on $512 \times 512$ HR crops from DIV2K-Val, DRealSR, and RealSR, as for Qwen-Image Edit and Nano Banana Pro.
> > >
> > > |Dataset|Method|PSNR↑|SSIM↑|LPIPS↓|MUSIQ↑|CLIPIQA↑|
> > > |---|---|---|---|---|---|---|
> > > |DIV2K-Val|RSD|23.91|0.6042|0.2857|68.05|0.6967|
> > > |DIV2K-Val|Flux.2-Klein-4B|15.55|0.3549|0.4683|68.71|0.6767|
> > > |DRealSR|RSD|27.40|0.7559|0.3042|62.03|0.7019|
> > > |DRealSR|Flux.2-Klein-4B|15.87|0.4125|0.5342|68.73|0.7354|
> > > |RealSR|RSD|25.61|0.7420|0.2675|66.02|0.6793|
> > > |RealSR|Flux.2-Klein-4B|14.75|0.3833|0.4714|70.79|0.7374|
> > >
> > > Flux produces visually rich but excessive details, which is similar to SUPIR (lines 372-375) and leads to high no-reference and poor reference-based metrics. In our response to Reviewer jmA4, we compare RSD with FluxSR (LoRA-finetuned Flux.1-dev), which improves fidelity but still suffers from a high number of parameters (12B) and periodic artifacts. Overall, RSD provides a better performance-efficiency trade-off.
> > >
> > > **Evaluation of RSD at higher resolutions**. We note that our evaluation of RSD includes full size DRealSR (Tab. 13), which consists of 93 HR images with a resolution around $3500 \times 4000$. In addition, other real-world datasets also contain images with a $1500 \times 1500$ resolution after upscaling, including 39 images in RealSR, 18 images in RealSet65, and 11 images in RealLR200.
> > >
> > > **Clarity and generalization of RSD**. We clarify the training procedure of RSD in Algorithm 1 and App. C provides implementation details, while our supplementary material contains the reproducible code. In Sec. 3, we note that RSD can be generalized to DDPM-based models, and App. A.4 clarifies the details of RSD adaptation for I2SB and LDM models. Thus, we provide all necessary tools and clarifications for further follow-up works and large-scale adoption.

---

### Official Review · Reviewer_D9nn · 2026-03-10

**Soundness:** 3
**Presentation:** 3
**Significance:** 3
**Originality:** 3
**Overall Recommendation:** 3
**Confidence:** 3

**Summary:**

This paper studies how to enable efficient one-step inference for diffusion-based real image super-resolution models while maintaining strong perceptual quality. It proposes RSD (Residual Shifting Distillation), whose core idea is not to directly force the student network to match the teacher outputs, but instead to train a fake ResShift model using data generated by the student, and then make this fake model approximate the teacher model, thereby constructing an optimizable distillation objective. The method further incorporates multi-timestep training, LPIPS perceptual loss, and GAN loss to improve perceptual quality and generation stability.
Experiments are conducted on multiple real and synthetic datasets, including RealSR, RealSet65, ImageNet-Test, DIV2K-Val, and DRealSR, and comparisons are made against ResShift, SinSR, CTMSR, OSEDiff, SUPIR, AdcSR, PiSA-SR, and TSD-SR. The results show that, compared with the most directly related methods, especially SinSR and ResShift, RSD achieves relatively consistent improvements on perceptual metrics such as LPIPS, CLIPIQA, and MUSIQ, while maintaining low parameter count, low memory usage, and efficient one-step inference, demonstrating a favorable performance–efficiency trade-off.

**Compliance With Llm Reviewing Policy:**

Affirmed.

**Key Questions For Authors:**

please see Strengths And Weaknesses

**Limitations:**

please see Strengths And Weaknesses

**Strengths And Weaknesses:**

Strengths
1.	The problem is practically meaningful: The paper focuses on the core tension in diffusion-based super-resolution models, namely their high perceptual quality but heavy computational cost. This is a well-motivated problem with clear research and practical significance, especially for real image super-resolution, where efficiency and deployment cost are critical considerations. The motivation and solution direction are straightforward: how to combine the strong perceptual quality of diffusion models with the high efficiency of one-step inference in real-world SR.
2.	The method contains a certain degree of novelty: Rather than applying standard output-level distillation, RSD introduces a fake ResShift model to impose an indirect distribution-level constraint. Compared with direct supervised distillation, this design is more structured and also distinguishes the method more clearly from the VSD-style approaches discussed in the paper.
3.	The experiments are fairly thorough and include a broad range of baselines: The paper compares not only with closely related methods such as ResShift, SinSR, and CTMSR, but also with strong T2I-based methods such as OSEDiff, SUPIR, AdcSR, PiSA-SR, and TSD-SR. The evaluation metrics cover fidelity, perceptual quality, and distribution quality, making the overall experimental setup relatively comprehensive.
4.	The efficiency advantage is clear, giving the method strong practical value: In terms of complexity, RSD is significantly better than large-scale T2I-based models in parameter count, memory usage, and training/inference cost, while still maintaining competitive perceptual quality. This gives the method strong practical deployment potential.
5.	The gains over the most relevant baseline are relatively consistent: One of the most convincing aspects of the paper is that, compared with SinSR, which is arguably the most directly related baseline, RSD achieves consistent improvements across multiple real-world datasets and perceptual metrics. This suggests that the proposed method indeed brings effective gains within the one-step ResShift distillation paradigm.
Weaknesses
1.	The methodological novelty is still somewhat limited: Although the paper proposes a new distillation objective, the overall framework still falls within the broader paradigm of improving and integrating existing diffusion distillation approaches. Components such as the fake model, perceptual loss, and GAN constraints all have related precedents in prior work. As a result, the contribution is better characterized as an effective improvement rather than a substantial conceptual breakthrough.
2.	The performance upper bound is constrained by the teacher model: Since RSD is essentially a distillation of ResShift, its performance ceiling is limited by the teacher’s capability. The paper also acknowledges this point in the conclusion. This means that while the method is effective for lightweight deployment, it is unlikely to fundamentally surpass stronger large-scale T2I-based models.
3.	It does not achieve comprehensive superiority over the strongest baselines: The main advantage of RSD lies in efficiency and resource cost, whereas in absolute perceptual quality it does not consistently and comprehensively outperform strong baselines such as PiSA-SR, TSD-SR, and OSEDiff. Therefore, the method is better positioned as a stronger trade-off solution rather than a new absolute SOTA.
4.	The theoretical analysis remains somewhat limited in depth: The theoretical part mainly provides an objective derivation and a tractable reformulation, but offers limited insight into why this fake-model distillation is particularly suitable for super-resolution, why it may be more robust than other distillation objectives, or what deeper optimization and generalization mechanisms are at play.

---

> ### Author Rebuttal · Authors · 2026-03-30
>
> Dear Reviewer D9nn, thank you for your comments. Here are the answers to your questions and comments:
>
> **(1)...The methodological novelty is still somewhat limited...**
>
> (1) Our main contribution is a **new discrete-time objective for one-step distillation** of diffusion SR models and its tractable reformulation (lines 80–109). Crucially, the gains come **without supervised losses**. In Tab. 1, with the same ResShift teacher and only distillation losses, **RSD (Ours, distill only)** outperforms SinSR (distill only) in all perceptual metrics, with a notable margin in CLIPIQA (+0.14) and MUSIQ (+9). Similarly, RSD outperforms ResShift-VSD on RealSR in all metrics, with a significant margin in PSNR (+1dB), SSIM (+0.08), and LPIPS (-0.11). Although fake models were previously used in OSEDiff with a VSD objective, our improvement comes from a new distillation formulation, not from LPIPS or GAN losses (lines 261-269). (2) We also contribute a **new derivation of the VSD objective** for diffusion SR without a T2I prior (ResShift-VSD; Algorithm 2), with conceptual and computational comparisons to RSD in Appendix A.1 (Section 3.6). We show that our **discrete-time distillation objective** outperforms continuous-time IBMD objective: Tab. 7 shows better results in all metrics, and Tab. 8 shows substantially lower training and inference costs.
>
> **(2) ...The performance upper bound is constrained by the teacher model ... [RSD] is unlikely to fundamentally surpass stronger large-scale T2I-based models...**
>
> As noted in lines 133–137, we do not claim that small diffusion SR models should outperform T2I-based SR models perceptually, which are pretrained on billions of real images. Non-T2I models (ResShift, SinSR, CTMSR, RSD) used only ImageNet data (lines 228–233), so comparing absolute quality alone is not fully fair. Accordingly, we also evaluate RSD in the setting used by prior non-T2I diffusion SR work, which mainly compared against earlier T2I-based methods such as LDM and StableSR. In this setting, Tabs. 16–17 show that RSD reaches comparable perceptual quality while using only **1 NFE** (Tab. 4).
>
> **(3) ...It does not achieve comprehensive superiority over the strongest baselines...**
>
> Our claim is a better **performance-efficiency trade-off** among diffusion SR models with small training and inference cost. Tabs. 1–3 show that RSD narrows the perceptual gap with T2I-based SR (e.g. OSEDiff) better than other lightweight non-T2I models. Tab. 4 and lines 427–435 further show that recent T2I-based methods require much more training time, memory, and parameters.
>
> Compared with OSEDiff (SD-based), RSD achieves **comparable** perceptual quality in LPIPS, CLIPIQA, and MUSIQ (Tabs. 1–2, 13), although Tab. 3 is mixed since the models are trained at different resolutions. Visual examples also show that OSEDiff can hallucinate nonexistent details (Figs. 1, 6–7). In efficiency, OSEDiff requires 5× longer training, 7× more inference GPU memory, and 10× more parameters than RSD (Tab. 4).
>
> We acknowledge that recent T2I-based methods such as **PiSA-SR** and **TSD-SR** achieve stronger perceptual quality and improve over OSEDiff (lines 400-403). However, they remain much heavier and still exhibit failure cases such as blur or unrelated hallucinations (Fig. 6). TSD-SR has 13x more parameters, PiSA-SR requires 9× more inference GPU memory. We offer a practical compromise: better perceptual quality than prior lightweight SR models, at far lower cost than large T2I-based models (Tabs. 13–17).
>
> **(4) ...The theoretical analysis remains somewhat limited in depth...**
>
> We use ResShift as the teacher because its diffusion process is specifically derived for SR (Appendix I), with a tractable posterior (Eq. 39), boundary conditions linking $x_{1}$ and $x_{T}$ to approximate HR and LR distributions, and a lower-bound-based training objective (Eq. 42). The RSD loss $\mathcal{L}_{\theta}$  in Eq. 7 is motivated by the assumption in Eq. 6; the conditions under which this holds are clarified in our response to Reviewer jmA4's Question 1. Directly optimizing Eq. 7 is computationally infeasible (lines 206–210; App. K). The fake model introduced in Prop. 3.1 yields an equivalent but tractable objective (Eq. 8), which is precisely why we used it for SR.
>
> RSD is also distinct from existing distillation objectives. Compared with SinSR’s knowledge distillation, RSD is **simulation-free** and allows 3× faster training (lines 418–424 and 1280–1297). Compared with VSD, the differences are discussed in our response to Question 1 and Appendix A.1; Appendix H further contrasts RSD with ADD. Most importantly, RSD is defined over the **full trajectory**  $x_{0:T}$, whereas SDS/VSD-style methods treat timesteps independently.
>
> **Concluding remarks**. We would be grateful if you could let us know if our explanations have been satisfactory. If so, we kindly ask that you consider increasing your rating. We would be happy to address any further questions.

---

> > ### Author Rebuttal · Reviewer_D9nn · 2026-04-01
> >
> > The author has largely addressed my concerns. Considering the other reviewers' comments and the author's responses, I have decided to maintain my original score.

---

> > > ### Author Response · Authors · 2026-04-05
> > >
> > > We thank Reviewer D9nn for fruitful discussion and for noting that our rebuttal has largely addressed theirs concerns. We appreciate that update.

---

### Official Review · Reviewer_Qp4R · 2026-03-11

**Soundness:** 3
**Presentation:** 4
**Significance:** 3
**Originality:** 4
**Overall Recommendation:** 4
**Confidence:** 3

**Summary:**

This paper presents RSD, a new distillation method for ResShift. This method is based on training the student network to produce images such that a new fake ResShift model trained on them will coincide with the teacher model. RSD achieves single-step restoration and outperforms the teacher by a noticeable margin in various perceptual metrics (LPIPS,CLIPIQA,MUSIQ).It shows that the distillation method can surpass SinSR, the other distillation-based method for ResShift, making it on par with state-of-the-art diffusion SR distillation methods with limited computational costs in terms of perceptual quality.

**Compliance With Llm Reviewing Policy:**

Affirmed.

**Key Questions For Authors:**

Please view weakness part.

**Limitations:**

Please view weakness part.

**Strengths And Weaknesses:**

Strengths
This paper proposes a new method, i.e., One-Step Residual Shifting Diffusion for Image Super-Resolution (RSD). RSD achieves the   competitive perceptual quality while requiring substantially lower computational cost and budget, bringing diffusion SR closer to real-time applications

Weakness

The quantitative results should be provided for each method in Figure 3 for compressive comparisons.  The difference of visual results are hardly to distinguished, please highlight their difference for easily viewing.

Some visual results of ablation study should be provided, rather than only quantitative results.

Some one-step based image super-resolution methods should be compared and discussed, such as:
[1] One-step effective diffusion network for real-world image super-resolution, NeurIPS, 2024
[2] Tsd-sr: One-step diffusion with target score distillation for real-world image super-resolution, CVPR 2025
[3] Degradation-guided one-step image super-resolution with diffusion priors, Arxiv, 2024
[4] Passionsr: Post-training quantization with adaptive scale in one-step diffusion based image super-resolution, CVPR, 2025
[5] Sinsr: diffusion-based image super-resolution in a single step, CVPR, 2024.

The future work should be discussed.

---

> ### Author Rebuttal · Authors · 2026-03-30
>
> Dear Reviewer Qp4R, thank you for your comments. Here are the answers to your questions and comments.
>
> **(1.a) "... The quantitative results should be provided for each method in Fig. 3 for compressive comparisons ..."**
>
> The quantitative results for all methods in Fig. 3 are provided in Tabs. 1-4, 13-14.
>
> **(1.b) "...The difference of visual results are hardly to distinguished, please highlight their difference for easily viewing ..."**
>
> We give a discussion and highlight the difference in visual results between RSD and the baselines in Fig. 3 in lines 380-406 ("Qualitative comparisons" in Section 4.2), lines 1698-1707 ("Qualitative comparisons" in App. E.1), and lines 1796-1804 ("Qualitative comparisons" in App. E.2). A summary is that our closest competitors with comparable architectures (ResShift, SinSR, CTMSR) suffer from blur, and models with complex T2I-based architectures (SUPIR, OSEDiff, AdcSR, PiSA-SR, TSD-SR) can hallucinate excessive details.
>
> **(1.c) "... Some visual results of ablation study should be provided ..."**
>
> We provide visual results for our ablation studies in Figs. 8 and 9 in App. F. Fig. 8 shows the training convergence of the PSNR and CLIPIQA metrics for different numbers of updates of the fake model per student update, $K$. Fig. 9 supports the quantitative results of Tab. 6 for the impact of supervised losses.
>
> **(2) ... Some one-step based image super-resolution methods should be compared and discussed, such as ...**
>
> We note that one-step OSEDiff [1], TSD-SR [2], and SinSR [5] are already included among our main baselines. We report comprehensive quantitative comparisons with these and other one-step SR methods across an extensive set of validation benchmarks, with results presented in Tabs. 1-3 of the main text and Tabs. 13-17, 23 of the appendix. These comparisons are discussed in Section 4.2 and Apps. D, E, and H.
>
> Compared with SinSR, RSD consistently achieves stronger perceptual quality on both synthetic and real-degradation benchmarks, with clear gains in LPIPS, CLIPIQA, and MUSIQ across ImageNet-Test, RealSR, and RealSet65 (lines 320-328). Relative to OSEDiff and TSD-SR, RSD offers a better fidelity–perception balance: it achieves stronger PSNR/SSIM on ImageNet-Test and RealSR, improves LPIPS on ImageNet-Test and cropped RealSR, and remains competitive in no-reference perceptual metrics on real-world data (lines 368-383 and 1790-1795).
>
> For the remaining methods [3, 4], we report the metrics from the original papers as follows. For S3Diff [3], we report the results for $512 \times 512$ crops from RealSR and DRealSR, which follows our Tab. 3. For PassionSR [4], we report the results for full size images of RealSR and DRealSR, which follows our Tabs. 1 and 13.
>
> **Comparison with S3Diff on cropped RealSR / DRealSR**:
>
> | Dataset | Method | PSNR↑ | SSIM↑ | LPIPS↓ | MUSIQ↑ | CLIPIQA↑ |
> |---|---|---:|---:|---:|---:|---:|
> | RealSR | RSD | 25.61 | 0.7420 | 0.2675 | 66.02 | 0.6793 |
> | RealSR | S3Diff | 25.03 | 0.7321 | 0.2699 | 67.89 | 0.6722 |
> | DRealSR | RSD | 27.40 | 0.7559 | 0.3042 | 62.03 | 0.7019 |
> | DRealSR | S3Diff | 26.89 | 0.7469 | 0.3122 | 64.19 | 0.7122 |
>
> **Comparison with PassionSR on full-size RealSR / DRealSR**:
>
> | Dataset | Method | PSNR↑ | SSIM↑ | LPIPS↓ | MUSIQ↑ | CLIPIQA↑ |
> |---|---|---:|---:|---:|---:|---:|
> | RealSR | RSD | 25.91 | 0.754 | 0.273 | 65.860 | 0.7060 |
> | RealSR | PassionSR | 25.67 | 0.7499 | 0.3140 | 65.88 | 0.6912 |
> | DRealSR | RSD | 27.66 | 0.7864 | 0.3105 | 38.340 | 0.7398 |
> | DRealSR | PassionSR | 27.41 | 0.8146 | 0.3422 | 33.56 | 0.7554 |
>
> As shown above, RSD consistently outperforms S3Diff on the cropped RealSR/DRealSR benchmarks in full-reference metrics (PSNR, SSIM, LPIPS), while the no-reference metrics are mixed. Under the full-size protocol, RSD outperforms PassionSR in PSNR and LPIPS on both RealSR and DRealSR, while the no-reference metrics remain mixed.
>
> In addition, compared to our method, PassionSR and S3Diff require more computing resources: S3Diff is built on SD-Turbo, a distilled variant of Stable Diffusion 2.1 (around 1B parameters in total); PassionSR, a pruned and quantized OSEDiff version, uses 238M parameters. Our RSD model uses only 174M parameters (Tab. 4).
>
> Overall, these comparisons further support that RSD remains competitive with recent one-step SR methods.
>
> **(3) ... The future work should be discussed...**
>
> We discuss future work in Section 5. In this section, we note that a promising direction for our method is to apply it to a more advanced teacher.
>
> **Concluding remarks**. We would be grateful if you could let us know if our explanations have been satisfactory. If so, we kindly ask that you consider increasing your rating. We would be happy to address any further questions.
>
> **References**. [3] Degradation-guided one-step image super-resolution with diffusion priors. arXiv, 2024.
>
> [4] PassionSR: Post-training quantization with adaptive scale in one-step diffusion based image super-resolution. CVPR, 2025.

---

> > ### Author Rebuttal · Reviewer_Qp4R · 2026-04-03
> >
> > I have review the response from authors. My main concerns are solved and I decide to maintain my score.

---

> > > ### Author Response · Authors · 2026-04-05
> > >
> > > We thank Reviewer Qp4R for fruitful discussion and for indicating that theirs concerns have been fully resolved. We appreciate that update.

---

### Official Review · Reviewer_jmA4 · 2026-03-14

**Soundness:** 2
**Presentation:** 2
**Significance:** 2
**Originality:** 2
**Overall Recommendation:** 3
**Confidence:** 4

**Summary:**

This paper studies one-step diffusion distillation for real-world image super-resolution. The research intends to present a notable topic: how to retain the perceptual quality of diffusion-based SR while reducing inference to a single step. The core idea is to distill a ResShift teacher into a stochastic one-step generator by training the generator such that a “fake” ResShift model trained on the generator’s outputs matches the teacher; the resulting objective is presented as a tractable surrogate for an otherwise intractable bilevel training problem. The method is further enhanced with multistep training during learning and optional LPIPS / GAN losses. Empirically, the paper reports that RSD improves substantially over SinSR and often offers a better efficiency–quality trade-off than much larger T2I-based SR systems such as OSEDiff and SUPIR, while using only about 174M parameters and low inference memory.

**Compliance With Llm Reviewing Policy:**

Affirmed.

**Final Justification:**

I still keep my rating. Please refer to Rebuttal Acknowledgment.

**Key Questions For Authors:**

1. Can the authors better justify Equation (6)? Under what assumptions does matching the fake model to the teacher imply matching the underlying conditional data distribution? Is there any weaker statement that can be proven?
2. How robust is the method to teacher quality? The conclusion states that performance is limited by teacher capacity. Have the authors tried distilling from a stronger SR teacher or mixing teachers, and does the proposed objective remain stable?
3. Could you compare more one-step image SR methods? The survey of one-step image SR methods is insufficient.

**Limitations:**

Please refer to Weaknesses and Key Questions For Authors

**Strengths And Weaknesses:**

Strengths:
1. The paper targets a real bottleneck in diffusion SR: existing methods either remain slow or produce overly smooth / hallucinated outputs.
2. Instead of directly imitating teacher outputs, the method introduces a fake ResShift model and derives a tractable objective connected to KL matching of full trajectories.
3. The paper studies multistep training and the effects of LPIPS/GAN losses, which helps separate the impact of the distillation-only version from the final model.

Weaknesses:
1. The main theoretical assumption is strong and insufficiently validated. Equation (6) assumes that if the fake ResShift trained on generated data matches the teacher, then the generated LR–HR distribution approximately matches the real data distribution. This is the conceptual hinge of the method, but the paper does not provide a convincing justification beyond empirical evidence. A mismatch between model equivalence and data-distribution equivalence is quite plausible in high-dimensional conditional generation.
2. The paper itself notes that RSD is trained on 256×256 ImageNet crops, while OSEDiff is trained on 512×512 LSDIR crops aligned with some evaluation settings.
3. The method may inherit a hard ceiling from ResShift unless paired with a stronger teacher.
4. RSD clearly improves over SinSR in perceptual metrics, but it does not dominate all methods. In several settings, T2I-based models still achieve better CLIPIQA/MUSIQ or stronger fidelity on selected datasets, and RSD can underperform ResShift/CTMSR in PSNR/SSIM.

---

> ### Author Rebuttal · Authors · 2026-03-30
>
> Dear Reviewer jmA4, thank you for your comments. Here are the answers to your questions and comments.
>
> **(1) ...Can the authors better justify Equation (6)? Under what assumptions does matching the fake model to the teacher imply matching the underlying conditional data distribution?...**
>
> As per request, we provide a theoretical justification that, under mild additional assumptions, matching the fake model to the teacher implies matching the underlying data distribution.
>
> We prove:$$f_{G_{\theta}} = f^{\*} \Rightarrow  \mathcal{L}\_{\theta}=0 \Rightarrow p_{\theta}(x_{0},y_{0})=p_{\text{data}}(x_{0},y_{0})$$under the additional assumptions$$\text{(A1)}\quad p^{\*}(x_{0}|y_{0})=p\_{\text{data}}(x_{0}|y_{0}),\qquad \text{(A2)}\quad p(x_{0}|y_{0})=p\_{\theta}(x_{0}|y_{0}).$$ Practically, (A1) means that the teacher reverse process exactly recovers the data conditional distribution, while (A2) means that the fake reverse process recovers the generator distribution ideally.
>
> **Proof.** From Eq. 7, exact matching of the fake and teacher models implies $\mathcal{L}\_{\theta}=0$. Eq. 22 gives$$\mathcal{L}\_{\theta}=\mathbb{E}\_{p(y_{0})}D_{\text{KL}}(p(x_{0:T}|y_{0})\|\|p^{\*}(x_{0:T}|y_{0})).$$Hence $\mathcal{L}\_{\theta}=0$ implies$$p(x_{0:T}|y_{0})=p^{\*}(x_{0:T}|y_{0}).$$Marginalizing over $x_{1:T}$ yields
> $$p(x_{0}|y_{0})=p^{\*}(x_{0}|y_{0}).$$ Using (A1) and (A2), we obtain $$p\_{\theta}(x_{0}|y_{0})=p(x_{0}|y_{0})=p^{\*}(x_{0}|y_{0})=p_{\text{data}}(x_{0}|y_{0}).$$Since both sides use the same LR marginal $p(y_{0})$, it follows that$$p\_{\theta}(x_{0},y_{0})=p_{\text{data}}(x_{0},y_{0}).$$Thus, under (A1) and (A2), matching drifts restores the coupling.
>
> **(2)  ...How robust is the method to teacher quality? Have the authors tried distilling from a stronger SR teacher or mixing teachers, and does the proposed objective remain stable?...**
>
> For a fair comparison with SinSR, we used the same ResShift teacher and did not study other teacher frameworks.
> App. G reports a stronger teacher trained on $512\times512$ LSDIR HR crops (Tab. 21), and the corresponding RSD student outperforms the ImageNet $256\times256$ counterparts (Tab. 1). Distilling ResShift with RSD retains a compact SR architecture (Tab. 4) while reducing the perceptual gap to T2I-based SR models. Thus, the choice of the ResShift teacher is consistent with our goal of improving the performance-efficiency trade-off for one-step diffusion SR models.
>
> **(3) ...The survey of one-step image SR methods is insufficient...**
>
> In Tabs. 1-2, we compare RSD with 6 one-step SR models (OSEDiff, AdcSR, PiSA-SR, TSD-SR, SinSR, CTMSR).
> App. D (Tabs. 13-17) further reports InvSR, CCSR, GAN- and transformer-based baselines; in the answer to Reviewer Qp4R, we also report S3Diff and PassionSR results. We consider our comparison with one-step SR models to be sufficient. **Could you please explain what makes our comparison with one-step SR methods insufficient?**
>
> **(4) ...T2I-based models still achieve better CLIPIQA/MUSIQ or stronger fidelity on selected datasets, and RSD can underperform ResShift/CTMSR in PSNR/SSIM...**
>
> **Note on comparison with T2I-based models**. (1) In lines 103-109, we position our RSD as a method with better perceptual results from the class of **lightweight** models, but not as an improvement over T2I-based SR quality. This is achieved in Tabs. 1-3, where we show that our RSD has the lowest gap in perceptual quality between T2I-based SR models and lightweight models with similar resources. In lines 427-435, we highlight that recent T2I-based SR methods require substantially more training time and inference GPU memory (Tab. 4). (2) We note that **each non-T2I model** does not dominate all T2I-based SR models in perceptual quality (Tabs. 1-3, 13-14), which is expected given that T2I models are trained on much larger datasets (lines 133-137); so comparison in absolute quality alone is not fully fair. In addition, earlier non-T2I models were compared only with StableSR and LDM (lines 251--254), whereas RSD is **competitive** with the stronger OSEDiff (lines 368-379).
>
> **Perception-distortion trade-off**. Eq. 9 shows that RSD minimizes the KL divergence between the ResShift teacher and student distributions. By the perception-distortion trade-off (Theorem 1 in Blau \& Michaeli, 2018), this can reduce PSNR, so ResShift may achieve better PSNR than RSD. The PSNR of RSD can be improved by multistep training with a larger $N$ (Tab. 5). Prior works (SUPIR, TSD-SR) show that PSNR and SSIM often misalign with human perception. In Figs. 3 and 5-7, ResShift and CTMSR often produce blurrier details (lines 394-397 and 1698-1707), while quantitatively RSD improves over them in LPIPS, CLIPIQA, and MUSIQ (lines 321-329 and 1677-1697).
>
> **Concluding remarks**. We would be grateful if you could let us know if our explanations have been satisfactory. If so, we kindly ask that you consider increasing your rating. We would be happy to address any further questions.

---

> > ### Author Rebuttal · Reviewer_jmA4 · 2026-04-02
> >
> > It is easy to do a survey on recent one-step diffusion SR methods during the rebuttal. However, there are no comparisons and discussions on these methods in the rebuttal, eg, D3SR[1], FLuxSR [2], etc. In addition, it would be better to search more related work.
> >
> > [1] Unleashing the Power of One-Step Diffusion based Image Super-Resolution via a Large-Scale Diffusion Discriminator. NeurIPS 2025
> > [2] One diffusion step to real-world super-resolution via flow trajectory distillation. ICML 2025

---

> > > ### Author Response · Authors · 2026-04-04
> > >
> > > Dear Reviewer jmA4, we discuss D³SR and FluxSR as follows.
> > >
> > > **We highlight that all 6 mentioned one-step diffusion SR baselines are recent and reproducible**. They include AdcSR (CVPR 2025), PiSA-SR (CVPR 2025), TSD-SR (CVPR 2025), CTMSR (ICCV 2025), and, in the appendix, InvSR (CVPR 2025) and CCSR (TIP 2026). Crucially, due to our main goal of improving the performance-efficiency trade-off (lines 107-109), all these methods are **reproducible**, and we can evaluate their quality (Tabs. 1-3, 14-17) and efficiency (Tab. 4). We discuss in detail the choice of these baselines in Appendix E.
> > >
> > > **Reproducibility limitations of D³SR and FluxSR**. Both public GitHub repositories of D³SR and FluxSR still do not contain pre-trained models, training, or inference code, which has also been reported by other users in the issues section. Thus, we were not able to directly evaluate the performance-efficiency trade-off for them and compare it with RSD.
> > >
> > > **Discussion on computational cost**. However, both models use the T2I prior with a much larger number of parameters. FluxSR is based on FLUX.1-dev, which its paper describes as having 12B parameters, and D³SR reports 966M inference parameters, while RSD uses 174M. D³SR also reports 100K training iterations on 4 NVIDIA A100-40GB GPUs using the SD-2.1-base generator, roughly comparable to  OSEDiff’s 1-day training. Our RSD was trained for 5 hours on 4 NVIDIA A100 GPUs. FluxSR does not report training and inference costs, but it requires sampling 2400 noise-image pairs with FLUX.1-dev and reports 30k training iterations for a 12B-parameter FLUX-based generator. The paper explicitly mentioned high computational costs of FluxSR in the conclusion. Therefore, like other T2I-based methods in Tab. 4, both are expected to require substantially more training compute cost and inference GPU memory than RSD.
> > >
> > > **Performance comparison**. We provide a quantitative comparison of RSD with FluxSR and D³SR using common datasets and metrics for all three models, including full-size RealSR and RealSet65 datasets. We use the results of Tabs. 1-2 in their papers, along with our Tab. 1. As shown below for RealSR, RSD has better reference-based metrics than both FluxSR and D3SR (PSNR, SSIM, LPIPS), while FluxSR and D³SR have better no-reference MUSIQ.
> > >
> > > |Method|PSNR↑|SSIM↑|LPIPS↓|MUSIQ↑|
> > > |--------|------:|------:|-------:|--------:|
> > > |RSD|25.91|0.754|0.273|65.86|
> > > |FluxSR|24.83|0.718|0.320|68.95|
> > > |D³SR|24.11|0.715|0.296|68.23|
> > >
> > >  For RealSet65, the methods report   MUSIQ 70.75 (FluxSR) and 70.25 (D³SR), which are slightly better compared to RSD (69.17). These results are consistent with our analysis of the comparison of RSD with other recent one-step T2I-based SR models (lines 366-384).
> > >
> > > **Improvements in the class of no-T2I models**. We found that the FluxSR repository provides reproducible SR outputs on the full size DRealSR (Table 13), RealLR200, and RealLQ250 (Table 14). To further support our claim regarding the performance-efficiency improvement of RSD, we compare one-step models without T2I prior (RSD, SinSR, CTMSR) with FluxSR on these datasets.
> > >
> > > Results on DRealSR:
> > >
> > > |Method|PSNR↑|SSIM↑|LPIPS↓|DISTS↓|CLIPIQA↑|MUSIQ↑|NIQE↓|MANIQA↑|
> > > |--------|------:|------:|-------:|--------:|------:|------:|------:|------:|
> > > |SinSR|27.32|0.7233|0.4452|0.2368|0.7223|32.800|5.5748|0.4757|
> > > |CTMSR|28.28|0.8017|0.3355|0.1946|0.6821|33.206|4.7795|0.4702|
> > > |RSD|27.66|0.7864|0.3105|0.1868|0.7398|38.340|4.6098|0.5314|
> > > |FluxSR|25.92|0.7592|0.3618|0.1928|0.7347|37.287|4.6947|0.5566
> > >
> > > Results on RealLR200:
> > >
> > > |Method|CLIPIQA↑|MUSIQ↑|NIQE↓|MANIQA↑|
> > > |--------|------:|------:|-------:|--------:|
> > > |SinSR|0.7089|64.90|5.3329|0.5561|
> > > |CTMSR|0.6754|67.63|4.2943|0.5426|
> > > |RSD|0.7151|68.66|4.7074|0.5949|
> > > |FluxSR|0.7101|71.60|5.1905|0.6117|
> > >
> > > Results on RealLQ250:
> > >
> > > |Method|CLIPIQA↑|MUSIQ↑|NIQE↓|MANIQA↑|
> > > |--------|------:|------:|-------:|--------:|
> > > |SinSR|0.7142|65.29|5.4630|0.5294|
> > > |CTMSR|0.6701|68.07|4.5831|0.5130|
> > > |RSD|0.7252|69.63|4.5531|0.5826|
> > > |FluxSR|0.7374|72.65|5.3973|0.5901|
> > >
> > > Results on no-reference datasets of RealLR200 and RealLQ200 show that RSD has the smallest gap in perceptual quality among the class of lightweight one-step diffusion SR models with respect to the expensive FluxSR. Results on the full size DRealSR show that RSD remains competitive compared to FluxSR, where it outperforms FluxSR on 7 of 8 metrics.
> > >
> > > **Summary**. We thank Reviewer jmA4 for the notice on D³SR and FluxSR, and we will add their discussion to the final version. However, due to the absence of reproducible code, it is impossible to provide a full and fair comparison with them. The provided comparison on common datasets justifies the main claims regarding the performance-efficiency positioning of RSD in relation to recent one-step T2I-based SR models (lines 071-109).

---

### Decision · Program_Chairs · 2026-04-30

**Decision:**

Accept (regular)

**Comment:**

This paper proposes an effective RSD for image super-resolution and  improves the trade-off between fidelity, perceptual quality, and computational efficiency for Real-ISR. It receives reviews with mixed ratings. The concerns of Reviewer jmA4 mainly include limited theoretical justifications of the proposed methods, e.g., Equation (6), robustness to the teacher quality, and more comparisons with one-step image SR methods. The concerns of Reviewer Qp4R include limited visual difference against compared methods, insufficient visual results of ablation study, and more comparisons with one-step image SR methods. Reviewer D9nn pointed out that the novelty of the paper is limited. The performance is unlikely to fundamentally surpass stronger large-scale T2I-based models due to the limit of adopted the teacher model. The performance improvement is marginal. The theoretical analysis remains somewhat limited in depth. Reviewer CvfC pointed out that the paper does not evaluate unseen degradations. The improvement in perceptual metrics may stem from the LPIPS and GAN losses rather than from the RSD Loss itself. The proposed method has limited practical relevance.

After rebuttal, Reviewer Qp4R and Reviewer CvfC recommended acceptance as the authors solve their most concerns. Reviewer D9nn also pointed out that the authors have largely addressed his/her concerns. For the Reviewer jmA4, the comparisons and discussions with D3SR, FLuxSR should be added. The AC checks the reviews\&rebuttal and finds that these concerns can be addressed in the final version.